# Optimal Regret and Hard Violation for Constrained Markov Decision Processes with Adversarial Losses and Constraints

**Srinjoy Roy**                                                                                     *srinjoyroyonline@gmail.com*
*Chennai Mathematical Institute, Siruseri, India*
*Institute for Advancing Intelligence, TCG-CREST, Kolkata, India*

**Swagatam Das**                                                                                  *swagatam.das@isical.ac.in*
*Electronics and Communication Sciences Unit*
*Indian Statistical Institute, Kolkata, India*

**Reviewed on OpenReview:** *https://openreview.net/forum?id=EsInBaXOko*

## Abstract

We investigate online learning in finite-horizon episodic Constrained Markov Decision Processes (CMDPs) under the most demanding setting: adversarial losses and constraints, bandit feedback, and unknown transitions. The most popular approaches, such as primal-dual or linear programming, either rely on Slater's condition (which can yield vacuous bounds) or require solving a complex optimization problem at each round. Inspired by the groundbreaking work of Sinha & Vaze (2024) in Constrained Online Convex Optimization (COCO), we map the CMDP instances to a corresponding COCO problem, thus creating simple and elegant algorithms that require only a single Euclidean projection per episode. Our algorithm first attains $\widetilde{\mathcal{O}}(\sqrt{T})$ regret and $\widetilde{\mathcal{O}}(\sqrt{T})$ hard cumulative constraint violation for adversarial losses and constraints, unknown transition dynamics, bandit feedback, without Slater's condition and also without access to a strictly feasible policy. We achieve $\mathcal{O}(\sqrt{T})$ regret and $\widetilde{\mathcal{O}}(\sqrt{T})$ hard violation for known transitions. Additionally, we study the remaining three permutations of known-unknown transitions and full-bandit feedback, again achieving optimal regret and hard violation bounds in each case. Besides closing several gaps in the literature, our simple construction of biased estimators for the sub-gradient could be of independent interest for didactic purposes. Finally, we conducted rigorous experiments on several CMDP instances to verify our theoretical results from a practical perspective.

## 1 Introduction

The arrival of AlphaGo (Silver et al., 2017) ignited an unprecedented curiosity about the capabilities of Reinforcement Learning (RL) (Sutton & Barto, 2018) among researchers. Numerous works highlight that RL is remarkably effective across multiple domains, including games (Jaderberg et al., 2019; Mathieu et al., 2023), robotic locomotion (Smith et al., 2024), control (Hegde et al., 2024; Du et al., 2023), and Large Language Models (LLMs) such as GPT-4 (OpenAI et al., 2024) and DeepSeek-V3 (DeepSeek-AI et al., 2024). Quite naturally, a comprehensive understanding of Markov Decision Processes (MDPs) (Puterman, 2014) is essential, as they lie at the core of any RL problem. In other words, RL addresses a sequential decision-making problem by learning an optimal policy; thus, MDPs are used to model any RL task. The ultimate goal in vanilla RL is to discover a policy that maximizes the expected cumulative reward. However, in many real-world scenarios, such as self-driving cars and recommender systems, the agent is often required to satisfy both safety and budget constraints in addition to maximizing reward. For instance, autonomous vehicles should not meet with an accident or crash (Wen et al., 2020), and bidding parties in an auction cannot exceed a budget (He et al., 2021). To address such scenarios, the Constrained Markov Decision Process (CMDP) (Altman, 1999) serves as an excellent tool, as it naturally incorporates constraints within

the classical MDP framework. In contrast to MDPs, the objective in CMDPs is to learn a policy that maximizes the expected cumulative reward, subject to satisfying the constraints.

Online learning in finite-horizon episodic CMDPs, a topic that has long piqued the community's interest (Wei et al., 2018; Efroni et al., 2020; Müller et al., 2024), is the central theme of our work. This setting necessitates that the learner's objective be to minimize both the *regret* and the *cumulative constraint violation* (also referred to as *violation* for brevity). The regret quantifies the difference between the learner's cumulative loss and the optimal policy's cumulative loss. Specifically, the optimal policy is the best-in-hindsight policy that satisfies the constraints during learning. On the other hand, the cumulative constraint violation tracks the total sum of constraint violations across all episodes. Both the regret and the cumulative violation should ideally be sublinear in $T$, i.e., the total number of episodes. We mention specific directions from the vast literature of online learning in CMDPs (see Section 2 for detailed related works) that have been instrumental in motivating this paper:

1. **Hard/Soft Violations:** Many works on CMDPs are bothered with *soft constraint violations* (Efroni et al., 2020; Qiu et al., 2020), in which the effect of the positive violations is nullified (or diminished) by the negative ones across the whole learning process (Ghosh et al., 2022; Wei et al., 2023). Such nullifications are absolutely impractical in real-world environments. On the contrary, *hard constraint violations* (Stradi et al., 2025b) are a significantly stronger and practical constraint violation condition that solely cares about the positive violations. An example: let a CMDP model a clinical trial for a newly discovered drug, where each episode represents treating a patient. The aim is to minimize disease symptoms, and the constraint is to keep the probability of causing a severity below 1%. Say, in the first episode, the drug causes a hemorrhage to the patient, incurring a massive constraint violation of $+0.99$ above the threshold. In the second episode, assume the drug is safe for the patient and that a violation of $-0.01$ occurs. The cumulative soft violation across these two episodes is $0.99 + (-0.01) = 0.98$, which seems to be lower than in the first episode. However, the hemorrhage caused in the first episode is irreversible and catastrophic. In contrast, a hard violation would have counted only the positive violations: $0.99 + 0 = 0.99$. Thus, correctly identifying that the drug was unsafe for the patient and the harm caused in an episode cannot be compensated for by good performance in subsequent episodes.

2. **Adversarial/Stochastic Loss and Constraints:** A critical aspect of online learning in CMDPs is the factor of how the losses (or rewards) and constraints are chosen in each episode – stochastically or adversarially? If the choice is made stochastically, then the losses and/or constraints are selected by sampling from an unknown and stationary probability distribution. In the adversarial case, there is no statistical assumption on the selection, and the adversary has complete freedom. Hence, it is widely acknowledged that CMDPs with adversarial losses and constraints are much more complex to solve than their stochastic counterparts. There exists a plethora of seminal works in the literature that deal with stochastic losses and constraints (Zheng & Ratliff, 2020; Efroni et al., 2020), adversarial losses and stochastic constraints (Wei et al., 2018; Qiu et al., 2020). The works of Germano et al. (2023) and Stradi et al. (2024b) were among the first ones to provide regret and violation bounds for adversarial constraints, but with a dependence on the Slater condition.

3. **Bandit/Full Feedback:** The feedback received at the end of an episode for the losses and constraints is another crucial component for online learning in CMDPs. In the *full feedback* case (Wei et al., 2018; Qiu et al., 2020), the loss and constraint costs for all the possible state-action pairs are revealed to the learner when an episode ends. While in *bandit feedback* (Müller et al., 2023; Müller et al., 2024), the loss and constraint costs for only those state-action pairs are given that the learner had visited on that specific episode. It is naturally understood that working with bandit feedback is significantly more challenging than working with full feedback. Moreover, such settings can naturally capture the whole essence of numerous real-life problems, e.g., recommender systems and budget depletion in online bidding.

Based on the above points 1, 2, and 3, we highlight some gaps that are omnipresent in the literature on online learning in CMDPs. We discuss them one-by-one: (G1) Several approaches have been employed to bound

Table 1: Comparing our theoretical results with the state-of-the-art methods. The symbol $\perp$ marks those works that consider the easier setup of stochastic losses (or rewards) and constraints. $\top$ denotes the work with adversarial losses and stochastic constraints. Zhu et al. (2025) is marked by $\ddagger$ to denote that it deals with bandit feedback for stochastic losses and full feedback for adversarial constraints. All works reported in the table address hard violations. "F/B" is a shorthand for "`Full/Bandit`".

| State-of-the-art | Transition | Feedback | Regret | Violation | With Slater |
|---|---|---|---|---|---|
| Kitamura et al. (2024) | `Known` | `F/B` | ✗/✗ | ✗/✗ | NA |
| | `Unknown` | `F/B` | $\widetilde{\mathcal{O}}(T^{6/7})^{\perp}$/✗ | $\widetilde{\mathcal{O}}(T^{6/7})^{\perp}$/✗ | ✓ |
| Müller et al. (2024) | `Known` | `F/B` | ✗/✗ | ✗/✗ | NA |
| | `Unknown` | `F/B` | ✗/$\widetilde{\mathcal{O}}(T^{0.93})^{\perp}$ | ✗/$\widetilde{\mathcal{O}}(T^{0.93})^{\perp}$ | ✓ |
| Zhu et al. (2025) | `Known` | `F/B` | ✗/✗ | ✗/✗ | NA |
| | `Unknown` | `F/B` | ✗/$\widetilde{\mathcal{O}}(\sqrt{T})^{\ddagger}$ | $\widetilde{\mathcal{O}}(\sqrt{T})^{\ddagger}$/✗ | ✗ |
| Stradi et al. (2025a) | `Known` | `F/B` | ✗/✗ | ✗/✗ | NA |
| | `Unknown` | `F/B` | ✗/$\widetilde{\mathcal{O}}(\sqrt{T})^{\perp}$ | ✗/$\widetilde{\mathcal{O}}(\sqrt{T})^{\perp}$ | ✓ |
| Stradi et al. (2025b) | `Known` | `F/B` | ✗/✗ | ✗/✗ | NA |
| | `Unknown` | `F/B` | ✗/$\widetilde{\mathcal{O}}(\sqrt{T})^{\top}$ | ✗/$\widetilde{\mathcal{O}}(\sqrt{T})^{\top}$ | ✓ |
| **This Work** | `Known` | `F/B` | $\mathcal{O}(\sqrt{T})/\mathcal{O}(\sqrt{T})$ | $\widetilde{\mathcal{O}}(\sqrt{T})/\widetilde{\mathcal{O}}(\sqrt{T})$ | ✗ |
| | `Unknown` | `F/B` | $\widetilde{\mathcal{O}}(\sqrt{T})/\widetilde{\mathcal{O}}(\sqrt{T})$ | $\widetilde{\mathcal{O}}(\sqrt{T})/\widetilde{\mathcal{O}}(\sqrt{T})$ | ✗ |

the regret and violation for online learning in CMDPs, e.g., linear programming (Efroni et al., 2020), upper confidence (Zheng & Ratliff, 2020), and primal-dual (Stradi et al., 2024a;b; 2025a; Müller et al., 2024). Primal-dual-based algorithms have arguably gained the most prominence over the years. However, these methods rely on Slater's condition, which assumes the existence of a policy satisfying all constraints with at least $\xi > 0$ slackness (Stradi et al., 2025b; Germano et al., 2023). The guarantees of such algorithms scale with $\frac{1}{\xi}$, leading to vacuous bounds (i.e., huge sub-optimal bounds), if $\xi$ is very small. Moreover, assuming Slater's condition is highly impractical because it requires prior knowledge of a strictly feasible policy or its slackness parameter, an information that is rarely available in real-world problems; (G2) A large portion of the works focus on stochastic loss and/or constraints (Efroni et al., 2020; Bai et al., 2023; Liu et al., 2021; Stradi et al., 2025a), while the ones for adversarial losses/constraints (Stradi et al., 2025a; Germano et al., 2023) are relatively less. The reason for this trend is the inherent difficulty of adversarial cases. (G3) Notably, the most challenging and non-trivial setup remains scarcely addressed in the literature: online learning in CMDPs with an unknown transition function and adversarial losses and constraints.

Sinha & Vaze (2024) obtained $\mathcal{O}(\sqrt{T})$ regret and $\widetilde{\mathcal{O}}(\sqrt{T})$ cumulative constraint violation (hard) in the domain of Constrained Online Convex Optimization (COCO) for the first time. The proposed first-order algorithm was efficient and straightforward, requiring only one projection per round. Most recently, Zhu et al. (2025) gave the `Optimistic Mirror Descent Primal-Dual (OMDPD)` algorithm, achieving the optimal $\widetilde{\mathcal{O}}(\sqrt{T})$ regret and $\widetilde{\mathcal{O}}(\sqrt{T})$ hard violation for online learning in finite-horizon episodic CMDPs. Employing some tools from Sinha & Vaze (2024) and optimizing dual variables, `OMDPD` was the first algorithm of its kind to derive optimal regret and violation bounds with adversarial constraints, without any need for Slater's condition. However, we elaborate on two critical gaps in `OMDPD` (Zhu et al., 2025): (G4) The losses were stochastic, i.e., sampled from a distribution, for all episodes; (G5) Full feedback was assumed (instead of the more realistic bandit feedback) while considering adversarial constraints.

**Our Contributions:** To the best of our knowledge, this work is the first to pose and tackle the following question for online learning in finite-horizon episodic CMDPs: (CQ) *"With no reliance on Slater's condition, with no access to a strictly feasible policy, for adversarial losses and constraints, with unknown transition function and bandit feedback, can an algorithm be designed with $\widetilde{\mathcal{O}}(\sqrt{T})$ regret and $\widetilde{\mathcal{O}}(\sqrt{T})$ hard cumulative constraint violation?".* We formally describe our contributions below:

- Although `OMDPD` borrowed elements from Sinha & Vaze (2024), they did not capitalize on the potential of using COCO to solve the setting described in (CQ). However, our work achieves this by mapping the CMDP problem to a corresponding COCO instance and employing techniques from Sinha & Vaze (2024) to provide an elegant analysis that yields optimal regret and hard violation bounds.

- Our proposed algorithms are also efficient, because only one Euclidean projection onto a simple polytope is performed per episode. Unlike primal-dual and linear-programming-based approaches, our algorithms are easy to understand. The simplicity and elegance of our framework make it a valuable didactic resource, especially for those interested in the connection between online learning in CMDPs and COCO.

- Considering adversarial losses and constraints, we solve four cases: (1) known transition function and full feedback; (2) known transition function and bandit feedback; (3) unknown transition function and full feedback; (4) unknown transition function and bandit feedback (the solution to CQ). Thus, we not only answer CQ in the resounding affirmative but also solve all possible combinations that could occur with adversarial losses and constraints with known/unknown transitions. To the best of our knowledge, an exhaustive case analysis of this nature is not present in the literature, nor does it rely on or assume Slater's condition.

- We derive optimal regret and cumulative constraint violation (hard) bounds in each case, i.e., $\mathcal{O}(\sqrt{T})$ regret and $\widetilde{\mathcal{O}}(\sqrt{T})$ violation for (1) and (2), and $\widetilde{\mathcal{O}}(\sqrt{T})$ regret and $\widetilde{\mathcal{O}}(\sqrt{T})$ violation for (3) and (4). We also construct biased estimators of the sub-gradient while solving (2) and (4), which may be of independent interest for didactic purposes. In addition to the earlier points, responding positively to (CQ) automatically resolves the gaps G1, G2, G3, G4, and G5. Table 1 compares our theoretical results with numerous state-of-the-art methods.

- Unlike Müller et al. (2023), we do not require access to a strictly feasible policy. We assume, as standard, that at least one feasible policy exists, but none of our algorithms need to know which one. This particular feasibility assumption is almost ubiquitous in the COCO literature (Yi et al., 2021; 2023).

The rest of this paper is structured as follows: In Section 2, we survey related work on online learning for MDPs, CMDPs, and constrained online optimization, highlighting both classical results and recent advances. Section 3 provides the necessary background, including the formal setup of CMDPs, occupancy measures, and COCO. Section 4 develops our algorithms and theoretical guarantees under known transition dynamics, analyzing both full and bandit feedback settings. Then, in Section 5, we extend to the more challenging regime of unknown transitions, again addressing full and bandit feedback. Section 6 presents the results of experiments we conducted on several toy CMDP instances to empirically validate the derived theoretical bounds. A brief yet insightful discussion on the optimality of our derived bounds is in Section 7. Finally, in Section 8, we state the concluding remarks.

## 2 Related Works

We categorize the prior works into three groups. First, we survey some notable works that have applied online learning to traditional MDPs over the years. Secondly, we discuss related work on online learning in CMDPs. Lastly, we briefly examined several critical works on the classical online learning problem with constraints (Cesa-Bianchi & Lugosi, 2006).

**Online Learning in MDPs:** The `UCRL2` algorithm (Jaksch et al., 2010) is one of the seminal works in this domain that proved $\widetilde{\mathcal{O}}(\sqrt{T})$ regret for undiscounted MDPs. Neu et al. (2010) showed a $\widetilde{\mathcal{O}}(T^{2/3})$ bound on the regret for undiscounted MDPs where an oblivious adversary chose the loss function. The work of Rosenberg & Mansour (2019b) used entropic regularization to establish $\widetilde{\mathcal{O}}(\sqrt{T})$ regret of episodic MDPs with unknown transitions, adversarial losses, and full feedback. An identical setting with bandit feedback has been dealt with by Rosenberg & Mansour (2019a) with $\widetilde{\mathcal{O}}(T^{3/4})$ regret. Interestingly enough, the elegant `UOB-REPS` algorithm (Jin et al., 2020) was the first to achieve $\widetilde{\mathcal{O}}(\sqrt{T})$ regret upper bound in the same problem setup

Figure 1: A brief taxonomy of online learning in CMDPs as discussed in Section 2.

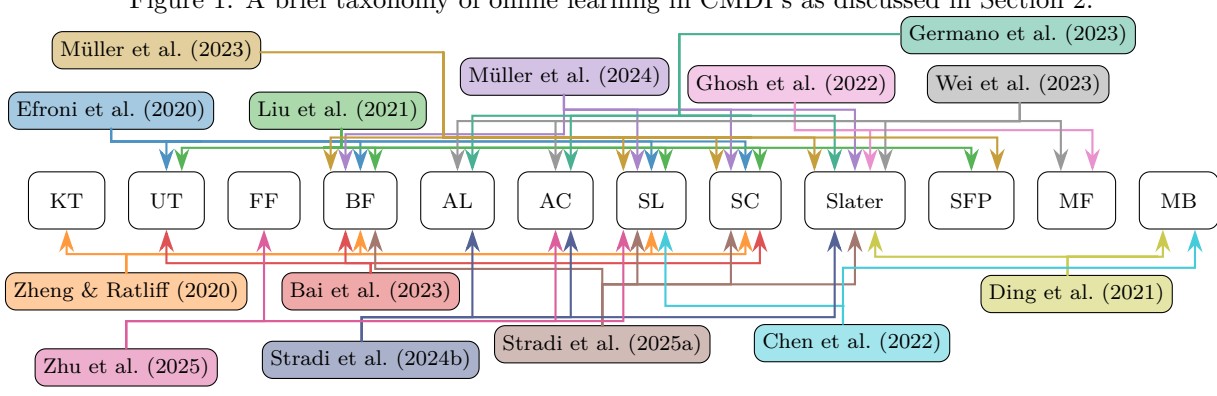

**LEGEND:** KT = Known Transition; UT = Unknown Transition; FF = Full Feedback; BF = Bandit Feedback; AL = Adversarial Loss; AC = Adversarial Constraint; SL = Stochastic Loss; SC = Stochastic Constraint; Slater = Using Slater's condition; SFP = Access to a strictly feasible policy; MF = Model Free; MB = Model Based;

as of Rosenberg & Mansour (2019a). Lee et al. (2020) obtained data-dependent high probability $\widetilde{\mathcal{O}}(\sqrt{T})$ regret bounds with an adaptive adversary and bandit feedback. It used standard unbiased estimators and a simple learning rate schedule. Furthermore, works such as Bacchiocchi et al. (2024) provided off-policy regret bounds for adversarial MDPs while Maran et al. (2024) studied online configuration of MDPs with stochastic losses, bandit feedback, and continuous decision spaces. Apart from the bandit feedback, there also exists the notion of *aggregate bandit feedback*. In such feedback, the learner observes only the total loss across the entire episode, rather than the individual losses at each state-action pair. Lancewicki & Mansour (2025) were the first to develop policy optimization algorithms for finite-horizon MDPs with adversarial losses and aggregate bandit-feedback. The cases of known and unknown transitions were handled, thereby improving earlier results. The work of Ito et al. (2025) provided the first *best-of-both-worlds* algorithm under the finite-horizon MDP setting with aggregate bandit feedback. For known transitions, the algorithms in Ito et al. (2025) attained $\mathcal{O}(\log T)$ regret with stochastic losses and $\mathcal{O}(\sqrt{T})$ regret with adversarial losses. For unknown transitions, Ito et al. (2025) employed confidence-based techniques to obtain $\widetilde{\mathcal{O}}(\sqrt{T})$ bounds.

**Online Learning in CMDPs:** Many works in this area emphasized stochastic losses and constraints. In the presence of bandit feedback, stochastic losses and constraints, and unknown transitions, Efroni et al. (2020) employed linear programming and primal-dual methods to tackle exploration-exploitation in episodic CMDPs. Sublinear regret and cumulative constraint violations were guaranteed. Zheng & Ratliff (2020) concentrated on fully-stochastic episodic CMDPs, under bandit feedback and known transitions, achieving $\widetilde{\mathcal{O}}(T^{3/4})$ regret. At the same time, its violation was shown to be below a threshold with a given probability. The seminal work of Bai et al. (2023) provided sublinear regret in the presence of peak stochastic constraints, unknown transitions, and deterministic rewards.

Focusing only on stochastic losses, numerous works (Liu et al., 2021; Müller et al., 2024; Stradi et al., 2025a) obtain sublinear bounds for hard violations of stochastic constraints. Various model-free (Ghosh et al., 2022; Wei et al., 2023) and model-based (Ding et al., 2021; Chen et al., 2022) works have also studied soft violation in CMDPs. Also, the work of Stradi et al. (2024b) gave bounds for soft constraint violations, but the losses were adversarial. With a reliance on the Slater's slackness parameter, Stradi et al. (2025a) dealt with hard constraint violation for the stochastic loss and constraints. The best-of-both-worlds regret and violation were established in Germano et al. (2023), where the loss and constraints could be both stochastic and adversarial. Although the results of Ding & Lavaei (2023); Wei et al. (2023), and Stradi et al. (2024c) do not work for adversarial losses, they establish regret and violation guarantees by considering non-stationary losses and constraints. Additionally, these works assume bounds on the variances of the loss and constraints. Very recently, the OMDPD algorithm (Zhu et al., 2025) tackled adversarial constraints and obtained $\widetilde{\mathcal{O}}(\sqrt{T})$ regret and $\widetilde{\mathcal{O}}(\sqrt{T})$ violation without Slater's condition. Given access to a strictly feasible policy and stochastic

losses and constraints, Müller et al. (2023) utilized an augmented Lagrangian approach to obtain an optimal hard violation. Figure 1 contains a schematic categorization of the works as mentioned above.

**Online Learning with Constraints:** Liakopoulos et al. (2019) examined adversarially chosen long-term budget constraints. However, their regret was defined with respect to a comparator satisfying the budget over a fixed window. Castiglioni et al. (2022a) and Castiglioni et al. (2022b) supplied the first best-of-both-worlds algorithm with long-term constraints. Hard constraint violations have also been studied in simple stochastic settings (Pacchiano et al., 2021; Bernasconi et al., 2022), in Online Convex Optimization (OCO) (Guo et al., 2022b), and in Constrained OCO (COCO) (Sinha & Vaze, 2024). Also, Sinha & Vaze (2024) first showed that it is possible to design an online policy in COCO without extra assumptions that achieves $\mathcal{O}(\sqrt{T})$ regret and $\widetilde{\mathcal{O}}(\sqrt{T})$ violation. The recent work of Lekeufack & Jordan (2024) considered a setup in which the loss predictions and the constraints are accessible. By utilizing the tools from Sinha & Vaze (2024), they (Lekeufack & Jordan, 2024) slightly improved upon the $\mathcal{O}(\sqrt{T})$ regret and $\widetilde{\mathcal{O}}(\sqrt{T})$ violation bounds.

## 3 Preliminaries

For any $n \in \mathbb{N}_{>0}$ and $z \in \mathbb{R}$, we define the notations $[n] \equiv \{1, 2, \ldots, n\}$, $[n]^{-1} \equiv \{0, 1, \ldots, n-1\}$, and $(z)^{+}$ (or $z^{+}$) $\equiv \max(0, z)$. We use the notation $\|\cdot\|$ to denote the $L^2$-norm throughout the document. Also, unless mentioned otherwise, we denote by $\nabla r$ the sub-gradient of an arbitrary convex function $r$.

### 3.1 Constrained Markov Decision Process

A finite episodic Constrained Markov Decision Process (CMDP) (Altman, 1999), is defined as the tuple $\mathcal{M} = (T, H, \mathcal{S}, \mathcal{A}, \mathcal{P}, \{\boldsymbol{\ell}_t\}_{t=1}^T, \{\boldsymbol{c}_t\}_{t=1}^T)$, where: $T$ is the total number of episodes; $H$ is the length of each episode; $\mathcal{S}$ and $\mathcal{A}$ are a finite state and action space with $|\mathcal{S}| = S$ and $|\mathcal{A}| = A$; $\mathcal{P} : \mathcal{S} \times \mathcal{A} \times \mathcal{S} \to [0, 1]$ is a transition probability function; $\{\boldsymbol{\ell}_t\}_{t=1}^T$ and $\{\boldsymbol{c}_t\}_{t=1}^T$ are the sequence of loss and constraint vectors respectively. For a fixed $t$ and for all $h \in [H]^{-1}$, the vector $\boldsymbol{\ell}_t \in [0, 1]^{S \times A \times H}$ constitutes of the loss $\ell_{t,h} : \mathcal{S} \times \mathcal{A} \to [0, 1]$, suffered by the learner for playing action $a \in \mathcal{A}$ in state $s \in \mathcal{S}$ at the $h$-th step in the $t$-th episode. Similarly, for a fixed $t$ and for all $h \in [H]^{-1}$, the components $c_{t,h} : \mathcal{S} \times \mathcal{A} \to [-1, 1]$ of the vector $\boldsymbol{c}_t \in [-1, 1]^{S \times A \times H}$, encode the cost of the constraint incurred by the learner on taking action $a \in \mathcal{A}$ in state $s \in \mathcal{S}$. Note that for each state-action pair, multiple constraints can be replaced by a single constraint, which is the point-wise maximum of the given constraints. Therefore, in this work, we assume that the learner is presented with only one constraint. The loss values $\ell_{t,h}(s, a)$ are confined to $[0, 1]$ for normalization, which is standard in the literature (Jin et al., 2020; Stradi et al., 2024a). The constraint costs $c_{t,h}(s, a)$, however, are allowed to range over $[-1, 1]$. A negative value indicates that the chosen action at state $s$ not only satisfies the constraint but does so with a margin, i.e., a "slack"[1]. In contrast, a positive value represents an actual violation. This signed representation is natural in constrained problems (Guo et al., 2022b; Yi et al., 2021; Sinha & Vaze, 2024) because it allows the learner to distinguish between safe choices (negative or zero cost) and unsafe ones (positive cost). Without loss of generality, we consider $\mathcal{M}$ to be *loop-free*, i.e., we assume that $\mathcal{S}$ is partitioned into $H + 1$ layers $\mathcal{S}_0, \ldots, \mathcal{S}_H$, such that $\mathcal{S}_0 = \{s_0\}$ and $\mathcal{S}_H = \{s_H\}$. Here, $s_0$ and $s_H$ are the initial and terminal states, respectively. For all $s \notin \mathcal{S}_H$, when playing action $a$ in state $s$, $\mathcal{P}(\cdot \mid s, a)$ is the distribution of the next state. We assume that $\mathcal{P}(s' \mid s, a) \neq 0$ only when $s \in \mathcal{S}_h$ and $s' \in \mathcal{S}_{h+1}$ for some $h < H$.

Online learning in CMDPs with adversarial losses and constraints is conducted over $T$ episodes, each consisting of $H$ steps. In each episode $t \in [T]$, the learner chooses a stochastic policy $\pi_t : \mathcal{S} \times \mathcal{A} \to [0, 1]$, where $\pi_t(a \mid s)$ is the probability of selecting the action $a \in \mathcal{A}$ in the state $s \in \mathcal{S}$. The adversary also selects the loss vector $\boldsymbol{\ell}_t$ and the constraint vector $\boldsymbol{c}_t$ at the beginning of an episode $t \in [T]$. Starting from $s_0$, the learner executes $\pi_t$ for $H$ steps and observes the trajectory $\left\{(s_h, a_h, \ell_{t,h}(s_h, a_h), c_{t,h}(s_h, a_h))\right\}_{h=0}^{H-1}$ (where the action $a_h \sim \pi_t(\cdot \mid s_h)$, and the next state $s_{h+1} \sim \mathcal{P}(\cdot \mid s_h, a_h)$) before reaching $s_H$. It is only when the $t$-th episode ends that the adversary reveals $\boldsymbol{\ell}_t$ and $\boldsymbol{c}_t$ to the learner, either in *full* or *bandit* feedback. In the full-feedback case, the loss and constraint costs for each state-action pair are disclosed to the learner. In

---

[1]For example, if the constraint limits a certain risk to be at most 0.1, a constraint cost of $-0.05$ means the incurred risk is only 0.05, leaving a slack of 0.05.

---
**Algorithm 1** Interaction between the learner and the CMDP

---
   **for** $t = 1, \ldots, T$ **do**

      The learner chooses a policy $\pi_t : \mathcal{S} \times \mathcal{A} \to [0,1]$.

      The adversary decides the loss and constraint vectors, i.e., $\boldsymbol{\ell}_t$ and $\boldsymbol{c}_t$.

      The learner starts from the fixed initial state $s_0$.

      **for** $h = 0, \ldots, H - 1$ **do**

         The learner plays the action $a_h \sim \pi_t(\cdot \mid s_h)$.

         A new state $s_{h+1} \sim \mathcal{P}(\cdot \mid s_h, a_h)$ is reached.

         The learner observes the new state $s_{h+1}$.

      **end for**

      The adversary reveals $\boldsymbol{\ell}_t$ and $\boldsymbol{c}_t$ to the learner in *full* or *bandit* feedback.

   **end for**

---

contrast, for bandit feedback, the loss and constraint costs for only the observed state-action pairs (in a trajectory) are revealed to the learner. We consider an episodic setting in which the policy remains fixed within each episode and is updated only at the end. Algorithm 1 formally describes how the learner communicates with the CMDP.

This work studies the case where both losses and constraints are adversarially chosen. It is very important to analyze adversarial settings because they naturally arise in many settings. For example, a routing agent minimizes latency (i.e., a form of loss) subject to bandwidth constraints. An adversary (e.g., network congestion) can dynamically spike latency or throttle bandwidth. Again, consider an online advertising auction in which a bidder aims to maximize clicks (i.e., minimize loss) while staying within a daily budget (i.e., a constraint). Competing bidders may adapt their bids in response to the learner's behavior, effectively making the cost per click and the remaining budget unpredictable and adversarial.

For an episode $t \in [T]$, a policy $\pi_t$, and a loss vector $\boldsymbol{\ell}_t \in [0,1]^{S \times A \times H}$, we call the *episodic loss* the expected total loss of the learner in that episode. It is defined as:

$$V^{\pi_t}(s_0; \boldsymbol{\ell}_t) := \mathbb{E}\left[\sum_{h=0}^{H-1} \ell_{t,h}(s_h, a_h) \;\middle|\; a_h \sim \pi_t(\cdot \mid s_h), s_{h+1} \sim \mathcal{P}(\cdot \mid s_h, a_h)\right], \tag{1}$$

where the learner starts from the initial state $s_0$ and follows $\pi_t$ subsequently. It is clear from the definition above that $V^{\pi_t}(s_H; \boldsymbol{\ell}_t) = 0$. The episodic loss can be generalized to start from any state $s$, with an arbitrary loss vector $\boldsymbol{\ell}$, and following $\pi$ afterwards as: $V^{\pi}(s; \boldsymbol{\ell}) := \mathbb{E}_{a \sim \pi(\cdot \mid s)}\left[Q^{\pi}(s, a; \boldsymbol{\ell})\right]$, where $Q^{\pi}(s, a; \boldsymbol{\ell}) := \ell(s,a) + \mathbb{1}_{s \notin \mathcal{S}_H}\mathbb{E}_{s' \sim \mathcal{P}(\cdot \mid s,a)}\left[V^{\pi}(s'; \boldsymbol{\ell})\right]$ (where $\ell(s,a)$ is a component of the vector $\boldsymbol{\ell}$) is the Bellman equation denoting the expected loss starting from $s$, taking action $a$, and following $\pi$ afterward. Similar to the episodic loss $V^{\pi_t}(s_0; \boldsymbol{\ell}_t)$, we define $V^{\pi_t}(s_0; \boldsymbol{c}_t)$ for computing the expected violation of the constraints in an episode as:

$$V^{\pi_t}(s_0; \boldsymbol{c}_t) := \mathbb{E}\left[\sum_{h=0}^{H-1} c_{t,h}(s_h, a_h) \;\middle|\; a_h \sim \pi_t(\cdot \mid s_h), s_{h+1} \sim \mathcal{P}(\cdot \mid s_h, a_h)\right]. \tag{2}$$

We term $V^{\pi_t}(s_0; \boldsymbol{c}_t)$ as the *episodic constraint violation* which can also be generalized to start from any state $s$, with an arbitrary constraint vector $\boldsymbol{c}$, and following $\pi$ afterwards as: $V^{\pi}(s; \boldsymbol{c}) := \mathbb{E}_{a \sim \pi(\cdot \mid s)}\left[Q^{\pi}(s, a; \boldsymbol{c})\right]$, where the Bellman equation $Q^{\pi}(s, a; \boldsymbol{c}) := c(s,a) + \mathbb{1}_{s \notin \mathcal{S}_H}\mathbb{E}_{s' \sim \mathcal{P}(\cdot \mid s,a)}\left[V^{\pi}(s'; \boldsymbol{c})\right]$ (where $c(s,a)$ is a component of the vector $\boldsymbol{c}$) denotes the expected constraint violations starting from $s$, taking action $a$, and following $\pi$ afterward. For a known transition function $\mathcal{P}$, the expectations in Eqn. 1 and Eqn. 2 will only be taken on the randomness in sampling the actions. One could simply write $V^{\pi_t}(\boldsymbol{\ell}_t)$ and $V^{\pi_t}(\boldsymbol{c}_t)$ when the starting state is clear from the context.

Let us assume $\pi^\star \in \arg\min_{\pi \in \Pi} \sum_{t=1}^{T} V^{\pi_t}(s_0; \boldsymbol{\ell}_t)$ to be an optimal policy in hindsight that satisfies the constraints over the episodes, i.e., $\sum_{t=1}^{T}(V^{\pi^\star}(s_0; \boldsymbol{c}_t))^+ = 0$. We denote by $\Pi$ the class of all stochastic policies. The final objective of the learner is to learn a policy that jointly minimizes the expected regret and

the expected cumulative constraint violation over all the episodes:

$$\mathbb{E}\big[\mathcal{R}_T\big] := \mathbb{E}\left[\sum_{t=1}^{T} V^{\pi_t}(s_0; \boldsymbol{\ell}_t)\right] - \sum_{t=1}^{T} V^{\pi^\star}(s_0; \boldsymbol{\ell}_t), \text{ and} \tag{3}$$

$$\mathbb{E}\big[\mathcal{Z}_T\big] := \mathbb{E}\left[\sum_{t=1}^{T} \max\big(0, V^{\pi_t}(s_0; \boldsymbol{c}_t)\big)\right] = \mathbb{E}\left[\sum_{t=1}^{T} \big(V^{\pi_t}(s_0; \boldsymbol{c}_t)\big)^+\right]. \tag{4}$$

In the bandit feedback setting, the expectations in the above equations are taken with respect to the randomness in the choice of $\pi_t$ at the beginning of each episode. In the full feedback case, there is no stochasticity in the policy, so expectations do not appear in Eqn. 3 and Eqn. 4.

### 3.2 Occupancy Measures

It is well known that any policy $\pi$ and a transition probability function $\mathcal{P}$ induce an *occupancy measure* $\rho^{\mathcal{P},\pi} : \mathcal{S} \times \mathcal{A} \to [0,1]$ (Altman, 1999; Rosenberg & Mansour, 2019b), where $\rho^{\mathcal{P},\pi}(s,a)$ is the probability of visiting the state-action pair $(s,a)$ when the learner starts from the initial state and acts according to $\pi$. Consider the following definition, which formalizes the notion of occupancy measures.

**Definition 1** (Occupancy Measure). *For every $s \in \mathcal{S}$ and $a \in \mathcal{A}$ the occupancy measure $\rho^{\mathcal{P},\pi} : \mathcal{S} \times \mathcal{A} \to [0,1]$ induced by a policy $\pi$ and a transition function $\mathcal{P}$ is the probability of visiting the pair $(s,a)$ when the agent begins from $s_0$ and then follows $\pi$ in an episode. Therefore, the probability of visiting a state $s \in \mathcal{S}$ in an episode will be:*

$$\rho^{\mathcal{P},\pi}(s) = \sum_{a \in \mathcal{A}} \rho^{\mathcal{P},\pi}(s,a). \tag{5}$$

From now on, we omit writing $\mathcal{P}$ in $\rho^{\mathcal{P},\pi}$ for simplicity (unless absolutely required). Let $\Omega = \{\rho^\pi \mid \pi \in \Pi\}$ be the set of all *valid occupancy measures*. From the work of Luo et al. (2021), we have an alternative characterization for $\Omega$ that is widely used in the literature, and it is elucidated in the following definition.

**Definition 2** (Valid Occupancy Measures). *We have the following equivalent definition of $\Omega$:*

$$\Omega = \left\{ \rho \in [0,1]^{S \times A \times H} \,\middle|\, \rho(s_0) = 1; \rho(s') = \sum_{s \in \mathcal{S}_h} \sum_{a \in \mathcal{A}} \rho(s,a)\mathcal{P}(s' \mid s,a), \forall s' \in \mathcal{S}_{h+1} \text{ and } \forall h \in [H]^{-1} \right\}. \tag{6}$$

Any $\rho \in \Omega$ corresponds to the occupancy measure induced by the policy $\pi^\rho$ with $\pi^\rho(a \mid s) = \frac{\rho(s,a)}{\rho(s)}$, i.e., $\pi^\rho(a \mid s) \propto \rho(s,a)$. It is evident from Eqn. 1 and Eqn. 2 that $V^{\pi_t}(s_0; \boldsymbol{\ell}_t)$ and $V^{\pi_t}(s_0; \boldsymbol{c}_t)$ are non-convex in $\pi_t$. It is important to note that $\rho^{\pi_t}$, $\boldsymbol{\ell}_t$, and $\boldsymbol{c}_t$ are vectors of dimension $S \times A \times H$. Thus, being equipped with Definition 1, the episodic loss $V^{\pi_t}(s_0; \boldsymbol{\ell}_t)$ and the episodic constraint violation $V^{\pi_t}(s_0; \boldsymbol{c}_t)$ can be re-written as $\langle \rho^{\pi_t}, \boldsymbol{\ell}_t \rangle$ and $\langle \rho^{\pi_t}, \boldsymbol{c}_t \rangle$ respectively, thereby, making $V^{\pi_t}(s_0; \boldsymbol{\ell}_t)$ and $V^{\pi_t}(s_0; \boldsymbol{c}_t)$ linear in the occupancy measure $\rho^{\pi_t}$. Consequently, the expected regret in Eqn. 3 and the expected cumulative constraint violation in Eqn. 4 can be equivalently expressed as:

$$\mathbb{E}\big[\mathcal{R}_T\big] := \mathbb{E}\left[\sum_{t=1}^{T} \langle \rho^{\pi_t} - \rho^{\pi^\star}, \boldsymbol{\ell}_t \rangle\right], \text{ and} \tag{7}$$

$$\mathbb{E}\big[\mathcal{Z}_T\big] := \mathbb{E}\left[\sum_{t=1}^{T} \max\big(0, \langle \rho^{\pi_t}, \boldsymbol{c}_t \rangle\big)\right] = \mathbb{E}\left[\sum_{t=1}^{T} \langle \rho^{\pi_t}, \boldsymbol{c}_t \rangle^+\right]. \tag{8}$$

As before, the expectations in Eqn. 7 and Eqn. 8 will not be present in the full feedback case. From now on, we will employ the shorthand $\rho_t$ and $\rho^\star$ instead of $\rho^{\pi_t}$ and $\rho^{\pi^\star}$ respectively. Also, note that Eqn. 8 and Eqn. 4 naturally encapsulate the notion of *hard constraint violation*. It is worth noting that we focus on achieving sublinear hard constraint violation and not on providing high-probability per-trajectory safety guarantees.

### 3.3 Constrained Online Convex Optimization

Online Convex Optimization (OCO) (Hazan, 2016; Orabona, 2025) provides a valuable arsenal for tackling online decision-making problems. The framework of Constrained Online Convex Optimization (COCO) (Guo et al., 2022a; Sinha & Vaze, 2024) generalizes OCO by modeling a round-based game between an online policy and an adversary. At each round $t \in [T]$, the online policy selects an action $x_t \in \mathcal{X}$, where $\mathcal{X}$ is called the *admissible set*. Then, a convex cost function $\mu_t : \mathcal{X} \to \mathbb{R}$ and a convex constraint function $\nu_t : \mathcal{X} \to \mathbb{R}$ are chosen by the adversary. To be specific, on playing the action $x_t$, the online policy suffers a cost $\mu_t(x_t)$ and a constraint violation $\nu_t(x_t)$.

Let $\mathcal{X}^\star$ be the set of all admissible actions satisfying the constraint on every round, i.e., $\mathcal{X}^\star = \{x \in \mathcal{X} \mid \nu_t(x) \leq 0, \forall t \geq 1\}$. The set $\mathcal{X}^\star$ is called the *feasible set* in the standard COCO literature. The end goal of any COCO problem is to build an online policy that jointly minimizes regret and cumulative constraint violation, which are defined as:

$$\text{Regret}_T := \sum_{t=1}^{T} \mu_t(x_t) - \inf_{x^\star \in \mathcal{X}^\star} \sum_{t=1}^{T} \mu_t(x^\star), \text{ and} \tag{9}$$

$$\text{CCV}_T := \sum_{t=1}^{T} \max\big(0, \nu_t(x_t)\big) = \sum_{t=1}^{T} \nu_t(x_t)^+. \tag{10}$$

We state three standard assumptions prevalent in the COCO literature (Yi et al., 2021; Guo et al., 2022a; Yi et al., 2023). The first one, i.e., Assumption 1, is on the convexity of the admissible set $\mathcal{X}$, while Assumption 2 describes the Lipschitz continuity of $\{\mu_t\}_{t=1}^{T}$ and $\{\nu_t\}_{t=1}^{T}$. The direct implication of this assumption is that the $L^2$-norm of $\{\nabla \mu_t\}_{t=1}^{T}$ and $\{\nabla \nu_t\}_{t=1}^{T}$ is uniformly upper bounded by the Lipschitz constant. Assumption 3 states that the feasible set $\mathcal{X}^\star$ is non-empty.

**Assumption 1** (Convexity). *The admissible set $\mathcal{X} \subseteq \mathbb{R}^d$ is closed and convex and has a finite Euclidean diameter of $D$. For all $t \in [T]$, the cost functions $\{\mu_t\}_{t=1}^{T}$ and the constraint functions $\{\nu_t\}_{t=1}^{T}$ are convex.*

**Assumption 2** (Lipschitzness). *All the costs $\{\mu_t\}_{t=1}^{T}$ and constraints $\{\nu_t\}_{t=1}^{T}$ are $L$-Lipschitz. Thus, for all $a, b \in \mathcal{X}$ and for every $t \in [T]$, we have:*

$$|\mu_t(a) - \mu_t(b)| \leq L \cdot \|a - b\|, \ |\nu_t(a) - \nu_t(b)| \leq L \cdot \|a - b\|. \tag{11}$$

**Assumption 3** (Feasibility). *The feasible set is non-empty, i.e., $\mathcal{X}^\star \neq \emptyset$, as there always exists an $x^\star \in \mathcal{X}$ for which $\nu_t(x^\star) \leq 0$, for all $t \in [T]$.*

It is essential to recognize that the objective in COCO and in online learning in CMDPs is the same: minimizing regret and cumulative constraint violation. This fact enables the solution of CMDPs using COCO algorithms after appropriate reductions. Inspired by Sinha & Vaze (2024), we utilize a Lyapunov potential function to regulate the growth of violations and construct a surrogate loss by linearly combining an upper bound on the change of the Lyapunov function with the cost function.

### 3.4 Reduction from CMDP to COCO - a simple toy example

We provide a toy example to illustrate the reduction that is central in the upcoming sections. Let us consider a CMDP $\mathcal{G} = (T, H, \mathcal{S}, \mathcal{A}, \mathcal{P}, \{\boldsymbol{\ell}_t\}_{t=1}^{T}, \{\boldsymbol{c}_t\}_{t=1}^{T})$ with $|\mathcal{S}| = S$, $|\mathcal{A}| = A$, and with horizon length of two, i.e., let $H = 2$. Assume that the transition function $\mathcal{P}$ is known. Since $\mathcal{G}$ is loop-free, the finite state space $\mathcal{S}$ can be written as: $\mathcal{S} = \bigcup_{h=0}^{2} \mathcal{S}_h = \mathcal{S}_0 \bigcup \mathcal{S}_1 \bigcup \mathcal{S}_2$ and $\mathcal{S}_k \bigcap \mathcal{S}_l = \emptyset$ for $k \neq l$. By the definition given in Section 3.1, the first and last layer only contain the fixed initial and terminal state respectively, i.e., $\mathcal{S}_0 = \{s_0\}$ and $\mathcal{S}_2 = \{s_2\}$. Let the intermediate state layer be $\mathcal{S}_1 = \{x, y\}$ and the finite action space be $\mathcal{A} = \{0, 1\}$. In this case, the occupancy vector is:

$$\rho = [\rho_0(s_0, 0), \rho_0(s_0, 1), \rho_1(x, 0), \rho_1(x, 1), \rho_1(y, 0), \rho_1(y, 1)]. \tag{12}$$

Moreover, the valid set $\Omega$ will contain any $\rho \in [0, 1]^{S \times A \times H}$ satisfying the following constraints:

Figure 2: Schematic to illustrate the CMDP-COCO reduction. Each CMDP episode $t$ corresponds to one COCO round. The occupancy measure $\rho_t$ maps to the decision variable $x_t$ in the admissible set $\mathcal{X}$, which equals $\Omega$. The vectors $\boldsymbol{\ell}_t$ and $\boldsymbol{c}_t$ help to define the linear cost function $\mu_t$ and constraint function $\nu_t$ in COCO.

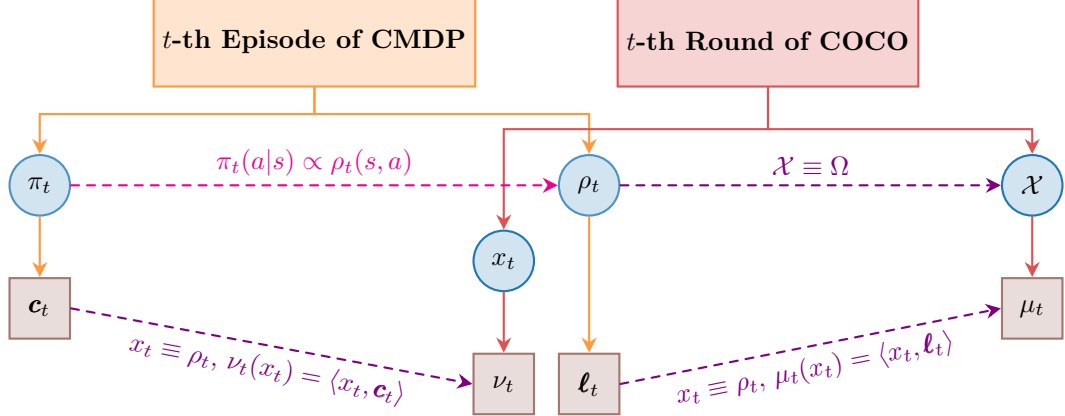

**LEGEND:** Solid arrows := CMDP/COCO internal relations; Dashed arrows := Reduction mapping
$\Omega$ := Valid Occupancy Measures; $\mathcal{X}$ := COCO Admissible Set

1. For $h = 0$: $\rho_0(s_0, 0) + \rho_0(s_0, 1) = 1$.

2. For $h = 1$: $\forall s' \in \{x, y\}$, $\rho_1(s', 0) + \rho_1(s', 1) = \sum_{a \in \mathcal{A}} \rho_0(s_0, a)\mathcal{P}(s' \mid s_0, a)$.

Any $\rho$ satisfying the above constraints is realizable by the policy: $\pi_\rho(a \mid s) = \frac{\rho_h(s,a)}{\sum_{a'} \rho_h(s,a')}$, whenever $\sum_{a'} \rho_h(s, a') > 0$. For episode $t \in [T]$, with losses $\ell_{t,h}(s, a)$ and constraints $c_{t,h}(s, a)$, we have the following definitions for the cost function $\mu_t$ and the constraint function $\nu_t$:

$$\mu_t(\rho) = \sum_{(s,a,h)} \rho_h(s,a) \cdot \ell_{t,h}(s,a), \text{ and} \tag{13}$$

$$\nu_t(\rho) = \sum_{(s,a,h)} \rho_h(s,a) \cdot c_{t,h}(s,a), \tag{14}$$

which are linear (and hence convex) in $\rho$. Thus, one CMDP episode is equivalent to one round in the COCO problem with the decision $\rho_t \in \Omega$. Figure 2 depicts the general mapping of the CMDP's elements to their counterparts in a COCO round. The left side shows a CMDP episode with policy $\pi_t$, occupancy measure $\rho_t$, loss vector $\boldsymbol{\ell}_t$, and constraint vector $\boldsymbol{c}_t$. The right side shows a COCO round with decision variable $x_t$, admissible set $\mathcal{X}$, cost function $\mu_t$, and constraint function $\nu_t$. The set of valid occupancy measures $\Omega$ in CMDP exactly corresponds to the COCO admissible set $\mathcal{X}$, while the loss and constraint functions are linearly defined by $\boldsymbol{\ell}_t$ and $\boldsymbol{c}_t$. The solid arrows indicate internal relationships within each framework, while the dashed arrows indicate the mapping. The violet-dashed arrows show that the CMDP's linearity in $\rho_t$ directly corresponds to COCO's $\mu_t$ and $\nu_t$.

## 4  Known Transition Function

When the transition function $\mathcal{P}$ is known for the CMDP $\mathcal{M}$, there is no model uncertainty regarding $\mathcal{P}$, but there will be randomness linked with the next-state $s_{h+1}$ in an episode $t \in [T]$. Throughout this section, we will use Eqn. 15 and Eqn. 16 as the definition of the episodic loss and episodic constraint violation,

respectively, as written below:

$$V^{\pi_t}(s_0; \boldsymbol{\ell}_t) := \mathbb{E}\left[\sum_{h=0}^{H-1} \ell_{t,h}(s_h, a_h) \,\middle|\, a_h \sim \pi_t(\cdot \mid s_h), s_{h+1} \sim \mathcal{P}(\cdot \mid s_h, a_h)\right], \text{ and} \tag{15}$$

$$V^{\pi_t}(s_0; \boldsymbol{c}_t) := \mathbb{E}\left[\sum_{h=0}^{H-1} c_{t,h}(s_h, a_h) \,\middle|\, a_h \sim \pi_t(\cdot \mid s_h), s_{h+1} \sim \mathcal{P}(\cdot \mid s_h, a_h)\right]. \tag{16}$$

### 4.1 Full Feedback and Known Transition

In addition to the transition function being known, the entire loss vector $\boldsymbol{\ell}_t$ and the constraint vector $\boldsymbol{c}_t$ are revealed to the learner at the end of an episode. Consequently, the regret $\mathcal{R}_T$ and the cumulative constraint violation $\mathcal{Z}_T$ to be minimized in this scenario are given as:

$$\mathcal{R}_T := \sum_{t=1}^{T} \langle \rho_t - \rho^\star, \boldsymbol{\ell}_t \rangle, \text{ and} \tag{17}$$

$$\mathcal{Z}_T := \sum_{t=1}^{T} \langle \rho_t, \boldsymbol{c}_t \rangle^+. \tag{18}$$

Owing to the above definitions, our optimization problem is to find an occupancy measure in the space of all valid occupancy measures, i.e., $\rho_t \in \Omega$ for all $t \in [T]$. We will jointly minimize Eqn. 17 and Eqn. 18 by mapping our problem to a corresponding instance of the COCO problem. As already described in Section 3.3, COCO proceeds as a game of $T$ rounds between an online policy and an adversary. Clearly, one COCO round corresponds to one episode of length $H$ in the CMDP. For every $t \in [T]$, we define the cost function $\mu_t : \Omega \to \mathbb{R}$ and constraint function $\nu_t : \Omega \to \mathbb{R}$ as:

$$\mu_t(\rho_t) = \sum_{h=0}^{H-1} \rho_t(s_h, a_h) \cdot \ell_{t,h}(s_h, a_h) = \langle \rho_t, \boldsymbol{\ell}_t \rangle, \text{ and} \tag{19}$$

$$\nu_t(\rho_t) = \sum_{h=0}^{H-1} \rho_t(s_h, a_h) \cdot c_{t,h}(s_h, a_h) = \langle \rho_t, \boldsymbol{c}_t \rangle. \tag{20}$$

It is clear from Eqn. 19 and Eqn. 20 that $\mu_t$ and $\nu_t$ are linear in $\rho_t$ (thus, convex). Hence, $\mu_t$ and $\nu_t$ are indeed Lipschitz continuous with respect to $\rho_t$. The gradients of $\mu_t(\rho_t)$ and $\nu_t(\rho_t)$ are: $\nabla\mu_t(\rho_t) = \boldsymbol{\ell}_t$ and $\nabla\nu_t(\rho_t) = \boldsymbol{c}_t$. It is easy to see that the maximum $L^2$-norm of $\boldsymbol{\ell}_t$ and $\boldsymbol{c}_t$ are $\|\boldsymbol{\ell}_t\| = \|\boldsymbol{c}_t\| \leq \sqrt{SHA}$. Therefore, the upper bound on the value of the Lipschitz constant $L$ for Eqn. 19 and Eqn. 20 directly follows from the gradient norms, i.e., $L \leq \sqrt{SHA}$.

Definition 2 necessitates that $\Omega$ should be a simple polytope with $\mathcal{O}(S)$-many linear constraints, implying $\Omega$ is closed and convex. Since $\Omega \subset [0,1]^{S \times A \times H}$, the largest possible Euclidean distance between any two points $\rho_1^\pi, \rho_2^\pi \in \Omega$ is the diagonal distance of the hypercube $[0,1]^{S \times A \times H}$, which is simply equal to $\sqrt{S \times A \times H}$. Therefore, we have the Euclidean diameter of $\Omega$ as: $D := \sup_{\rho_1^\pi, \rho_2^\pi \in \Omega} \|\rho_1^\pi - \rho_2^\pi\| = \sqrt{S \times A \times H} = \sqrt{SHA}$. At this juncture, we can now define the regret and the cumulative constraint violation of the corresponding COCO problem as follows:

$$\text{Regret}_T := \sum_{t=1}^{T} \mu_t(\rho_t) - \sum_{t=1}^{T} \mu_t(\rho^\star), \text{ and} \tag{21}$$

$$\text{CCV}_T := \sum_{t=1}^{T} \nu_t(\rho_t)^+. \tag{22}$$

In each episode $t \in [T]$, we perform the scaling: $\widetilde{\mu}_t \leftarrow \omega\mu_t$, $\widetilde{\nu}_t \leftarrow \omega(\nu_t)^+$, where $\omega > 0$. The scaled cost function $\widetilde{\mu}_t$ and the scaled constraint function $\widetilde{\nu}_t$ are both $\omega L$-Lipschitz for all $t \geq 1$. Let $\varphi : \mathbb{R}^+ \to \mathbb{R}^+$ be

---

**Algorithm 2** `Full AdaGrad with Known Transition (FAG-K)`

---

**Require:** $L$, $D$, Euclidean projection operator $\Pi_\Omega(\cdot)$ on $\Omega$.

   Set the parameters $\omega = \frac{1}{2LD}$, $\theta = \frac{1}{2\sqrt{T}}$, and choose $\varphi(\zeta_t) = \exp(\theta\zeta_t) - 1$, $\forall t \geq 1$.

   Intialize $\rho_1 \in \Omega$ arbitrarily (e.g., uniformly) and set $\zeta_0 = 0$.

   **for** $t = 1, \ldots, T$ **do**

      Extract the policy $\pi_t$ such that $\pi_t(a \mid s) \propto \rho_t(s, a)$, $\forall(s, a) \in \mathcal{S} \times \mathcal{A}$.

      The adversary decides $\boldsymbol{\ell}_t$ and $\boldsymbol{c}_t$.

      **for** $h = 0, \ldots, H - 1$ **do**

         The learner plays $a_h \sim \pi_t(\cdot \mid s_h)$.

         The learner reaches new state $s_{h+1} \sim \mathcal{P}(\cdot \mid s_h, a_h)$ and observes $s_{h+1}$.

      **end for**

      The adversary reveals $\boldsymbol{\ell}_t$ and $\boldsymbol{c}_t$ in *full* feedback.

      Define $\mu_t(\rho_t) = \langle \rho_t, \boldsymbol{\ell}_t \rangle$, and $\nu_t(\rho_t) = \langle \rho_t, \boldsymbol{c}_t \rangle$.

      Compute $\widetilde{\mu}_t \leftarrow \omega\mu_t$, and $\widetilde{\nu}_t \leftarrow \omega(\nu_t)^+$.

      Compute $\zeta_t = \zeta_{t-1} + \widetilde{\nu}_t(\rho_t)$ and $\widehat{\mu}_t(\rho_t) := \widetilde{\mu}_t(\rho_t) + \varphi'(\zeta_t)\widetilde{\nu}_t(\rho_t)$.

      According to Eqn. 25, compute the sub-gradient $\nabla_t = \nabla\widehat{\mu}_t(\rho_t)$.

      Update $\rho_{t+1} = \Pi_\Omega(\rho_t - \eta_t\nabla_t)$, where $\eta_t = \dfrac{\sqrt{2}D}{2\sqrt{\sum_{\tau=1}^{t}\|\nabla_\tau\|^2}}$.

   **end for**

   **return** $\rho_T$ and $\pi_T$.

---

any non-decreasing, differentiable, and convex Lyapunov function such that $\varphi(0) = 0$. Also, let $\zeta_t$ be the cumulative constraint violation for the scaled constraint function till the $t$-th episode, where $\zeta_t = \zeta_{t-1} + \widetilde{\nu}_t(\rho_t)$, $t \geq 1$ (with $\zeta_0 = 0$). It follows from the convexity of $\varphi(\cdot)$:

$$\varphi(\zeta_{t-1}) \geq \varphi(\zeta_t) + \varphi'(\zeta_t)(\zeta_{t-1} - \zeta_t) \implies \varphi(\zeta_t) \leq \varphi(\zeta_{t-1}) + \varphi'(\zeta_t)(\zeta_t - \zeta_{t-1})$$
$$\implies \varphi(\zeta_t) - \varphi(\zeta_{t-1}) \leq \varphi'(\zeta_t)\widetilde{\nu}_t(\rho_t). \tag{23}$$

It is important to note that the scaling factor $\omega = \frac{1}{2LD}$ is introduced to normalize the Lipschitz constants and the diameter of the decision set $\Omega$. Specifically, $L$ is the Lipschitz constant of $\mu_t$ and $\nu_t$, and $D$ is the Euclidean diameter of $\Omega$. Scaling by $\omega$ ensures that the gradients of $\widetilde{\mu}_t$ and $\widetilde{\nu}_t$ have norm at most $\omega L \leq \frac{1}{2D}$, which simplifies the regret analysis. From the stochastic drift-plus-penalty framework of Neely (2010), we define the surrogate loss as (taking the penalty to be 1):

$$\widehat{\mu}_t(\rho_t) := \widetilde{\mu}_t(\rho_t) + \varphi'(\zeta_t)\widetilde{\nu}_t(\rho_t), \ \forall t \geq 1. \tag{24}$$

The surrogate loss combines the scaled cost $\widetilde{\mu}_t$ with a penalty term $\varphi'(\zeta_t)\widetilde{\nu}_t$. The term $\varphi'(\zeta_t)$ acts as an adaptive weight on the constraint violation: if cumulative violations $\zeta_t$ are large, $\varphi'(\zeta_t)$ increases, thereby penalizing violations more heavily in the surrogate loss. This mechanism helps control the growth of constraint violations over time. By minimizing the surrogate loss $\widehat{\mu}_t$, the algorithm implicitly balances cost minimization and constraint satisfaction, yielding simultaneous sublinear regret and sublinear hard constraint violation.

The subgradient of $\widehat{\mu}_t$ is computed as follows:

$$\nabla_t = \nabla\widehat{\mu}_t(\rho_t) = \nabla\widetilde{\mu}_t(\rho_t) + \nabla\varphi'(\zeta_t)\widetilde{\nu}_t(\rho_t) = \nabla\langle\rho_t, \omega\boldsymbol{\ell}_t\rangle + \varphi'(\zeta_t)\nabla\langle\rho_t, \omega\boldsymbol{c}_t\rangle^+$$
$$\implies \nabla_t = \begin{cases} \omega\boldsymbol{\ell}_t + \varphi'(\zeta_t)\omega\boldsymbol{c}_t, & \text{if } \langle\rho_t, \omega\boldsymbol{c}_t\rangle > 0, \\ \omega\boldsymbol{\ell}_t, & \text{if } \langle\rho_t, \omega\boldsymbol{c}_t\rangle \leq 0. \end{cases} \tag{25}$$

We can upper bound $\|\nabla_t\|$ as:

$$\|\nabla_t\| = \|\nabla\widehat{\mu}_t(\rho_t)\| = \|\nabla\widetilde{\mu}_t(\rho_t)\| + \varphi'(\zeta_t)\|\nabla\widetilde{\nu}_t(\rho_t)\| \leq \omega L(1 + \varphi'(\zeta_t)). \tag{26}$$

---

**Algorithm 3** `Online AdaGrad policy with adaptive step-sizes`

---

**Require:** A closed convex set $\mathcal{Y}$ with Euclidean diameter $D$, positive step sizes $\{\eta_t\}_{t=1}^T$, convex cost functions
  $\{\mu_t\}_{t=1}^T$, projection operator $\mathcal{P}_{\mathcal{Y}}(\cdot)$.
  Set $y_1 \in \mathcal{Y}$ arbitrarily.
  **for** $t = 1, \ldots, T$ **do**
    Execute $y_t$ and observe $\mu_t$.
    Suffer a cost of $\mu_t(y_t)$.
    Compute sub-gradient $\nabla_t \equiv \nabla\mu_t(y_t)$.
    Update $y_{t+1} = \mathcal{P}_{\mathcal{Y}}(y_t - \eta_t\nabla_t)$.
  **end for**

---

By the feasibility condition, we have $\nu_\tau(\rho^\star) \leq 0$ (for all $\tau \geq 1$), which implies that $\widetilde{\nu}_\tau(\rho^\star) = 0$. Consequently, the following observation is easily made:

$$\widehat{\mu}_\tau(\rho^\star) = \widetilde{\mu}_\tau(\rho^\star) + \varphi'(\zeta_\tau)\widetilde{\nu}_\tau(\rho^\star)$$
$$\implies \widehat{\mu}_\tau(\rho^\star) = \widetilde{\mu}_\tau(\rho^\star), \, \forall \tau \geq 1. \tag{27}$$

For any $\tau \geq 1$, using Eqn. 27 and Eqn. 24 in Eqn. 23, we have:

$$\varphi(\zeta_\tau) - \varphi(\zeta_{\tau-1}) \leq \varphi'(\zeta_\tau)\widetilde{\nu}_\tau(\rho_\tau)$$
$$\implies \varphi(\zeta_\tau) - \varphi(\zeta_{\tau-1}) \leq \varphi'(\zeta_\tau)\frac{\widehat{\mu}_\tau(\rho_\tau) - \widetilde{\mu}_\tau(\rho_\tau)}{\varphi'(\zeta_\tau)}$$
$$\implies \varphi(\zeta_\tau) - \varphi(\zeta_{\tau-1}) \leq \widehat{\mu}_\tau(\rho_\tau) - \widetilde{\mu}_\tau(\rho_\tau)$$
$$\implies \varphi(\zeta_\tau) - \varphi(\zeta_{\tau-1}) - \widehat{\mu}_\tau(\rho^\star) \leq \widehat{\mu}_\tau(\rho_\tau) - \widetilde{\mu}_\tau(\rho_\tau) - \widehat{\mu}_\tau(\rho^\star)$$
$$\implies \varphi(\zeta_\tau) - \varphi(\zeta_{\tau-1}) + \widetilde{\mu}_\tau(\rho_\tau) - \widehat{\mu}_\tau(\rho^\star) \leq \widehat{\mu}_\tau(\rho_\tau) - \widehat{\mu}_\tau(\rho^\star).$$

Summing the above inequality for $1 \leq \tau \leq t$ and using $\varphi(0) = 0$, we get:

$$\sum_{\tau=1}^t \varphi(\zeta_\tau) - \varphi(\zeta_{\tau-1}) + \sum_{\tau=1}^t \widetilde{\mu}_\tau(\rho_\tau) - \widetilde{\mu}_\tau(\rho^\star) \leq \sum_{\tau=1}^t \widehat{\mu}_\tau(\rho_\tau) - \widehat{\mu}_\tau(\rho^\star)$$
$$\implies \varphi(\zeta_t) + \text{Regret}_t(\rho^\star) \leq \text{Regret}'_t(\rho^\star), \tag{28}$$

where $\text{Regret}_t$ on the LHS and $\text{Regret}'_t$ on the RHS of Eqn. 28 refer to the regret for learning the pre-processed cost functions $\{\widetilde{\mu}_t\}_{t\geq 1}$ and the surrogate loss functions $\{\widehat{\mu}_t\}_{t\geq 1}$ respectively.

We utilize the online `AdaGrad` policy (Zinkevich, 2003) with adaptive step sizes (Duchi et al., 2011) as a sub-routine, described in Algorithm 3, to minimize the surrogate regret $\text{Regret}'_t(\rho^\star)$. Let us recall an important theorem below (given as Theorem 1) from Orabona (2025) and Duchi et al. (2011) that gives the adaptive regret bound attained by the online AdaGrad policy.

**Theorem 1.** *Given a sequence of convex cost functions $\{\mu_t\}_{t=1}^T$, the adaptive step size schedule for all $t \geq 1$:*
$\eta_t = \frac{\sqrt{2}D}{2\sqrt{\sum_{\tau=1}^t \|\nabla_\tau\|^2}}$ *(D is the diameter of $\mathcal{Y}$), and $\|\nabla_t\|$. Hence, the regret of Algorithm 3 is given by:*

$$\text{Regret}_T \leq \sqrt{2}D\sqrt{\sum_{t=1}^T \|\nabla_t\|^2}. \tag{29}$$

We name our algorithm in this scenario as **F**ull **A**da**G**rad with **K**nown Transition (`FAG-K`), and it is formally presented in Algorithm 2. Using Eqn. 29 from Theorem 1, we can upper bound the surrogate regret as (see Appendix A.1 for the detailed calculation):

$$\text{Regret}'_t(\rho^\star) \leq 2D\omega L\sqrt{t}\left(1 + \varphi'(\zeta_t)\right). \tag{30}$$

Putting $\omega = \frac{1}{2LD}$, choosing $\varphi(\zeta_t) = \exp(\theta\zeta_t) - 1$, $\forall t \geq 1$, and substituting Eqn. 30 into the regret decomposition inequality of Eqn. 28, we have:

$$\varphi(\zeta_t) + \text{Regret}_t(\rho^\star) \leq \text{Regret}'_t(\rho^\star)$$
$$\implies \exp(\theta\zeta_t) - 1 + \text{Regret}_t(\rho^\star) \leq 2D\omega L\sqrt{t}\left(1 + \theta\exp(\theta\zeta_t)\right)$$
$$\implies \text{Regret}_t(\rho^\star) \leq 2D\omega L\sqrt{t}\left(1 + \theta\exp(\theta\zeta_t)\right) + 1 - \exp(\theta\zeta_t)$$
$$\implies \text{Regret}_t(\rho^\star) \leq \sqrt{t} + \theta\sqrt{t}\exp(\theta\zeta_t) + 1 - \exp(\theta\zeta_t)$$
$$\implies \text{Regret}_t(\rho^\star) \leq \exp(\theta\zeta_t)\left(\theta\sqrt{t} - 1\right) + \sqrt{t} + 1. \tag{31}$$

Setting any $\theta \leq \frac{1}{\sqrt{T}}$ for all $t \geq 1$, the term $\exp(\theta\zeta_t)\left(\theta\sqrt{t} - 1\right)$ in the above inequality, becomes non-positive for any $t \in [T]$. Therefore, we obtain the following upper bound on $\text{Regret}_t(\rho^\star)$ for all $t \in [T]$:

$$\text{Regret}_t(\rho^\star) \leq \sqrt{t} + 1. \tag{32}$$

Owing to the functions $\{\widetilde{\mu}_t\}_{t\geq 1}$ being $\frac{1}{2D}$-Lipschitz, it is easy to realize that $\text{Regret}_t(\rho^\star) = \sum_{\tau=1}^t \widetilde{\mu}_\tau(\rho_\tau) - \widetilde{\mu}_\tau(\rho^\star) \geq -\frac{t}{2}$. For any $t \in [T]$ and $\theta < \frac{1}{\sqrt{T}}$, we write this lower bound along with Eqn. 31 to get:

$$\exp(\theta\zeta_t)\left(\theta\sqrt{t} - 1\right) + \sqrt{t} + 1 \geq -\frac{t}{2}$$
$$\implies \exp(\theta\zeta_t)\left(1 - \theta\sqrt{t}\right) \leq \sqrt{t} + 1 + \frac{t}{2}$$
$$\implies \exp(\theta\zeta_t)\left(1 - \theta\sqrt{t}\right) \leq \frac{2\sqrt{t} + 2 + 2t}{2}$$
$$\implies \exp(\theta\zeta_t) \leq \frac{2\sqrt{t} + 2 + 2t}{2\left(1 - \theta\sqrt{t}\right)}$$
$$\implies \zeta_t \leq \frac{1}{\theta}\ln\frac{2\sqrt{t} + 2 + 2t}{2\left(1 - \theta\sqrt{t}\right)}$$
$$\implies \zeta_T \leq 2\sqrt{T}\ln\left(2\sqrt{T} + 2 + 2T\right), \tag{33}$$

where the last line is obtained by setting $\theta = \frac{1}{2\sqrt{T}}$. By multiplying $\frac{1}{\omega}$ to Eqn. 32 and Eqn. 33, we get the bounds for Eqn. 21 and Eqn. 22. It is straightforward to realize that minimizing Eqn. 21 and Eqn. 22 is equivalent to minimizing Eqn. 17 and Eqn. 18. Therefore, we formally state the bounds on Eqn. 17 and Eqn. 18 in the theorem below.

**Theorem 2.** *Having $\omega = \frac{1}{2LD}$, $L \leq \sqrt{SHA}$, $D = \sqrt{SHA}$, $\varphi(\zeta_T) = \exp(\theta\zeta_T) - 1$, $\theta = \frac{1}{2\sqrt{T}}$, with adversarial loss and constraints, under full feedback, and known transition, the regret and cumulative constraint violation (hard) of* `FAG-K` *(in Algorithm 2) is bounded, $\forall t \in [T]$ as:*

$$\mathcal{R}_t \leq 2SHA\left(\sqrt{t} + 1\right) \text{ and } \mathcal{Z}_T \leq 4SHA\sqrt{T}\ln\left(2\sqrt{T} + 2 + 2T\right). \tag{34}$$

For all the upcoming sections and subsections and for all $t \geq 1$, the definitions of the cost function $\mu_t$, the constraint function $\nu_t$, and the surrogate function $\widehat{\mu}_t$ will be the same as those of Eqn. 19, Eqn. 20, and Eqn. 24 respectively. As a result, the regret decomposition inequality in Eqn. 28 will remain unchanged for all cases and will come in handy in every situation. The online `AdaGrad` policy (as in Algorithm 3) with suitably tailored sub-gradient vectors is used to minimize the surrogate regret in the subsequent cases.

## 4.2 Bandit Feedback and Known Transition

Here, in this subsection, the loss and constraint costs for only the observed state-action pairs (i.e., only the corresponding entries of $\boldsymbol{\ell}_t$ and $\boldsymbol{c}_t$) are revealed to the learner at the end of an episode. The expected regret

$\mathbb{E}[\mathcal{R}_T]$ and the expected cumulative constraint violation $\mathbb{E}[\mathcal{Z}_T]$ to be minimized in this case are:

$$\mathbb{E}[\mathcal{R}_T] := \mathbb{E}\left[\sum_{t=1}^{T} \langle \rho_t - \rho^{\star}, \boldsymbol{\ell}_t \rangle\right], \text{ and} \tag{35}$$

$$\mathbb{E}[\mathcal{Z}_T] := \mathbb{E}\left[\sum_{t=1}^{T} \langle \rho_t, \boldsymbol{c}_t \rangle^+\right]. \tag{36}$$

The learner observes only the values of $H$ state-action pairs for the vectors $\boldsymbol{\ell}_t$ and $\boldsymbol{c}_t$. We employ the widely popular technique of *implicit exploration* (Kocák et al., 2014; Neu, 2015), i.e., a small value is added to the importance weight, to construct biased estimators $\forall t \in [T]$ and $\forall h \in [H]^{-1}$:

$$\widehat{\ell}_{t,h}(s,a) = \frac{\ell_{t,h}(s,a)}{\rho_t(s,a) + \Lambda_t} \mathbf{1}_t(s,a), \text{ and } \widehat{c}_{t,h}(s,a) = \frac{c_{t,h}(s,a)}{\rho_t(s,a) + \Lambda_t} \mathbf{1}_t(s,a), \tag{37}$$

where $\Lambda_t > 0$ is an appropriately chosen parameter (to be fixed later) and $\mathbf{1}_t(s,a)$ is 1 if $(s,a)$ is visited during episode $t$ and 0 otherwise. The estimated loss and constraint-cost vectors are respectively defined as $\widehat{\boldsymbol{\ell}}_t$ and $\widehat{\boldsymbol{c}}_t$, having entries of the form $\widehat{\ell}_{t,h}$ and $\widehat{c}_{t,h}$ for all $t \in [T]$ and $h \in [H]^{-1}$.

Clearly, $\widehat{\boldsymbol{\ell}}_t$ and $\widehat{\boldsymbol{c}}_t$ both have at most $H$ non-zero entries. The term $\Lambda_t$ enforces a minimal exploration in the learner, induces a small bias, and ensures that the variance of the estimator remains bounded (Kocák et al., 2014; Neu, 2015). This trick is essential for keeping the regret and the violation terms under control. We state two useful lemmas below.

**Lemma 1.** *The estimators defined in Eqn. 37 satisfy* $\mathbb{E}_t[\widehat{\ell}_{t,h}(s,a)] = \frac{\ell_{t,h}(s,a)}{\rho_t(s,a)+\Lambda_t}\rho_t(s,a)$, $\mathbb{E}_t[\widehat{c}_{t,h}(s,a)] = \frac{c_{t,h}(s,a)}{\rho_t(s,a)+\Lambda_t}\rho_t(s,a)$, $\mathbb{E}_t[\widehat{\ell}_{t,h}(s,a)^2] \leq \frac{1}{\rho_t(s,a)+\Lambda_t}$, *and* $\mathbb{E}_t[\widehat{c}_{t,h}(s,a)^2] \leq \frac{1}{\rho_t(s,a)+\Lambda_t}$.

*Proof.* See Appendix A.2. □

**Lemma 2.** *Show that* $0 \leq \ell_{t,h}(s,a) - \mathbb{E}_t[\widehat{\ell}_{t,h}(s,a)] \leq \frac{\Lambda \ell_{t,h}(s,a)}{\rho_t(s,a)}$ *and* $0 \leq c_{t,h}(s,a) - \mathbb{E}_t[\widehat{c}_{t,h}(s,a)] \leq \frac{\Lambda c_{t,h}(s,a)}{\rho_t(s,a)}$.

*Proof.* See Appendix A.3. □

Again, for this subsection, the regret and the cumulative constraint violation (hard) of the equivalent COCO problem can be naturally defined as in Eqn. 21 and Eqn. 22. It is not possible to compute the exact subgradient of the surrogate loss under bandit feedback, unlike in the full feedback case. However, we can define a biased estimate of the true sub-gradient $\nabla_t$ (as given in Eqn. 25) of the surrogate loss as follows:

$$\widehat{\nabla}_t = \begin{cases} \omega \widehat{\boldsymbol{\ell}}_t + \varphi'(\zeta_t)\omega \widehat{\boldsymbol{c}}_t, & \text{if } \mathcal{C}_t > 0, \\ \omega \widehat{\boldsymbol{\ell}}_t, & \text{if } \mathcal{C}_t \leq 0, \end{cases} \tag{38}$$

where $\mathcal{C}_t = \sum_{h=0}^{H-1} c_{t,h}(s_h, a_h)$ is the observed constraint violation in the $t$-th episode. Let $\boldsymbol{b}_t$ denote the bias vector for $\widehat{\nabla}_t$ given as: $\boldsymbol{b}_t = \mathbb{E}_t[\widehat{\nabla}_t] - \nabla_t$. We can upper bound the $L^2$-norm of $\boldsymbol{b}_t$ as: $\|\boldsymbol{b}_t\| \leq \omega L + \omega\varphi'(\zeta_t)(L + \sqrt{H}/\Lambda_t)$ (see Appendix A.4 for detailed calculations). Additionally, it is easy to see that the upper bound on the $L^2$-norm of $\widehat{\nabla}_t$ is: $\left\|\widehat{\nabla}_t\right\| \leq \frac{\omega\sqrt{H}}{\Lambda_t}(1 + \varphi'(\zeta_t))$. By the triangle inequality for norms:

$$\left\|\mathbb{E}_t[\widehat{\nabla}_t]\right\| \leq \|\boldsymbol{b}_t\| + \|\nabla_t\| \leq \omega L + \omega\varphi'(\zeta_t)(L + \sqrt{H}/\Lambda_t) + \omega L(1 + \varphi'(\zeta_t)). \tag{39}$$

Our proposed algorithm for this section, **B**andit **A**da**G**rad with **K**nown Transition (`BAG-K`), is described in Algorithm 4. We will use $\widehat{\nabla}_t$ (as given by Eqn. 38) in the online `AdaGrad` policy (described in Algorithm 3)

---

**Algorithm 4** `Bandit AdaGrad with Known Transition (BAG-K)`

---

**Require:** $L$, $D$, Euclidean projection operator $\Pi_\Omega(\cdot)$ on $\Omega$.

Set the parameters $\omega = \frac{1}{2LD}$, $\theta = \frac{D+\frac{1}{2}}{3\sqrt{T}(1+D)^2}$, $\Lambda_t = \omega\sqrt{H}$, and choose $\varphi(\zeta_t) = \exp(\theta\zeta_t) - 1$, $\forall t \geq 1$.

Intialize $\rho_1 \in \Omega$ arbitrarily (e.g., uniformly) and set $\zeta_0 = 0$.

**for** $t = 1, \dots, T$ **do**

    Extract the policy $\pi_t$ such that $\pi_t(a \mid s) \propto \rho_t(s, a)$, $\forall(s, a) \in \mathcal{S} \times \mathcal{A}$.

    The adversary decides $\boldsymbol{\ell}_t$ and $\boldsymbol{c}_t$.

    Set $\mathcal{C}_t \leftarrow 0$

    **for** $h = 0, \dots, H-1$ **do**

        The learner plays $a_h \sim \pi_t(\cdot \mid s_h)$.

        The learner reaches new state $s_{h+1} \sim \mathcal{P}(\cdot \mid s_h, a_h)$ and observes $s_{h+1}$.

    **end for**

    The adversary reveals $\boldsymbol{\ell}_t$ and $\boldsymbol{c}_t$ in *bandit* feedback.

    Compute $\mathcal{C}_t = \sum_{h=0}^{H-1} c_{t,h}(s_h, a_h)$ for the observed state-action pairs.

    Define $\mu_t(\rho_t) = \langle\rho_t, \boldsymbol{\ell}_t\rangle$, and $\nu_t(\rho_t) = \langle\rho_t, \boldsymbol{c}_t\rangle$.

    Compute $\widetilde{\mu}_t \leftarrow \omega\mu_t$, and $\widetilde{\nu}_t \leftarrow \omega(\nu_t)^+$.

    Construct estimators $\widehat{\ell}_{t,h}(s, a)$ and $\widehat{c}_{t,h}(s, a)$ according to Eqn. 37.

    Compute $\zeta_t = \zeta_{t-1} + \widetilde{\nu}_t(\rho_t)$ and $\widehat{\mu}_t(\rho_t) := \widetilde{\mu}_t(\rho_t) + \varphi'(\zeta_t)\widetilde{\nu}_t(\rho_t)$.

    Compute $\widehat{\nabla}_t$ by Eqn. 38.

    Update $\rho_{t+1} = \Pi_\Omega(\rho_t - \eta_t\widehat{\nabla}_t)$, where $\eta_t = \frac{\sqrt{2}D}{2\sqrt{\sum_{\tau=1}^{t}\left\|\widehat{\nabla}_\tau\right\|^2}}$.

**end for**

**return** $\rho_T$ and $\pi_T$.

---

for minimizing the surrogate regret $\text{Regret}'_t(\rho^\star)$. By the convexity of $\widehat{\mu}_\tau$, (for all $\tau \geq 1$), the surrogate regret $\text{Regret}'_t(\rho^\star)$ could be decomposed as:

$$
\begin{aligned}
\text{Regret}'_t(\rho^\star) &= \sum_{\tau=1}^{t} \widehat{\mu}_\tau(\rho_\tau) - \widehat{\mu}_\tau(\rho^\star) \\
&\leq \sum_{\tau=1}^{t} \langle\rho_\tau - \rho^\star, \nabla_\tau\rangle \\
&= \sum_{\tau=1}^{t} \langle\rho_\tau - \rho^\star, \mathbb{E}_\tau[\widehat{\nabla}_\tau]\rangle + \sum_{\tau=1}^{t} \langle\rho_\tau - \rho^\star, \nabla_\tau - \mathbb{E}_\tau[\widehat{\nabla}_\tau]\rangle \\
&= \sum_{\tau=1}^{t} \langle\rho_\tau - \rho^\star, \mathbb{E}_\tau[\widehat{\nabla}_\tau]\rangle + \sum_{\tau=1}^{t} \langle\rho_\tau - \rho^\star, -\boldsymbol{b}_\tau\rangle \\
&= \overbrace{\sum_{\tau=1}^{t} \langle\rho_\tau - \rho^\star, \mathbb{E}_\tau[\widehat{\nabla}_\tau]\rangle}^{T_1} - \overbrace{\sum_{\tau=1}^{t} \langle\rho_\tau - \rho^\star, \boldsymbol{b}_\tau\rangle}^{T_2},
\end{aligned}
\tag{40}
$$

where $T_1$ is simply the regret from Eqn. 29, with $\left\|\mathbb{E}_\tau[\widehat{\nabla}_\tau]\right\|$ being used instead of $\|\nabla_\tau\|$, and $T_2$ is the bias term. The computations for upper bounding $T_1$ and $T_2$, are deferred to Appendix A.5.

Setting $\Lambda_t = \omega\sqrt{H}$ for all $t \geq 1$, and from Eqn. 75 and Eqn. 76 of Appendix A.5, we have:

$$
\text{Regret}'_t(\rho^\star) \leq \sqrt{12t} \cdot \varphi'(\zeta_t) + \frac{\sqrt{6t}}{2} + \frac{\sqrt{12t}}{2} + D\sqrt{12t} \cdot \varphi'(\zeta_t) - \frac{t}{2} - \frac{t}{2} \cdot \varphi'(\zeta_t) - Dt \cdot \varphi'(\zeta_t).
\tag{41}
$$

Choosing $\varphi(\zeta_t) = \exp(\theta\zeta_t) - 1$, $\forall t \geq 1$, and putting Eqn. 41 into the regret decomposition inequality of Eqn. 28, we observe:

$$\varphi(\zeta_t) + \text{Regret}_t(\rho^\star) \leq \text{Regret}'_t(\rho^\star)$$

$$\implies \exp(\theta\zeta_t) - 1 + \text{Regret}_t(\rho^\star) \leq \sqrt{12t} \cdot \theta \exp(\theta\zeta_t) + \frac{\sqrt{6t}}{2} + \frac{\sqrt{12t}}{2} + D\sqrt{12t} \cdot \theta \exp(\theta\zeta_t)$$

$$- \frac{t}{2} - \frac{t}{2} \cdot \theta \exp(\theta\zeta_t) - Dt \cdot \theta \exp(\theta\zeta_t)$$

$$\implies \text{Regret}_t(\rho^\star) \leq \sqrt{12t} \cdot \theta \exp(\theta\zeta_t) + D\sqrt{12t} \cdot \theta \exp(\theta\zeta_t) - \frac{t}{2} \cdot \theta \exp(\theta\zeta_t)$$

$$- Dt \cdot \theta \exp(\theta\zeta_t) - \exp(\theta\zeta_t) + 1 + \frac{\sqrt{6t}}{2} + \frac{\sqrt{12t}}{2} - \frac{t}{2}$$

$$\implies \text{Regret}_t(\rho^\star) \leq \exp(\theta\zeta_t)\left(\theta\sqrt{12t} + \theta D\sqrt{12t} - \frac{\theta t}{2} - \theta Dt - 1\right)$$

$$+ 1 + \frac{\sqrt{6t}}{2} + \frac{\sqrt{12t}}{2}. \tag{42}$$

Let $k(t) = \sqrt{12t} + D\sqrt{12t} - \frac{t}{2} - Dt$ be a function for any $t \geq 1$. The maximum of $k(t)$ occurs at $t^* = \frac{3(1+D)^2}{(D+\frac{1}{2})^2}$ and the maximum value is $k(t^*) = \frac{3(1+D)^2}{D+\frac{1}{2}}$. We express Eqn. 42 as: $\text{Regret}_t(\rho^\star) \leq \exp(\theta\zeta_t)(\theta k(t) - 1) + 1 + \frac{\sqrt{6t}}{2} + \frac{\sqrt{12t}}{2}$. With $\theta = \frac{D+\frac{1}{2}}{3(1+D)^2}$ for all $t \geq 1$, the term $\theta k(t) - 1 \leq 0$, so: $\exp(\theta\zeta_t)(\theta k(t) - 1) \leq 0$. Therefore, by choosing any $\theta \leq \frac{D+\frac{1}{2}}{3(1+D)^2}$, we can bound the regret as:

$$\text{Regret}_t(\rho^\star) \leq 1 + \frac{\sqrt{6t}}{2} + \frac{\sqrt{12t}}{2}, \forall t \in [T]. \tag{43}$$

For any $t \in [T]$, any $\theta < \frac{D+\frac{1}{2}}{3(1+D)^2}$, and utilizing the fact that $\text{Regret}_t(\rho^\star) \geq -\frac{t}{2}$ along with Eqn. 42 we obtain an upper bound on $\zeta_t$:

$$\exp(\theta\zeta_t)(\theta k(t) - 1) + 1 + \frac{\sqrt{6t}}{2} + \frac{\sqrt{12t}}{2} - \frac{t}{2} \geq -\frac{t}{2}$$

$$\implies \exp(\theta\zeta_t)(1 - \theta k(t)) \leq 1 + \frac{\sqrt{6t}}{2} + \frac{\sqrt{12t}}{2}$$

$$\implies \exp(\theta\zeta_t) \leq \frac{1 + \frac{\sqrt{6t}}{2} + \frac{\sqrt{12t}}{2}}{1 - \theta k(t)}$$

$$\implies \zeta_t \leq \frac{1}{\theta} \ln \frac{1 + \frac{\sqrt{6t}}{2} + \frac{\sqrt{12t}}{2}}{1 - \theta\sqrt{12t} + \theta D\sqrt{12t} - \frac{\theta t}{2} - \theta Dt}$$

$$\implies \zeta_T \leq \frac{6\sqrt{T}(1+D)^2}{2D+1} \ln \frac{1 + \frac{\sqrt{6T}}{2} + \frac{\sqrt{12T}}{2}}{1 - \frac{1}{\sqrt{T}}}, \tag{44}$$

where the last line is obtained by setting $\theta = \frac{D+\frac{1}{2}}{3\sqrt{T}(1+D)^2}$. We multiply $\frac{1}{\omega}$ to Eqn. 43 and Eqn. 44 to obtain the bounds for Eqn. 21 and Eqn. 22. In this scenario, minimizing Eqn. 21 and Eqn. 22 leads to an upper bound of Eqn. 35 and Eqn. 36, and we formalize the final bounds in the following theorem.

**Theorem 3.** *Having $\omega = \frac{1}{2LD}$, $L \leq \sqrt{SHA}$, $D = \sqrt{SHA}$, $\varphi(\zeta_T) = \exp(\theta\zeta_T) - 1$, $\theta = \frac{D+\frac{1}{2}}{3\sqrt{T}(1+D)^2}$, with adversarial loss and constraints, under bandit feedback, and known transition, the expected regret and expected cumulative constraint violation (hard) of BAG-K (in Algorithm 4) is bounded, $\forall t \in [T]$ as:*

$$\mathbb{E}[\mathcal{R}_t] \leq 2SHA\left(1 + \frac{\sqrt{6t}}{2} + \frac{\sqrt{12t}}{2}\right), \text{ and } \mathbb{E}[\mathcal{Z}_T] \leq \frac{12\sqrt{TSHA}\left(1 + \sqrt{SHA}\right)^2}{2\sqrt{SHA} + 1} \ln \frac{1 + \frac{\sqrt{6T}}{2} + \frac{\sqrt{12T}}{2}}{1 - \frac{1}{\sqrt{T}}}. \tag{45}$$

# 5 Unknown Transition Function

An unknown transition function for the CMDP $\mathcal{M}$ presents two significant challenges. Firstly, there would be a randomness linked with the next-state $s_{h+1}$ in an episode $t \in [T]$. Therefore, the episodic loss in Eqn. 1 and episodic constraint violation in Eqn. 2 would be applicable throughout this section. We re-mention them below for the sake of convenience:

$$V^{\pi_t}(s_0; \boldsymbol{\ell}_t) := \mathbb{E}\left[\sum_{h=0}^{H-1} \ell_{t,h}(s_h, a_h) \,\middle|\, a_h \sim \pi_t(\cdot \mid s_h), s_{h+1} \sim \mathcal{P}(\cdot \mid s_h, a_h)\right], \text{ and}$$

$$V^{\pi_t}(s_0; \boldsymbol{c}_t) := \mathbb{E}\left[\sum_{h=0}^{H-1} c_{t,h}(s_h, a_h) \,\middle|\, a_h \sim \pi_t(\cdot \mid s_h), s_{h+1} \sim \mathcal{P}(\cdot \mid s_h, a_h)\right].$$

Secondly, the decision space $\Omega$ is not known in advance owing to the unknown $\mathcal{P}$. The occupancy measure of $\pi_t$, i.e., $\rho_t$, is also unknown. We denote by $\Omega_{\mathcal{P}_i} \subset \Omega$ the set of occupancy measures whose induced transition function belongs to a set of transition functions $\mathcal{P}_i$.

To tackle both the aforementioned challenges, we resort to maintaining a *confidence set* for the unknown transition function $\mathcal{P}$ (Burnetas & Katehakis, 1997) and an epoch-doubling strategy (Jin et al., 2020). Let $X_i(s,a)$ and $Y_i(s' \mid s,a)$ denote the total number of visits made by the algorithm to the pair $(s,a)$ and the triplet $(s,a,s')$ before the epoch $i > 1$. For any $i$ and any $h \in [H]^{-1}$, if we have $X_i(s_h, a_h) \geq \max\{1, 2X_{i-1}(s_h, a_h)\}$, then we increment the epoch index $i$ by 1. We define the empirical transition function for the $i$-th epoch as:

$$\bar{\mathcal{P}}_i(s' \mid s, a) = \frac{Y_i(s' \mid s, a)}{\max\{1, X_i(s, a)\}}. \tag{46}$$

For any $\delta \in (0, 1)$, let $\epsilon_i(s' \mid s, a)$ be given by (Jin et al., 2020):

$$\epsilon_i(s' \mid s, a) := 2\sqrt{\frac{\bar{\mathcal{P}}_i(s' \mid s, a) \ln\left(\frac{SAT}{\delta}\right)}{\max\{1, X_i(s, a) - 1\}}} + \frac{14 \ln\left(\frac{SAT}{\delta}\right)}{3 \max\{1, X_i(s, a) - 1\}}. \tag{47}$$

Similarly to Jin et al. (2020), for each triple $(s, a, s')$, we build a confidence set containing all transitions with $\epsilon_i(s' \mid s, a)$ distance from $\bar{\mathcal{P}}_i(s' \mid s, a)$ as given below:

$$\mathcal{P}_i = \left\{\widehat{\mathcal{P}} : \left|\widehat{\mathcal{P}}(s' \mid s, a) - \bar{\mathcal{P}}_i(s' \mid s, a)\right| \leq \epsilon_i(s' \mid s, a), \forall (s, a, s') \in \mathcal{S}_h \times \mathcal{A} \times \mathcal{S}_{h+1}, h = 0, \dots, H-1\right\}. \tag{48}$$

It is naturally understood that, for $i = 1$, $\mathcal{P}_i$ is the set of all transitions such that $\Omega_{\mathcal{P}_i} = \Omega$. In any episode $t \in [T]$, we maintain an occupancy measure $\widehat{\rho}_t$ and execute the induced policy $\pi_t = \pi^{\widehat{\rho}_t}$, because $\rho_t$ is unknown. Again, from Jin et al. (2020), we have: The true transition function $\mathcal{P}$ is present in the confidence set $\mathcal{P}_i$, i.e., $\mathcal{P} \in \mathcal{P}_i, \forall i$, with probability at least $1 - 4\delta$.

## 5.1 Full Feedback and Unknown Transition

Because of full feedback, we get to know every component of the vectors $\boldsymbol{\ell}_t$ and $\boldsymbol{c}_t$ at the end of an episode. The regret $\mathcal{R}_T$ and the hard violation $\mathcal{Z}_T$ to be minimized are respectively given by Eqn. 17 and Eqn. 18. However, since $\rho_t$ is unknown, we cannot compute $\nabla_t$ (as in Eqn. 25) like we did in the full feedback case of Section 4.1. We slightly tweak $\nabla_t$ from Eqn. 25 to obtain an estimated sub-gradient of $\widehat{\mu}_t(\rho_t)$ as:

$$\nabla_t = \begin{cases} \omega\boldsymbol{\ell}_t + \varphi'(\zeta_t)\omega\boldsymbol{c}_t, & \text{if } \langle\widehat{\rho}_t, \omega\boldsymbol{c}_t\rangle > 0, \\ \omega\boldsymbol{\ell}_t, & \text{if } \langle\widehat{\rho}_t, \omega\boldsymbol{c}_t\rangle \leq 0. \end{cases} \tag{49}$$

Instead of $\rho_t$, we here use $\widehat{\rho}_t$ for sign determination, which is perfectly doable. The norm of $\nabla_t$ (as given in Eqn. 49) has the same upper bound as given in Eqn. 26, i.e., $\|\nabla_t\| \leq \omega L\big(1 + \varphi'(\zeta_t)\big)$. The algorithm we propose for this section, named **F**ull **A**da**G**rad with **U**nknown Transition (`FAG-U`), is fully described in Algorithm 5.

---

**Algorithm 5** `Full AdaGrad with Unknown Transition (FAG-U)`

---

**Require:** $L$, $D$, Euclidean projection operator $\Pi_{\Omega_{\mathcal{P}_i}}(\cdot)$ on the decision set $\Omega_{\mathcal{P}_i}$, $\delta \in (0,1)$.

Set the parameters $\omega = \frac{1}{2LD}$, $\theta = \frac{1}{2k(T)}$, and choose $\varphi(\zeta_t) = \exp(\theta\zeta_t) - 1$, $\forall t \geq 1$.
Initialize epoch index $i = 1$ and set $\zeta_0 = 0$.
Initialize $\mathcal{P}_1$ to be the set of all transition functions.
**for** $h = 0, \ldots, H - 1$ and $\forall(s, a, s') \in \mathcal{S}_h \times \mathcal{A} \times \mathcal{S}_{h+1}$ **do**
    Initialize counters: $X_0(s, a) = X_1(s, a) = Y_0(s' \mid s, a) = Y_1(s' \mid s, a) = 0$.
    Initialize occupancy measure $\widehat{\rho}_1(s, a) = \frac{1}{|\mathcal{S}_h| \times |\mathcal{A}| \times |\mathcal{S}_{h+1}|}$.
**end for**
Initialize policy $\pi_1 = \pi^{\widehat{\rho}_1}$.
**for** $t = 1, \ldots, T$ **do**
    The adversary decides $\boldsymbol{\ell}_t$ and $\boldsymbol{c}_t$.
    **for** $h = 0, \ldots, H - 1$ **do**
        The learner plays $a_h \sim \pi_t(\cdot \mid s_h)$.
        The learner reaches new state $s_{h+1} \sim \mathcal{P}(\cdot \mid s_h, a_h)$ and observes $s_{h+1}$.
        $X_i(s_h, a_h) \leftarrow X_i(s_h, a_h) + 1$.
        $Y_i(s_{h+1} \mid s_h, a_h) \leftarrow Y_i(s_{h+1} \mid s_h, a_h) + 1$.
        **if** $X_i(s_h, a_h) \geq \max\{1, 2X_{i-1}(s_h, a_h)\}$ **then**
            $i \leftarrow i + 1$.
            Initialize new counters $\forall(s, a, s') : X_i(s, a) = X_{i-1}(s, a), Y_i(s' \mid s, a) = Y_{i-1}(s' \mid s, a)$.
            Update the confidence set $\mathcal{P}_i$ based on Eqn. 48.
        **end if**
    **end for**
    The adversary reveals $\boldsymbol{\ell}_t$ and $\boldsymbol{c}_t$ in *full* feedback.
    Define $\mu_t(\rho_t) = \langle \rho_t, \boldsymbol{\ell}_t \rangle$, and $\nu_t(\rho_t) = \langle \rho_t, \boldsymbol{c}_t \rangle$.
    Compute $\widetilde{\mu}_t \leftarrow \omega\mu_t$, and $\widetilde{\nu}_t \leftarrow \omega(\nu_t)^+$.
    Compute $\zeta_t = \zeta_{t-1} + \widetilde{\nu}_t(\rho_t)$ and $\widehat{\mu}_t(\rho_t) := \widetilde{\mu}_t(\rho_t) + \varphi'(\zeta_t)\widetilde{\nu}_t(\rho_t)$.
    According to Eqn. 49, compute the subgradient $\nabla_t$.
    Update $\widehat{\rho}_{t+1} = \Pi_{\Omega_{\mathcal{P}_i}}(\widehat{\rho}_t - \eta_t\nabla_t)$, where $\eta_t = \frac{\sqrt{2}D}{2\sqrt{\sum_{\tau=1}^t \|\nabla_\tau\|^2}}$.

    Update policy $\pi_{t+1} = \pi^{\widehat{\rho}_{t+1}}$.
**end for**
**return** $\rho_T$ and $\pi_T$.

---

In Lemma 3, we recall a vital lemma from Jin et al. (2020) regarding how the size of the confidence set $\mathcal{P}_i$ gets smaller with time. This lemma plays a pivotal role in bounding a key term in the decomposition of the surrogate regret.

**Lemma 3.** *Given a collection of transition functions $\{\mathcal{P}_t^s\}_{s\in\mathcal{S}}$ such that $\mathcal{P}_t^s \in \mathcal{P}_{i_t}$. Here, we use $i_t$ to denote the index of the epoch to which episode $t$ belongs. Let $n_t = \left\{\left(s_h, a_h, \ell_{t,h}(s_h, a_h), c_{t,h}(s_h, a_h)\right)\right\}_{h=0}^{H-1}$ be the observation of the learner in episode $t$, and $\mathcal{F}_t$ be the $\sigma$-algebra generated by the observations $(n_1, \ldots, n_{t-1})$. Then, with probability at least $1 - 6\delta$, the following holds:*

$$\sum_{t=1}^{T} \sum_{s\in\mathcal{S}, a\in\mathcal{A}} \left| \rho^{\mathcal{P}_t^s, \pi_t}(s, a) - \rho_t(s, a) \right| = \mathcal{O}\left( HS\sqrt{AT \ln\left(\frac{SAT}{\delta}\right)} \right),$$

*where $\mathcal{P}_{i_t}$ and $\widehat{\rho}_t$ are both $\mathcal{F}_t$-measurable.*

Again, by the convexity of $\widehat{\mu}_\tau$, (for all $\tau \geq 1$), we could decompose the surrogate regret $\text{Regret}_t'(\rho^\star)$ as:

$$\text{Regret}_t'(\rho^\star) = \sum_{\tau=1}^t \widehat{\mu}_\tau(\rho_\tau) - \widehat{\mu}_\tau(\rho^\star) \leq \overbrace{\sum_{\tau=1}^t \langle \widehat{\rho}_\tau - \rho^\star, \nabla_\tau \rangle}^{\text{Reg}} + \overbrace{\sum_{\tau=1}^t \langle \rho_\tau - \widehat{\rho}_\tau, \nabla_\tau \rangle}^{\text{Error}}, \tag{50}$$

where the first term "Reg" is bounded by the regret of AdaGrad used with $\nabla_t$ (as given in Eqn. 49), and the second term "Error" quantifies the error of using $\widehat{\rho}_t$ to approximate $\rho_t$. The detailed derivation of the upper bound on "Error" is in Appendix A.6. We can upper bound $\text{Regret}'_t(\rho^\star)$ as (see Appendix A.7 for details):

$$\text{Regret}'_t(\rho^\star) \leq \left(1 + \varphi'(\zeta_t)\right)\left(2D\omega L\sqrt{t} + \omega LHS\sqrt{At\ln\left(\frac{SAt}{\delta}\right)}\right). \tag{51}$$

Choosing $\varphi(\zeta_t) = \exp(\theta\zeta_t) - 1$, putting $\omega = \frac{1}{2LD}$, $D = \sqrt{SHA}$, $L \leq \sqrt{SHA}$, and substituting Eqn. 51 into the regret decomposition inequality of Eqn. 28, we get:

$$\exp(\theta\zeta_t) - 1 + \text{Regret}_t(\rho^\star) \leq \left(1 + \theta\exp(\theta\zeta_t)\right)\left(2D\omega L\sqrt{t} + \omega LHS\sqrt{At\ln\left(\frac{SAt}{\delta}\right)}\right)$$

$$\implies \text{Regret}_t(\rho^\star) \leq \left(1 + \theta\exp(\theta\zeta_t)\right)\left(\sqrt{t} + \sqrt{\frac{SHt}{4}\ln\left(\frac{SAt}{\delta}\right)}\right) + 1 - \exp(\theta\zeta_t)$$

$$\implies \text{Regret}_t(\rho^\star) \leq 1 + k(t) + \exp(\theta\zeta_t)\left(\theta k(t) - 1\right), \tag{52}$$

where $k(t) = \sqrt{t} + \sqrt{\frac{SHt}{4}\ln\left(\frac{SAt}{\delta}\right)}$. For upper bounding $\text{Regret}_t(\rho^\star)$ in Eqn. 52, we need to choose $\theta$ such that the co-efficient of $\exp(\theta\zeta_t)$ is non-positive. In other words, we require $\theta k(t) - 1 \leq 0 \implies \theta \leq \frac{1}{k(t)}$. Therefore, for any $\theta$ less than or equal to $\frac{1}{k(T)}$, we can bound the regret as:

$$\text{Regret}_t(\rho^\star) \leq 1 + \sqrt{t} + \sqrt{\frac{SHt}{4}\ln\left(\frac{SAt}{\delta}\right)}, \forall t \in [T]. \tag{53}$$

Choosing any $\theta < \frac{1}{k(T)}$, and combining $\text{Regret}_t(\rho^\star) \geq -\frac{t}{2}$ with Eqn. 52, for any $t \in [T]$, we obtain:

$$1 + k(t) + \exp(\theta\zeta_t)\left(\theta k(t) - 1\right) \geq -\frac{t}{2}$$

$$\implies \exp(\theta\zeta_t)\left(1 - \theta k(t)\right) \leq 1 + k(t) + \frac{t}{2}$$

$$\implies \exp(\theta\zeta_t) \leq \frac{1 + k(t) + \frac{t}{2}}{1 - \theta k(t)}$$

$$\implies \zeta_t \leq \frac{1}{\theta}\ln\frac{1 + k(t) + \frac{t}{2}}{1 - \theta k(t)}$$

$$\implies \zeta_T \leq \left(2\sqrt{T} + 2\sqrt{\frac{SHT}{4}\ln\left(\frac{SAT}{\delta}\right)}\right)$$

$$\times \ln\left(2 + 2T + \sqrt{T} + \sqrt{\frac{SHT}{4}\ln\left(\frac{SAT}{\delta}\right)}\right). \tag{54}$$

The last line is obtained by selecting $\theta = \frac{1}{2k(T)} = \frac{1}{2\sqrt{T} + 2\sqrt{\frac{SHT}{4}\ln\left(\frac{SAT}{\delta}\right)}}$. On multiplying $\omega^{-1}$ to Eqn. 53 and Eqn. 54, we get the bounds for Eqn. 21 and Eqn. 22. In this scenario, minimizing Eqn. 21 and Eqn. 22 leads to an upper bound of Eqn. 17 and Eqn. 18, and we formalize the final bounds of `FAG-U` in the following theorem.

**Theorem 4.** *Set the parameters* $\omega = \frac{1}{2LD}$, $L \leq \sqrt{SHA}$, $D = \sqrt{SHA}$, $\theta = \frac{1}{2k(T)}$, *and choose* $\varphi(\zeta_T) = \exp(\theta\zeta_T) - 1$. *Also, we have* $k(T) = \sqrt{T} + \sqrt{\frac{SHT}{4}\ln\left(\frac{SAT}{\delta}\right)}$. *Under adversarial loss and constraints, with full feedback, and unknown transition, the regret* $\mathcal{R}_T$ *and cumulative hard violation* $\mathcal{Z}_T$ *of* `FAG-U` *(in Algorithm 5)*

---

**Algorithm 6** `Bandit AdaGrad with Unknown Transition (BAG-U)`

---

**Require:** $L$, $D$, Euclidean projection operator $\Pi_{\Omega_{\mathcal{P}_i}}(\cdot)$ on the decision set $\Omega_{\mathcal{P}_i}$, $\delta \in (0,1)$.

Set the parameters $\omega = \frac{1}{2LD}$, $\theta = \frac{1}{2m(T)}$, $\Lambda_t = \omega\sqrt{H}$, and choose $\varphi(\zeta_t) = \exp(\theta\zeta_t) - 1$, $\forall t \geq 1$.

Initialize epoch index $i = 1$ and set $\zeta_0 = 0$.

Initialize $\mathcal{P}_1$ to be the set of all transition functions.

**for** $h = 0, \ldots, H-1$ and $\forall(s,a,s') \in \mathcal{S}_h \times \mathcal{A} \times \mathcal{S}_{h+1}$ **do**

    Initialize counters: $X_0(s,a) = X_1(s,a) = Y_0(s' \mid s, a) = Y_1(s' \mid s, a) = 0$.

    Initialize occupancy measure $\widehat{\rho}_1(s,a) = \frac{1}{|\mathcal{S}_h| \times |\mathcal{A}| \times |\mathcal{S}_{h+1}|}$.

**end for**

Initialize policy $\pi_1 = \pi^{\widehat{\rho}_1}$.

**for** $t = 1, \ldots, T$ **do**

    The adversary decides $\boldsymbol{\ell}_t$ and $\boldsymbol{c}_t$.

    Set $\mathcal{C}_t \leftarrow 0$

    **for** $h = 0, \ldots, H-1$ **do**

        The learner plays $a_h \sim \pi_t(\cdot \mid s_h)$.

        The learner reaches new state $s_{h+1} \sim \mathcal{P}(\cdot \mid s_h, a_h)$ and observes $s_{h+1}$.

        $X_i(s_h, a_h) \leftarrow X_i(s_h, a_h) + 1$.

        $Y_i(s_{h+1} \mid s_h, a_h) \leftarrow Y_i(s_{h+1} \mid s_h, a_h) + 1$.

        Compute $u_t(s_h, a_h) = \texttt{COMP-UOB}(\pi_t, s_h, a_h, \mathcal{P}_i)$.

        **if** $X_i(s_h, a_h) \geq \max\{1, 2X_{i-1}(s_h, a_h)\}$ **then**

            $i \leftarrow i + 1$.

            Initialize new counters $\forall(s,a,s') : X_i(s,a) = X_{i-1}(s,a), Y_i(s' \mid s, a) = Y_{i-1}(s' \mid s, a)$.

            Update the confidence set $\mathcal{P}_i$ based on Eqn. 48.

        **end if**

    **end for**

    The adversary reveals $\boldsymbol{\ell}_t$ and $\boldsymbol{c}_t$ in *bandit* feedback.

    Compute $\mathcal{C}_t = \sum_{h=0}^{H-1} c_{t,h}(s_h, a_h)$ for the observed state-action pairs.

    Define $\mu_t(\rho_t) = \langle \rho_t, \boldsymbol{\ell}_t \rangle$, and $\nu_t(\rho_t) = \langle \rho_t, \boldsymbol{c}_t \rangle$.

    Compute $\widetilde{\mu}_t \leftarrow \omega\mu_t$, and $\widetilde{\nu}_t \leftarrow \omega(\nu_t)^+$.

    Construct estimators $\widehat{\ell}_{t,h}(s,a)$ and $\widehat{c}_{t,h}(s,a)$ according to Eqn. 57.

    Compute $\zeta_t = \zeta_{t-1} + \widetilde{\nu}_t(\rho_t)$ and $\widehat{\mu}_t(\rho_t) := \widetilde{\mu}_t(\rho_t) + \varphi'(\zeta_t)\widetilde{\nu}_t(\rho_t)$.

    Compute $\widehat{\nabla}_t$ by Eqn. 58.

    Update $\widehat{\rho}_{t+1} = \Pi_{\Omega_{\mathcal{P}_i}}(\widehat{\rho}_t - \eta_t\widehat{\nabla}_t)$, where $\eta_t = \dfrac{\sqrt{2}D}{2\sqrt{\sum_{\tau=1}^{t}\left\|\widehat{\nabla}_\tau\right\|^2}}$.

    Update policy $\pi_{t+1} = \pi^{\widehat{\rho}_{t+1}}$.

**end for**

**return** $\rho_T$ and $\pi_T$.

---

*are bounded, $\forall t \in [T]$, with probability at least $1 - \mathcal{O}(\delta)$ as:*

$$\mathcal{R}_t \leq 2SHA\left(1 + \sqrt{t} + \sqrt{\frac{SHt}{4}\ln\left(\frac{SAt}{\delta}\right)}\right), \text{ and}$$

$$\mathcal{Z}_T \leq 2SHA\left(2\sqrt{T} + 2\sqrt{\frac{SHT}{4}\ln\left(\frac{SAT}{\delta}\right)}\right)\ln\left(2 + 2T + \sqrt{T} + \sqrt{\frac{SHT}{4}\ln\left(\frac{SAT}{\delta}\right)}\right). \tag{55}$$

## 5.2 Bandit Feedback and Unknown Transition

In this case, the expected regret $\mathbb{E}[\mathcal{R}_T]$ and the expected hard cumulative constraint violation $\mathbb{E}[\mathcal{Z}_T]$ to be minimized are respectively given by Eqn. 35 and Eqn. 36. Due to the unknown occupancy measure $\rho_t$, estimators cannot be constructed using Eqn. 37. Inspired by Jin et al. (2020), we replace $\rho_t(s,a)$ with an

*upper occupancy bound* given by:

$$u_t(s,a) = \max_{\widehat{\mathcal{P}} \in \mathcal{P}_i} \rho^{\widehat{\mathcal{P}}, \pi_t}(s,a). \tag{56}$$

Thus, we can now have the following estimators:

$$\widehat{\ell}_{t,h}(s,a) = \frac{\ell_{t,h}(s,a)}{u_t(s,a) + \Lambda_t} \mathbf{1}_t(s,a), \text{ and } \widehat{c}_{t,h}(s,a) = \frac{c_{t,h}(s,a)}{u_t(s,a) + \Lambda_t} \mathbf{1}_t(s,a), \tag{57}$$

where $\Lambda_t > 0$ is an appropriately chosen parameter (to be fixed later) and $\mathbf{1}_t(s,a)$ is 1 if $(s,a)$ is visited during episode $t$ and 0 otherwise. The estimated loss and constraint-cost vectors are respectively defined as $\widehat{\ell}_t$ and $\widehat{c}_t$, having entries of the form $\widehat{\ell}_{t,h}$ and $\widehat{c}_{t,h}$ for all $t \in [T]$ and $h \in [H]^{-1}$. Clearly, $\widehat{\ell}_t$ and $\widehat{c}_t$ both have at most $H$ non-zero entries. Unlike Eqn. 49, we cannot fully compute the sub-gradient. Hence, we resort to a biased estimate as follows:

$$\widehat{\nabla}_t = \begin{cases} \omega\widehat{\ell}_t + \varphi'(\zeta_t)\omega\widehat{c}_t, & \text{if } \mathcal{C}_t > 0, \\ \omega\widehat{\ell}_t, & \text{if } \mathcal{C}_t \leq 0, \end{cases} \tag{58}$$

where $\mathcal{C}_t = \sum_{h=0}^{H-1} c_{t,h}(s_h, a_h)$ is the observed constraint violation in the $t$-th episode. Let $\boldsymbol{b}_t$ denote the bias vector of $\widehat{\nabla}_t$ which is given by: $\boldsymbol{b}_t = \mathbb{E}_t[\widehat{\nabla}_t] - \nabla_t$. Performing similar calculations as in Appendix A.4, it can be shown for $\boldsymbol{b}_t$ and $\widehat{\nabla}_t$ (as given in Eqn. 58) that,

$$\|\boldsymbol{b}_t\| \leq \omega L + \omega\varphi'(\zeta_t)\big(L + \sqrt{H}/\Lambda_t\big), \text{ and } \left\|\widehat{\nabla}_t\right\| \leq \frac{\omega\sqrt{H}}{\Lambda_t}\big(1 + \varphi'(\zeta_t)\big).$$

Thus, implying by the triangle inequality for norms:

$$\left\|\mathbb{E}_t[\widehat{\nabla}_t]\right\| \leq \|\boldsymbol{b}_t\| + \|\nabla_t\|$$
$$\leq \omega L + \omega\varphi'(\zeta_t)\big(L + \sqrt{H}/\Lambda_t\big) + \omega L\big(1 + \varphi'(\zeta_t)\big). \tag{59}$$

We chalked the **B**andit **A**da**G**rad with **U**nknown Transition (`BAG-U`) algorithm for this section. It is formally depicted in Algorithm 6, and the `COMP-UOB` method is as given in Algorithm 3 of Jin et al. (2020). By the convexity of $\widehat{\mu}_\tau$, (for all $\tau \geq 1$), the surrogate regret $\text{Regret}'_t(\rho^\star)$ could be decomposed into four terms as:

$$\text{Regret}'_t(\rho^\star) = \sum_{\tau=1}^{t} \widehat{\mu}_\tau(\rho_\tau) - \widehat{\mu}_\tau(\rho^\star)$$
$$\leq \sum_{\tau=1}^{t} \langle \rho_\tau - \rho^\star, \nabla_\tau \rangle$$
$$\leq \sum_{\tau=1}^{t} \langle \widehat{\rho}_\tau - \rho^\star, \nabla_\tau \rangle + \sum_{\tau=1}^{t} \langle \rho_\tau - \widehat{\rho}_\tau, \nabla_\tau \rangle$$
$$\leq \overbrace{\sum_{\tau=1}^{t} \langle \widehat{\rho}_\tau - \rho^\star, \widehat{\nabla}_\tau \rangle}^{\text{Reg}} + \overbrace{\sum_{\tau=1}^{t} \langle \rho_\tau - \widehat{\rho}_\tau, \nabla_\tau \rangle}^{\text{Error}} + \overbrace{\sum_{\tau=1}^{t} \langle \widehat{\rho}_\tau, \nabla_\tau - \widehat{\nabla}_\tau \rangle}^{\text{Bias1}} + \overbrace{\sum_{\tau=1}^{t} \langle \rho^\star, \widehat{\nabla}_\tau - \nabla_\tau \rangle}^{\text{Bias2}}, \tag{60}$$

where "Reg" is simply bounded by the regret of AdaGrad used with $\widehat{\nabla}_t$ (as presented in Eqn. 58), "Error" is the error of using $\widehat{\rho}_t$ to approximate $\rho_t$, "Bias1" measures how much $\widehat{\nabla}_\tau$ underestimates $\nabla_\tau$ weighted by $\widehat{\rho}_\tau$, and "Bias2" measures the error of $\widehat{\nabla}_\tau$ relative to $\nabla_\tau$ when weighted by $\rho^\star$.

With probability at least $1 - \mathcal{O}(\delta)$ and with $\Lambda_t = \omega\sqrt{H}$, we have the following upper bound on $\text{Regret}'_t(\rho^\star)$ (see Appendix A.8 for detailed calculations):

$$\text{Regret}'_t(\rho^\star) \leq 2D\sqrt{t}\left(1 + \varphi'(\zeta_t)\right) + \omega LHS\sqrt{At\ln\left(\frac{SAt}{\delta}\right)} \cdot \left(1 + \varphi'(\zeta_t)\right)$$

$$+ \frac{2\left(1 + \varphi'(\zeta_t)\right)}{\sqrt{H}}\sqrt{2t\ln\left(2/\delta\right)} - \omega L - \omega Lt \cdot \varphi'(\zeta_t) - t \cdot \varphi'(\zeta_t)$$

$$+ \omega H\ln\frac{H}{\delta} + \omega\varphi'(\zeta_t) \cdot H\ln\frac{H}{\delta} - \omega t \cdot \varphi'(\zeta_t). \tag{61}$$

Substituting Eqn. 61 into the regret decomposition inequality of Eqn. 28, and choosing $\varphi(\zeta_t) = \exp(\theta\zeta_t) - 1$, we have:

$$\exp(\theta\zeta_t) - 1 + \text{Regret}_t(\rho^\star) \leq 2D\sqrt{t}\left(1 + \theta\exp(\theta\zeta_t)\right) + \omega LHS\sqrt{At\ln\left(\frac{SAt}{\delta}\right)} \cdot \left(1 + \theta\exp(\theta\zeta_t)\right)$$

$$+ \frac{2\left(1 + \theta\exp(\theta\zeta_t)\right)}{\sqrt{H}}\sqrt{2t\ln\left(2/\delta\right)} - \omega L - \omega Lt \cdot \theta\exp(\theta\zeta_t)$$

$$- t \cdot \theta\exp(\theta\zeta_t) + \omega H\ln\frac{H}{\delta} + \omega\theta\exp(\theta\zeta_t) \cdot H\ln\frac{H}{\delta} - \omega t \cdot \theta\exp(\theta\zeta_t).$$

Grouping all the terms involving $\exp(\theta\zeta_t)$ into one side in the above expression,

$$\text{Regret}_t(\rho^\star) \leq \exp(\theta\zeta_t)\Bigg(\theta 2D\sqrt{t} + \theta\omega LHS\sqrt{At\ln\left(\frac{SAt}{\delta}\right)} + \frac{2\theta}{\sqrt{H}}\sqrt{2t\ln\left(2/\delta\right)}$$

$$- \theta\omega Lt - \theta t + \theta\omega H\ln\frac{H}{\delta} - \theta\omega t - 1\Bigg) + 2D\sqrt{t} + \omega LHS\sqrt{At\ln\left(\frac{SAt}{\delta}\right)}$$

$$+ \frac{2}{\sqrt{H}}\sqrt{2t\ln\left(2/\delta\right)} - \omega L + \omega H\ln\frac{H}{\delta} + 1$$

$$\implies \text{Regret}_t(\rho^\star) \leq 2D\sqrt{t} + \omega LHS\sqrt{At\ln\left(\frac{SAt}{\delta}\right)} + \frac{2}{\sqrt{H}}\sqrt{2t\ln\left(2/\delta\right)} - \omega L + \omega H\ln\frac{H}{\delta} + 1$$

$$+ \exp(\theta\zeta_t)\Bigg(\theta 2D\sqrt{t} + \theta\omega LHS\sqrt{At\ln\left(\frac{SAt}{\delta}\right)} + \frac{2\theta}{\sqrt{H}}\sqrt{2t\ln\left(2/\delta\right)}$$

$$+ \theta\omega H\ln\frac{H}{\delta} - 1\Bigg). \tag{62}$$

Let $m(t) = 2D\sqrt{t} + \omega LHS\sqrt{At\ln\left(\frac{SAt}{\delta}\right)} + \frac{2}{\sqrt{H}}\sqrt{2t\ln\left(2/\delta\right)} + \omega H\ln\frac{H}{\delta}$, for all $t \in [T]$. We can rewrite the regret in Eqn. 62 as: $\text{Regret}_t(\rho^\star) \leq 2D\sqrt{t} + \omega LHS\sqrt{At\ln\left(\frac{SAt}{\delta}\right)} + \frac{2}{\sqrt{H}}\sqrt{2t\ln\left(2/\delta\right)} - \omega L + \omega H\ln\frac{H}{\delta} + 1 + \exp(\theta\zeta_t)\left(\theta m(t) - 1\right)$. Thus, having any $\theta \leq \frac{1}{m(T)}$, we can ensure that the regret is nicely bounded with probability at least $1 - \mathcal{O}(\delta)$, as given below:

$$\text{Regret}_t(\rho^\star) \leq 2D\sqrt{t} + \omega LHS\sqrt{At\ln\left(\frac{SAt}{\delta}\right)} + \frac{2}{\sqrt{H}}\sqrt{2t\ln\left(2/\delta\right)} - \omega L + \omega H\ln\frac{H}{\delta} + 1, \forall t \in [T]. \tag{63}$$

Selecting any $\theta < \frac{1}{m(T)}$, and combining $\text{Regret}_t(\rho^\star) \geq -\frac{t}{2}$ with Eqn. 62, we obtain an upper bound on $\zeta_t$, for any $t \in [T]$, with probability at least $1 - \mathcal{O}(\delta)$ as follows:

$$2D\sqrt{t} + \omega LHS\sqrt{At\ln\left(\frac{SAt}{\delta}\right)} + \frac{2}{\sqrt{H}}\sqrt{2t\ln(2/\delta)} - \omega L + \omega H\ln\frac{H}{\delta} + 1 + \exp(\theta\zeta_t)\left(\theta m(t) - 1\right) \geq -\frac{t}{2}$$

$$\implies \exp(\theta\zeta_t)\left(1 - \theta m(t)\right) \leq 1 + 2D\sqrt{t} + \omega LHS\sqrt{At\ln\left(\frac{SAt}{\delta}\right)} + \frac{2}{\sqrt{H}}\sqrt{2t\ln(2/\delta)} - \omega L + \omega H\ln\frac{H}{\delta} + \frac{t}{2}$$

$$\implies \exp(\theta\zeta_t) \leq \frac{1 + 2D\sqrt{t} + \omega LHS\sqrt{At\ln\left(\frac{SAt}{\delta}\right)} + \frac{2}{\sqrt{H}}\sqrt{2t\ln(2/\delta)} - \omega L + \omega H\ln\frac{H}{\delta} + \frac{t}{2}}{1 - \theta m(t)}$$

$$\implies \zeta_t \leq \frac{1}{\theta}\ln\frac{1 + 2D\sqrt{t} + \omega LHS\sqrt{At\ln\left(\frac{SAt}{\delta}\right)} + \frac{2}{\sqrt{H}}\sqrt{2t\ln(2/\delta)} - \omega L + \omega H\ln\frac{H}{\delta} + \frac{t}{2}}{1 - \theta m(t)}$$

$$\implies \zeta_T \leq 4D\sqrt{T} + 2\omega LHS\sqrt{AT\ln\left(\frac{SAT}{\delta}\right)} + \frac{4}{\sqrt{H}}\sqrt{2T\ln(2/\delta)} + 2\omega H\ln\frac{H}{\delta}$$

$$\times \ln\left(2 + 4D\sqrt{T} + 2\omega LHS\sqrt{AT\ln\left(\frac{SAT}{\delta}\right)} + \frac{4}{\sqrt{H}}\sqrt{2T\ln(2/\delta)} - 2\omega L + 2\omega H\ln\frac{H}{\delta} + T\right), \quad (64)$$

where the last line is obtained by choosing $\theta = \frac{1}{2m(T)}$. Putting $\omega = \frac{1}{2LD}$, $L \leq \sqrt{SHA}$, and $D = \sqrt{SHA}$ into Eqn. 63 we have $\forall t \in [T]$:

$$\text{Regret}_t(\rho^\star) \leq 2\sqrt{SHAt} + \frac{1}{2}\sqrt{SHt\ln\left(\frac{SAt}{\delta}\right)} + \frac{2}{\sqrt{H}}\sqrt{2t\ln(2/\delta)} - \frac{1}{2\sqrt{SHA}} + \frac{H}{2SHA}\ln\frac{H}{\delta} + 1$$

$$\implies \text{Regret}_t(\rho^\star) \leq \mathcal{O}\left(\sqrt{SHAt} + \sqrt{SHt\ln\left(\frac{SAt}{\delta}\right)} + \sqrt{\frac{t\ln(2/\delta)}{H}}\right). \quad (65)$$

Putting $\omega = \frac{1}{2LD}$, $L \leq \sqrt{SHA}$, and $D = \sqrt{SHA}$ into Eqn. 64 we obtain:

$$\zeta_T \leq 4\sqrt{SHAT} + \sqrt{SHT\ln\left(\frac{SAT}{\delta}\right)} + \frac{4}{\sqrt{H}}\sqrt{2T\ln(2/\delta)} + \frac{1}{SA}\ln\frac{H}{\delta}$$

$$\times \ln\left(2 + 4\sqrt{SHAT} + \sqrt{SHT\ln\left(\frac{SAT}{\delta}\right)} + \frac{4}{\sqrt{H}}\sqrt{2T\ln(2/\delta)} - \frac{1}{\sqrt{SHA}} + \frac{1}{SA}\ln\frac{H}{\delta} + T\right)$$

$$\implies \zeta_T \leq \mathcal{O}\left(\sqrt{SHAT} + \sqrt{SHT\ln\left(\frac{SAT}{\delta}\right)} + \sqrt{\frac{T\ln(2/\delta)}{H}}\right.$$

$$\left.\times \ln\left(\sqrt{SHAT} + \sqrt{SHT\ln\left(\frac{SAT}{\delta}\right)} + \sqrt{\frac{T\ln(2/\delta)}{H}} + T\right)\right). \quad (66)$$

Scaling back Eqn. 65 and Eqn. 66 by a factor of $\frac{1}{\omega}$ respectively attains an upper bound for Eqn. 35 and Eqn. 36. We formally state the final bounds in the theorem below.

**Theorem 5.** *We set the parameters $\delta \in (0,1)$, $\theta = \frac{1}{2m(T)}$, $\omega = \frac{1}{2LD}$, $L \leq \sqrt{SHA}$, $D = \sqrt{SHA}$, and choose $\varphi(\zeta_T) = \exp(\theta\zeta_T) - 1$. Also, we have $m(T) = 2D\sqrt{T} + \omega LHS\sqrt{AT\ln\left(\frac{SAT}{\delta}\right)} + \frac{2}{\sqrt{H}}\sqrt{2T\ln(2/\delta)} + \omega H\ln\frac{H}{\delta}$. Having adversarial loss and constraints, under bandit feedback, and unknown transition, the expected regret and the expected cumulative constraint violation (hard) of BAG-U (in Algorithm 6) are bounded, $\forall t \in [T]$,*

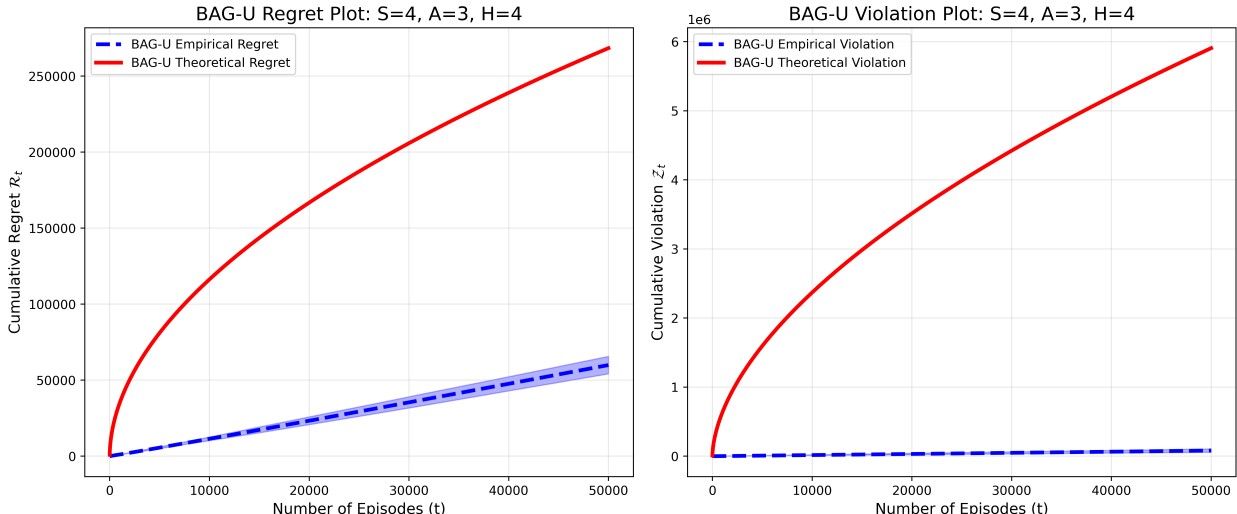

Figure 3: Comparing the empirical regret and empirical violation of `BAG-U` with its corresponding theoretical values on an adversarial CMDP instance with $S = 4$, $A = 3$, $H = 4$. The empirical regret and violation curves were plotted by averaging over five independent runs (with different seeds) and a 95% confidence interval. The solid red line represents the worst-case theoretical regret and hard violation values, while the dashed blue line is for the empirical ones.

*with probability at least* $1 - \mathcal{O}(\delta)$ *as:*

$$\mathbb{E}[\mathcal{R}_t] \leq \mathcal{O}\left((SHA)^{\frac{3}{2}}\sqrt{t} + SHA\sqrt{SHt\ln\left(\frac{SAt}{\delta}\right)} + SHA\sqrt{\frac{t\ln(2/\delta)}{H}}\right), \ and \tag{67}$$

$$\mathbb{E}[\mathcal{Z}_T] \leq \mathcal{O}\left((SHA)^{\frac{3}{2}}\sqrt{T} + SHA\sqrt{SHT\ln\left(\frac{SAT}{\delta}\right)} + SHA\sqrt{\frac{T\ln(2/\delta)}{H}}\right.$$
$$\left. \times \ln\left((SHA)^{\frac{3}{2}}\sqrt{T} + SHA\sqrt{SHT\ln\left(\frac{SAT}{\delta}\right)} + SHA\sqrt{\frac{T\ln(2/\delta)}{H}} + T\right)\right). \tag{68}$$

All of our proposed algorithms, i.e., `FAG-K`, `BAG-K`, `FAG-U`, and `BAG-U`, perform only one Euclidean projection onto $\Omega$ per episode. Since $\Omega$ is a simple polytope (as given in Definition 2), the projection amounts to solving a sparse quadratic program with linear flow constraints. In contrast, primal-dual methods (Stradi et al., 2024a;b; 2025a; Müller et al., 2024) must maintain dual variables and update them at each step, which requires two expensive coupled updates (e.g., adding regularizers and using approximations to the Lagrangian). Hence, the computational cost of our updates is lower: one first-order gradient step followed by a single projection, without dependence on Slater-type conditions or instance-dependent feasible policies.

## 6 Experimental Evaluations

We evaluate the performance of `FAG-K`, `BAG-K`, `FAG-U`, and `BAG-U` in solving CMDP instances. The experiments have been designed as follows: First, a loop-free, finite-horizon (i.e., each episode has length $H$), and episodic CMDP is created by exactly following the setup described in Section 3.1; Second, each algorithm is implemented to solve the CMDP, tracking the cumulative regret and cumulative hard constraint violation in the process. We term them *empirical regret* and *empirical violation*, i.e., the actual cumulative regret and actual cumulative hard violation obtained by the learning algorithm while solving a CMDP. On the other hand, *theoretical regret* and *theoretical violation* refer to the worst-case bounds of the algorithms as provided in Theorem 2–5.

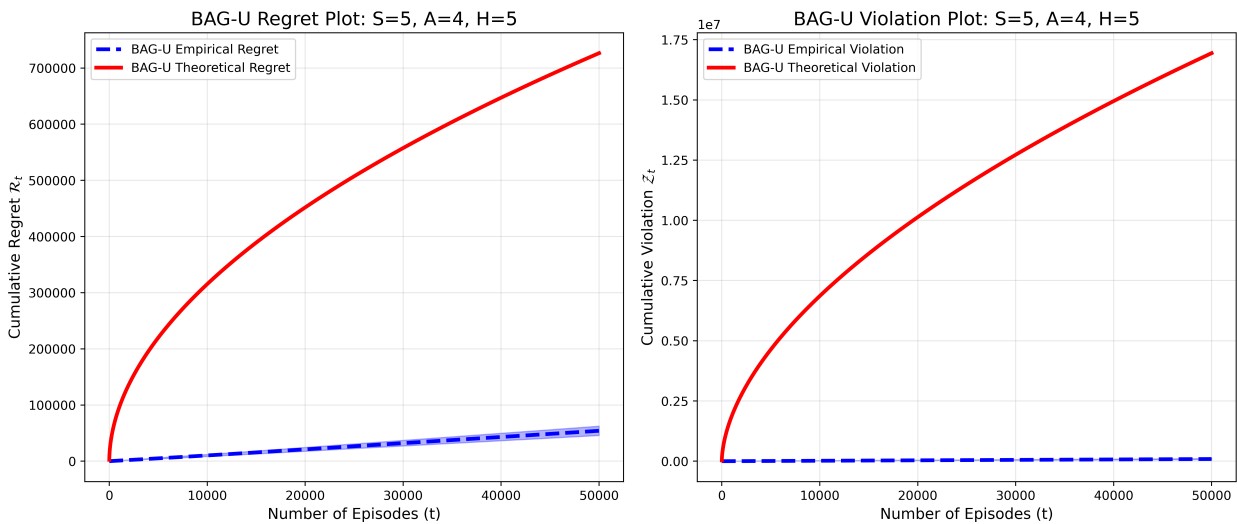

Figure 4: Theoretical regret (and violation) vs empirical regret (and violation) of `BAG-U` on a CMDP with $S = 5$, $A = 4$, $H = 5$. The empirical curves are averaged over five runs, with 95% confidence intervals.

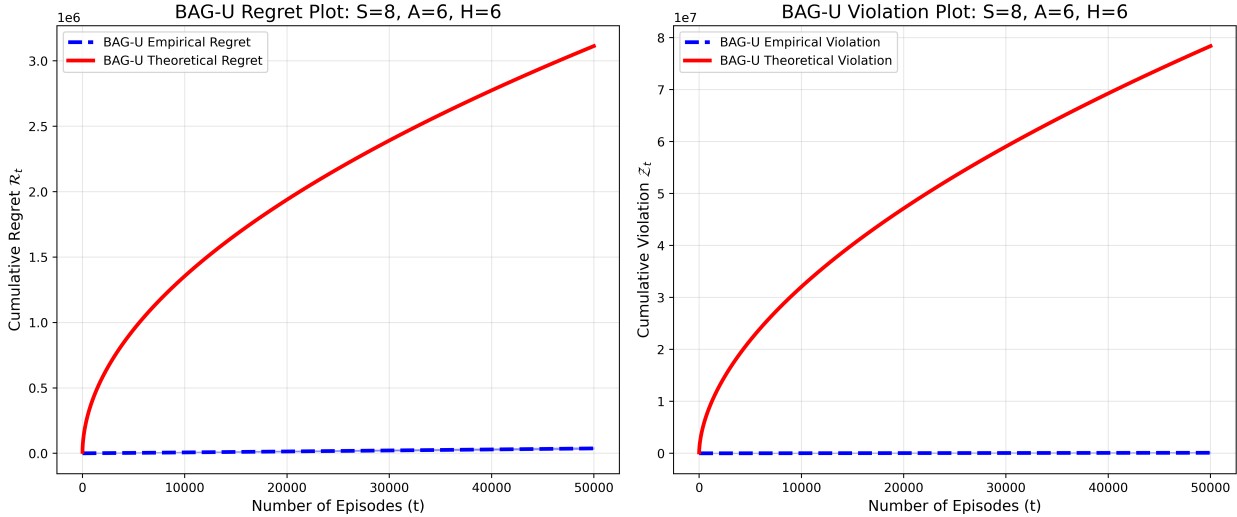

Figure 5: Theoretical regret (and violation) vs empirical regret (and violation) of `BAG-U` on a CMDP with $S = 8$, $A = 6$, $H = 6$. The empirical curves are averaged over five runs, with 95% confidence intervals.

All algorithms were run for $T = 50000$ episodes in each experiment. The adversarial losses and constraints are generated via an Online Gradient Descent (OGD) algorithm (Orabona, 2025), which takes as a gradient a vector that contains a fixed initial vector of losses (or constraints) and the negative product of the policy played at that round for each state. Each algorithm was executed five times independently with different random seeds, and we report the average across these runs, along with 95% confidence intervals. The confidence parameter $\delta$ is set to 0.01 for all experiments. Moreover, all the algorithms have been tested on three CMDP instances, i.e., $S = 4$, $A = 3$, $H = 4$; $S = 5$, $A = 4$, $H = 5$; and $S = 8$, $A = 6$, $H = 6$. However, we only present the evaluation results of `BAG-U` in this section (as it is the solution to CQ), while the results for `FAG-K`, `BAG-K`, and `FAG-U` have been deferred to Appendix A.9, Appendix A.10, and Appendix A.11.

Figure 3 compares the theoretical regret and violation with the empirical regret and violation of the `BAG-U` algorithm that solved the adversarial CMDP with $S = 4$, $A = 3$, $H = 4$. The solid red line represents the

Table 2: Listing the regret and hard violation bounds of all four algorithms. We use the classic $\widetilde{\mathcal{O}}(\cdot)$ notation, which ignores all the logarithmic factors.

| Algorithm | Transition | Feedback | Regret Bound | Hard Violation Bound |
|:---:|:---:|:---:|:---:|:---:|
| FAG-K | Known | Full | $\mathcal{O}(SHA\sqrt{T})$ | $\widetilde{\mathcal{O}}(SHA\sqrt{T})$ |
| BAG-K | Known | Bandit | $\mathcal{O}(SHA\sqrt{T})$ | $\widetilde{\mathcal{O}}(SHA\sqrt{T})$ |
| FAG-U | Unknown | Full | $\widetilde{\mathcal{O}}(SHA\sqrt{T})$ | $\widetilde{\mathcal{O}}(SHA\sqrt{T})$ |
| BAG-U | Unknown | Bandit | $\widetilde{\mathcal{O}}(S^{\frac{3}{2}}H^{\frac{3}{2}}A^{\frac{3}{2}}\sqrt{T})$ | $\widetilde{\mathcal{O}}(S^{\frac{3}{2}}H^{\frac{3}{2}}A^{\frac{3}{2}}\sqrt{T})$ |

worst-case theoretical regret and hard-violation values of `BAG-U`, while the dashed blue line represents the empirical ones. The x-axis captures the number of episodes, and the y-axis represents the cumulative regret $\mathcal{R}_t$ and the cumulative violation $\mathcal{Z}_t$. Similarly, the plots in Figure 4 and Figure 5 compare the theoretical and empirical performance of `BAG-U` in respectively solving the adversarial CMDP with $S = 5$, $A = 4$, $H = 5$ and $S = 8$, $A = 6$, $H = 6$.

The plots in Figure 3, Figure 4, and Figure 5 show that the empirical regret (blue) grows sublinearly and stays consistently below the theoretical envelope (red). It confirms that the actual regret incurred by `BAG-U` is not only sublinear but also significantly lower than the worst-case bound provided in Theorem 5. The observed $\widetilde{\mathcal{O}}(\sqrt{T})$ trend validates the theoretical regret guarantee even under the most challenging conditions: adversarial losses and constraints, bandit feedback, and unknown transitions. Similarly, the empirical cumulative hard violation (blue) remains sublinear and well below the theoretical curve (red). As the empirical violation is consistently much less than the theoretical upper bound, `BAG-U` effectively controls constraint violations in practice, even without access to a strictly feasible policy or Slater's condition. The plots in Appendix A.9, Appendix A.10, and Appendix A.11 clearly indicate that the observed behavior is consistent: each algorithm achieves sublinear empirical regret and sublinear empirical violation that are orders of magnitude smaller than their corresponding theoretical bounds[2].

**Visual interpretation of sublinear growth:** Note that the empirical curves in Figure 3, Figure 4, and Figure 5 might appear to rise in an approximately straight line over the plotted range. This is expected because a $\sqrt{T}$ function (which is sublinear) can look nearly linear on a standard scale, especially over a sufficient number of episodes, i.e., $T = 50,000$. The critical observation is that the empirical curves remain consistently below the theoretical $\widetilde{\mathcal{O}}(\sqrt{T})$ limit. Since the red curve itself represents a sublinear upper bound, the empirical performance is necessarily sublinear as well. For clarity, one could plot the same data on a log-log scale to make the sublinear growth more visually apparent. However, the linear-scale plots suffice to confirm that the theoretical bounds are not violated and that the algorithms perform significantly better than the worst-case analysis predicts.

# 7 Optimality of the bounds

**Minimax Optimality:** It is stated in Jin et al. (2018) and Jin et al. (2020) that the regret of any algorithm for solving episodic unconstrained adversarial MDPs with full feedback should be at least $\Omega(\sqrt{H^2SAT})$. To the best of our knowledge, no regret and violation lower bounds are known for episodic adversarial CMDPs. For COCO with adversarial constraints, a lower bound of $\Omega(\sqrt{T})$ exists for both regret and hard constraint violation (Sinha & Vaze, 2024). Owing to all the aforementioned results from different settings, we believe that the $\widetilde{\mathcal{O}}(\sqrt{T})$ regret and violation bounds in our adversarial CMDPs ($\mathcal{O}(\sqrt{T})$ regret for known transitions) are tight and cannot be improved in the minimax sense. This optimality holds across all four feedback/transition settings we address, making ours the first comprehensive set of minimax-optimal algorithms for adversarial CMDPs with hard cumulative constraint violation, without Slater's condition, and without access to a strictly feasible policy. The trade-off between regret and violation is necessary because aggressive loss minimization often violates constraints. Our derived $\widetilde{\mathcal{O}}(\sqrt{T})$ bounds show that both can be

---

[2]Implementations are available here.

sublinear, implying that the learner approaches optimality without incurring unbounded violations. This framework directly applies to budget-constrained settings (e.g., auction bidding): regret quantifies the loss of utility, and violation tracks the budget overrun. For an easy interpretation, sublinear regret and violation imply that the average per-episode performance converges to the optimal feasible policy.

**Constant Factors:** Like any other well-known algorithm in the vast expanse of online learning in finite-horizon episodic CMDPs, the effect of the constants (i.e., every variable apart from $T$) can matter in practice. In Table 2, we re-state all our derived bounds as given in Theorem 2, Theorem 3, Theorem 4, and Theorem 5. The results of Germano et al. (2023) and Stradi et al. (2024b) are not directly comparable with ours because, although they consider adversarial loss and constraints, their $\widetilde{\mathcal{O}}(\sqrt{T})$ bounds are reliant on the slackness parameter of Slater's condition. However, for the sake of a loose comparison, we mention that both works have a $SH^2A$ factor in their bounds. As stated in Theorem 5.1 of Zhu et al. (2025), constant factors of $S^2AH^3$ and $H^{\frac{3}{2}}\sqrt{SA}$ are present both in the regret and violation bounds. Given our challenging problem setup, the gaps we close, and the optimal bounds we derive without assumptions, we argue that the constants of $SHA$ and $S^{\frac{3}{2}}H^{\frac{3}{2}}A^{\frac{3}{2}}$ in our attained results might not be optimal, but are not too bad either. In the light of this statement, we leave an intriguing open problem as a future work: improving the $SHA$ and $S^{\frac{3}{2}}H^{\frac{3}{2}}A^{\frac{3}{2}}$ dependence, respectively, for known and unknown transitions in fully adversarial CMDPs. As noted, lower bounds for adversarial CMDPs have not yet been established. However, based on adversarial MDPs scaling as $\Omega(\sqrt{H^2SAT})$, and COCO scaling as $\Omega(\sqrt{T})$, we conjecture the lower bound for adversarial CMDPs is likely $\Omega(H\sqrt{SAT})$.

# 8 Conclusion

Without access to any strictly feasible policy and Slater's condition, this is the first work to tackle and solve the hallowed problem of online learning in finite-horizon episodic CMDPs under adversarial losses and constraints, bandit feedback, and unknown transition dynamics. Our bounds ensure the learner achieves near-optimal loss (i.e., $\widetilde{\mathcal{O}}(\sqrt{T})$ regret) while keeping total hard violations bounded by $\widetilde{\mathcal{O}}(\sqrt{T})$. In practice, this means safe exploration in adversarial environments, unlike soft violation, which allows compensatory negatives. By leveraging a reduction to COCO and building on the techniques introduced by the seminal work of Sinha & Vaze (2024), we developed simple and efficient algorithms that require only a single Euclidean projection per episode. Our approach achieves optimal regret and hard cumulative constraint violation bounds across all four combinations of known-unknown transitions and full-bandit feedback settings – without relying on Slater's condition or any knowledge of a strictly feasible policy. In other words, we make no additional assumptions except for the standard assumptions in the COCO literature.

Our results not only close several theoretical gaps in the literature but also provide a unified, pedagogically valuable framework for understanding the connections between online learning in CMDPs and COCO. The construction of biased estimators for bandit feedback settings may also be of independent interest for future research and educational purposes. Moreover, we validate our theoretical results through rigorous experiments. This work lays the foundation for more practical and robust constrained reinforcement learning systems, opening new avenues for exploring the interplay among online learning, constrained convex optimization, and adversarial CMDPs.

# 9 Future Directions

One can view our work in the tabular setting, and an interesting idea is extending our guarantees to large state-action spaces, a typical characteristic of deep RL. Recent frameworks for uncertainty propagation in model-free RL, such as Wasserstein Actor-Critic (Likmeta et al., 2023) and others (Metelli et al., 2019; Roy et al., 2026), demonstrate that posterior estimations can be effectively scaled into large state-action spaces. Integrating such uncertainty-propagation mechanisms into our algorithms could bridge the gap between theoretical safety guarantees and practical high-dimensional applications.

Many interesting works couple function approximation with MDPs and CMDPs. For example, in adversarial MDPs with linear function approximation (Dai et al., 2023), refined regret bounds have been derived that align with our $\widetilde{\mathcal{O}}(\sqrt{T})$ rates, thereby enabling linear projections over feature spaces rather than full

tabular representations. For more general function approximation, safe representation learning in CMDPs has been explored (Ding & Lavaei, 2023), showing how embeddings can be learned to satisfy constraints episodically. All these strategies could augment our algorithms in deep RL settings, such as Soft Actor-Critic (SAC) (Haarnoja et al., 2018) and other actor-critic methods (Chen et al., 2021; Roy et al., 2023; Yang et al., 2021), by embedding adversarial robustness into the critic network.

It is worth noting that "hard constraint" is sometimes interpreted more stringently as requiring trajectory-level or per-episode safety guarantees. For instance, ensuring that with high probability, each trajectory avoids catastrophic events (e.g., a self-driving car never collides). Our notion of cumulative hard violation, while still much stronger than soft violation, is an aggregate measure over the entire learning process. Although our algorithms do not provide high-probability per-trajectory safety, they constitute a foundational advance in the most challenging adversarial setting without additional assumptions such as Slater's condition. Obtaining trajectory-wise guarantees under adversarial losses and constraints remains an interesting and important direction for future work.

As already mentioned, for both known and unknown transitions in fully adversarial CMDPs, improving our polynomial dependence on $S$, $H$, and $A$ remains an appealing direction for future research. While we handle a single constraint per episode, handling multiple constraints per episode, possibly with conflicting requirements, is an important practical challenge and an attractive extension. Lastly, developing model-free variants of our algorithms that do not require maintaining a confidence set for the transitions would be valuable.

## Broader Impact Statement

We propose efficient algorithms for constrained online learning in CMDPs that achieve optimal regret and hard violation bounds in adversarial environments. Thus, this strengthens the theoretical foundations of safe decision-making in CMDPs. One can apply our algorithms to domains such as healthcare, autonomous driving, and resource allocation, where respecting safety and budget constraints is critical.

Like any progress in adversarial learning, these methods could be misused in settings such as manipulative recommendation systems or exploitative bidding strategies. The contributions of this work are primarily theoretical and not intended for direct deployment in safety-critical systems without multiple layers of safeguards. Responsible application requires rigorous testing, domain-specific validation, and ethical oversight.

## Acknowledgments

The authors of this manuscript would like to thank Dr. Abhishek Sinha (Tata Institute of Fundamental Research, Mumbai, India) for his insightful comments and suggestions.

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

# A    Appendix

First, we present all omitted proofs, calculations, and algorithmic descriptions in the same order as in the main paper. We frequently make use of some algebraic inequalities throughout the section: (1) $(a+b)^2 \leq 2(a^2+b^2)$, $\forall a, b \in \mathbb{R}$; (2) $\sqrt{a+b} \leq \sqrt{a} + \sqrt{b}$, $\forall a, b \geq 0$, (3) $(a+b+c)^2 \leq 3(a^2+b^2+c^2)$, $\forall a, b, c \in \mathbb{R}$; and (4) $\sqrt{a+b+c} \leq \sqrt{a} + \sqrt{b} + \sqrt{c}$, $\forall a, b, c \geq 0$.

From the definition of $\varphi(\cdot)$, we observe that $\varphi'(\cdot)$ is non-decreasing. Additionally, we have $\widetilde{\nu}_t \geq 0$ due to the clipping and scaling of the constraints, which implies $\zeta_1 \leq \zeta_2 \leq \cdots \leq \zeta_t$, for any $t \geq 1$. Therefore, we obtain two relations, which we also use throughout this section: 1) $\sum_{\tau=1}^{t} \varphi'(\zeta_\tau) \leq t \cdot \varphi'(\zeta_t)$; and 2) $\sum_{\tau=1}^{t} \varphi'(\zeta_\tau)^2 \leq t \cdot \varphi'(\zeta_t)^2$. Lastly, we present the experimental results of `FAG-K`, `BAG-K`, and `FAG-U`.

## A.1    Upper bound of the surrogate regret in Section 4.1

We make use of Eqn. 26 and also of Eqn. 29 from Theorem 1.

$$\text{Regret}'_t(\rho^\star) \leq \sqrt{2}D\sqrt{\sum_{\tau=1}^{t} \|\nabla_\tau\|^2}$$

$$\leq \sqrt{2}D\sqrt{\sum_{\tau=1}^{t} (\omega L)^2 \left(1 + \varphi'(\zeta_\tau)\right)^2}$$

$$= \sqrt{2}D\omega L\sqrt{\sum_{\tau=1}^{t} \left(1 + \varphi'(\zeta_\tau)\right)^2}$$

$$\leq \sqrt{2}D\omega L\sqrt{\sum_{\tau=1}^{t} 2\left(1 + \varphi'(\zeta_\tau)^2\right)}$$

$$\leq 2D\omega L\sqrt{t} + 2D\omega L\sqrt{\sum_{\tau=1}^{t} \varphi'(\zeta_\tau)^2}$$

$$\leq 2D\omega L\sqrt{t}\left(1 + \varphi'(\zeta_t)\right).$$

## A.2    Proof of Lemma 1 in Section 4.2

The random variable $\mathbf{1}_t(s,a)$ is Bernoulli with success probability $\rho_t(s,a)$. We show via direct calculations,

$$\mathbb{E}_t[\widehat{\ell}_{t,h}(s,a)] = \mathbb{E}_t\left[\frac{\ell_{t,h}(s,a)}{\rho_t(s,a) + \Lambda_t}\mathbf{1}_t(s,a)\right]$$

$$= \frac{\ell_{t,h}(s,a)}{\rho_t(s,a) + \Lambda_t}\mathbb{E}_t\left[\mathbf{1}_t(s,a)\right]$$

$$= \frac{\ell_{t,h}(s,a)}{\rho_t(s,a) + \Lambda_t}\rho_t(s,a).$$

Also, we can show that

$$\mathbb{E}_t[\widehat{\ell}_{t,h}(s,a)^2] = \mathbb{E}_t\left[\frac{\ell_{t,h}(s,a)^2}{(\rho_t(s,a) + \Lambda_t)^2}\mathbf{1}_t(s,a)\right]$$

$$= \frac{\rho_t(s,a)}{(\rho_t(s,a) + \Lambda_t)^2}$$

$$\leq \frac{\rho_t(s,a) + \Lambda_t}{(\rho_t(s,a) + \Lambda_t)^2}$$

$$\leq \frac{1}{\rho_t(s,a) + \Lambda_t}.$$

Similarly, we can easily prove that $\mathbb{E}_t[\widehat{c}_{t,h}(s,a)] = \frac{c_{t,h}(s,a)}{\rho_t(s,a)+\Lambda_t}\rho_t(s,a)$ and $\mathbb{E}_t[\widehat{c}_{t,h}(s,a)^2] \leq \frac{1}{\rho_t(s,a)+\Lambda_t}$.

## A.3 Proof of Lemma 2 in Section 4.2

By direct calculations, we have:

$$
\begin{aligned}
&\ell_{t,h}(s,a) - \mathbb{E}_t[\widehat{\ell}_{t,h}(s,a)] \\
&= \ell_{t,h}(s,a) - \frac{\ell_{t,h}(s,a)}{\rho_t(s,a)+\Lambda}\rho_t(s,a) \\
&= \ell_{t,h}(s,a)\left(1 - \frac{\rho_t(s,a)}{\rho_t(s,a)+\Lambda}\right) \\
&= \frac{\Lambda\ell_{t,h}(s,a)}{\rho_t(s,a)+\Lambda},
\end{aligned}
$$

which is always non-negative and $\ell_{t,h}(s,a) - \mathbb{E}_t[\widehat{\ell}_{t,h}(s,a)] \leq \frac{\Lambda\ell_{t,h}(s,a)}{\rho_t(s,a)}$. Proceeding similarly, we also have:
$0 \leq c_{t,h}(s,a) - \mathbb{E}_t[\widehat{c}_{t,h}(s,a)] \leq \frac{\Lambda c_{t,h}(s,a)}{\rho_t(s,a)}$.

## A.4 Bounding the norm of the bias of the gradient estimate in Section 4.2

Recall that $\widehat{\ell}_t$ and $\widehat{c}_t$ are biased estimators of $\ell_t$ and $c_t$. It is clear from Eqn. 25 and Eqn. 38 that the bias vector $b_t$ should be given by:

$$
\begin{aligned}
b_t &= \mathbb{E}_t[\widehat{\nabla}_t] - \nabla_t \\
&= \omega\mathbb{E}_t[\widehat{\ell}_t] + \varphi'(\zeta_t)\omega\mathbb{E}_t[\widehat{c}_t \cdot \mathbf{1}_{\{\mathcal{C}_t>0\}}] - \omega\ell_t - \varphi'(\zeta_t)\omega c_t \cdot \mathbf{1}_{\{\langle\rho_t,\omega c_t\rangle>0\}} \\
&= \omega\big(\mathbb{E}_t[\widehat{\ell}_t] - \ell_t\big) + \omega\varphi'(\zeta_t)\big(\mathbb{E}_t[\widehat{c}_t \cdot \mathbf{1}_{\{\mathcal{C}_t>0\}}] - c_t \cdot \mathbf{1}_{\{\langle\rho_t,\omega c_t\rangle>0\}}\big).
\end{aligned}
$$

where $\mathbf{1}_{\{\mathcal{C}_t>0\}}$ and $\mathbf{1}_{\{\langle\rho_t,\omega c_t\rangle>0\}}$ are equal to 1 if $\mathcal{C}_t > 0$ and $\langle\rho_t,\omega c_t\rangle > 0$ respectively (0 otherwise). By the triangle inequality for norms, we have:

$$
\|b_t\| \leq \overbrace{\left\|\omega\big(\mathbb{E}_t[\widehat{\ell}_t] - \ell_t\big)\right\|}^{\|b_{t,\ell}\|} + \underbrace{\left\|\omega\varphi'(\zeta_t)\big(\mathbb{E}_t[\widehat{c}_t \cdot \mathbf{1}_{\{\mathcal{C}_t>0\}}] - c_t \cdot \mathbf{1}_{\{\langle\rho_t,\omega c_t\rangle>0\}}\big)\right\|}_{\|b_{t,c}\|}.
$$

$$
\implies \|b_t\| \leq \|b_{t,\ell}\| + \|b_{t,c}\|. \tag{69}
$$

Observe that $\|b_{t,\ell}\|^2 = \left\|\omega\big(\mathbb{E}_t[\widehat{\ell}_t] - \ell_t\big)\right\|^2 = \omega^2\left\|\mathbb{E}_t[\widehat{\ell}_t] - \ell_t\right\|^2$. Since the squared norm is the sum of the squared differences over all the $(s,a,h)$ components, we get from Lemma 2:

$$
\left\|\mathbb{E}_t[\widehat{\ell}_t] - \ell_t\right\|^2 = \sum_{(s,a,h)}\big(\mathbb{E}_t[\widehat{\ell}_{t,h}(s,a)] - \ell_{t,h}(s,a)\big)^2 = \sum_{(s,a,h)}\frac{\Lambda_t^2 \cdot \ell_{t,h}(s,a)^2}{(\rho_t(s,a)+\Lambda_t)^2}.
$$

Note that the losses are bounded, i.e., $\ell_{t,h}(s,a) \in [0,1]$, for all $t \in [T]$ and for all $h \in [H]^{-1}$. Also, in the earlier expression, the denominator is at least $\Lambda_t^2$, since $\rho_t(s,a) \geq 0$. Therefore, we have:

$$
\left\|\mathbb{E}_t[\widehat{\ell}_t] - \ell_t\right\|^2 \leq \sum_{(s,a,h)}\frac{\Lambda_t^2 \cdot 1^2}{\Lambda_t^2} = \sum_{(s,a,h)}1 \leq SHA.
$$

$$
\implies \|b_{t,\ell}\| \leq \omega\sqrt{SHA}. \tag{70}
$$

We will now upper bound the term $\|b_{t,c}\|$ in Eqn. 69. Decomposing $b_{t,c}$ without the norm as follows:

$$
\begin{aligned}
b_{t,c} &= \omega\varphi'(\zeta_t)\big(\mathbb{E}_t[\widehat{c}_t \cdot \mathbf{1}_{\{\mathcal{C}_t>0\}}] - c_t \cdot \mathbf{1}_{\{\langle\rho_t,\omega c_t\rangle>0\}}\big) \\
&= \omega\varphi'(\zeta_t)\big(\mathbb{E}_t[\widehat{c}_t \cdot \mathbf{1}_{\{\mathcal{C}_t>0\}}] - c_t \cdot \mathbf{1}_{\{\langle\rho_t,\omega c_t\rangle>0\}} + \mathbb{E}_t[\widehat{c}_t \cdot \mathbf{1}_{\{\langle\rho_t,\omega c_t\rangle>0\}}] - \mathbb{E}_t[\widehat{c}_t \cdot \mathbf{1}_{\{\langle\rho_t,\omega c_t\rangle>0\}}]\big) \\
&= \omega\varphi'(\zeta_t)\Big(\big(\mathbb{E}_t[\widehat{c}_t] - c_t\big) \cdot \mathbf{1}_{\{\langle\rho_t,\omega c_t\rangle>0\}} + \mathbb{E}_t[\widehat{c}_t \cdot (\mathbf{1}_{\{\mathcal{C}_t>0\}} - \mathbf{1}_{\{\langle\rho_t,\omega c_t\rangle>0\}})]\Big).
\end{aligned}
$$

Applying the triangle inequality on the norm of $\boldsymbol{b}_{t,\boldsymbol{c}}$,

$$\|\boldsymbol{b}_{t,\boldsymbol{c}}\| \leq \omega\varphi'(\zeta_t)\bigg(\big\|\big(\mathbb{E}_t[\widehat{\boldsymbol{c}}_t] - \boldsymbol{c}_t\big)\cdot\mathbf{1}_{\{\langle\rho_t,\omega\boldsymbol{c}_t\rangle>0\}}\big\| + \big\|\mathbb{E}_t[\widehat{\boldsymbol{c}}_t\cdot(\mathbf{1}_{\{\mathcal{C}_t>0\}} - \mathbf{1}_{\{\langle\rho_t,\omega\boldsymbol{c}_t\rangle>0\}})]\big\|\bigg).$$

We separately bound each term inside the parentheses. For the first term, we have

$$\big\|\big(\mathbb{E}_t[\widehat{\boldsymbol{c}}_t] - \boldsymbol{c}_t\big)\cdot\mathbf{1}_{\{\langle\rho_t,\omega\boldsymbol{c}_t\rangle>0\}}\big\| \leq \|\mathbb{E}_t[\widehat{\boldsymbol{c}}_t] - \boldsymbol{c}_t\|\cdot\mathbf{1}_{\{\langle\rho_t,\omega\boldsymbol{c}_t\rangle>0\}} \leq \|\mathbb{E}_t[\widehat{\boldsymbol{c}}_t] - \boldsymbol{c}_t\|.$$

Again from Lemma 2 and using the fact that $c_{t,h}(s,a) \in [-1,1]$, for all $t \in [T]$ and for all $h \in [H]^{-1}$,

$$\begin{aligned}
&\|\mathbb{E}_t[\widehat{\boldsymbol{c}}_t] - \boldsymbol{c}_t\|^2 \\
&= \sum_{(s,a,h)}\big(\mathbb{E}_t[\widehat{c}_{t,h}(s,a)] - c_{t,h}(s,a)\big)^2 \\
&= \sum_{(s,a,h)}\frac{\Lambda_t^2\cdot c_{t,h}(s,a)^2}{(\rho_t(s,a) + \Lambda_t)^2} \\
&\leq \sum_{(s,a,h)}\frac{\Lambda_t^2\cdot 1^2}{\Lambda_t^2} = \sum_{(s,a,h)}1 \leq SHA \\
&\implies \|\mathbb{E}_t[\widehat{\boldsymbol{c}}_t] - \boldsymbol{c}_t\| \leq \sqrt{SHA}.
\end{aligned} \tag{71}$$

On applying Jensen's inequality to the second term, we obtain:

$$\begin{aligned}
\big\|\mathbb{E}_t[\widehat{\boldsymbol{c}}_t\cdot(\mathbf{1}_{\{\mathcal{C}_t>0\}} - \mathbf{1}_{\{\langle\rho_t,\omega\boldsymbol{c}_t\rangle>0\}})]\big\| &\leq \mathbb{E}_t\bigg[\big\|\widehat{\boldsymbol{c}}_t\cdot(\mathbf{1}_{\{\mathcal{C}_t>0\}} - \mathbf{1}_{\{\langle\rho_t,\omega\boldsymbol{c}_t\rangle>0\}})\big\|\bigg] \\
&\leq \mathbb{E}_t\bigg[\|\widehat{\boldsymbol{c}}_t\|\cdot(\mathbf{1}_{\{\mathcal{C}_t>0\}} - \mathbf{1}_{\{\langle\rho_t,\omega\boldsymbol{c}_t\rangle>0\}})\bigg] \leq \mathbb{E}_t\bigg[\|\widehat{\boldsymbol{c}}_t\|\bigg].
\end{aligned}$$

Bounding the square $L^2$-norm of the sparse vector $\widehat{\boldsymbol{c}}_t$ (i.e., having only $H$ non-zero entries),

$$\begin{aligned}
\|\widehat{\boldsymbol{c}}_t\|^2 &= \sum_{h=0}^{H-1}\bigg(\frac{c_{t,h}(s,a)}{\rho_t(s,a) + \Lambda_t}\bigg)^2\cdot\mathbf{1}_t(s,a) \\
&= \sum_{h=0}^{H-1}\frac{c_{t,h}(s,a)^2}{(\rho_t(s,a) + \Lambda_t)^2} \\
&\leq \sum_{h=0}^{H-1}\frac{1}{\Lambda_t^2} = \frac{H}{\Lambda_t^2}. \implies \|\widehat{\boldsymbol{c}}_t\| = \frac{\sqrt{H}}{\Lambda_t}.
\end{aligned}$$

The final bound on the second term of the parentheses is

$$\big\|\mathbb{E}_t[\widehat{\boldsymbol{c}}_t\cdot(\mathbf{1}_{\{\mathcal{C}_t>0\}} - \mathbf{1}_{\{\langle\rho_t,\omega\boldsymbol{c}_t\rangle>0\}})]\big\| \leq \mathbb{E}_t\bigg[\|\widehat{\boldsymbol{c}}_t\|\bigg] \leq \frac{\sqrt{H}}{\Lambda_t}. \tag{72}$$

Using Eqn. 71 and Eqn. 72 we arrive at

$$\|\boldsymbol{b}_{t,\boldsymbol{c}}\| \leq \omega\varphi'(\zeta_t)\bigg(\sqrt{SHA} + \frac{\sqrt{H}}{\Lambda_t}\bigg). \tag{73}$$

Putting Eqn. 70 and Eqn. 73 in Eqn. 69, we have the final upper bound on the $L^2$-norm of the bias as

$$\|\boldsymbol{b}_t\| \leq \omega\sqrt{SHA} + \omega\varphi'(\zeta_t)\bigg(\sqrt{SHA} + \frac{\sqrt{H}}{\Lambda_t}\bigg) \leq \omega L + \omega\varphi'(\zeta_t)\bigg(L + \frac{\sqrt{H}}{\Lambda_t}\bigg). \tag{74}$$

### A.5 Upper bounding the component terms in Eqn. 40 of Section 4.2

We will use the Cauchy-Schwarz inequality, which is stated as: for all vectors $\boldsymbol{p}, \boldsymbol{q} \in \mathbb{R}$, $|\langle \boldsymbol{p}, \boldsymbol{q} \rangle| \leq \|\boldsymbol{p}\| \|\boldsymbol{q}\|$.

$$T_1 = \sum_{\tau=1}^{t} \langle \rho_\tau - \rho^\star, \mathbb{E}_\tau[\widehat{\nabla}_\tau] \rangle$$

$$\leq \sqrt{2}D \sqrt{\sum_{\tau=1}^{t} \left\| \mathbb{E}_\tau[\widehat{\nabla}_\tau] \right\|^2}$$

Setting $\Lambda_\tau = \omega\sqrt{H}$, for all $\tau \geq 1$, and from Eqn. 39, we have:

$$T_1 \leq \sqrt{2}D \sqrt{\sum_{\tau=1}^{t} \left\| \mathbb{E}_\tau[\widehat{\nabla}_\tau] \right\|^2}$$

$$\leq \sqrt{2}D \sqrt{\sum_{\tau=1}^{t} \left( \omega L + \omega\varphi'(\zeta_\tau) \left( L + \frac{\sqrt{H}}{\Lambda_\tau} \right) + \omega L \left( 1 + \varphi'(\zeta_\tau) \right) \right)^2}$$

$$\leq \sqrt{2}D \sqrt{\sum_{\tau=1}^{t} 3 \left( \omega^2 L^2 + \left( \omega\varphi'(\zeta_\tau) \left( L + \frac{\sqrt{H}}{\Lambda_\tau} \right) \right)^2 + \omega^2 L^2 \left( 1 + \varphi'(\zeta_\tau) \right)^2 \right)}$$

$$= \sqrt{6}D \sqrt{\sum_{\tau=1}^{t} \omega^2 L^2 + \left( \omega\varphi'(\zeta_\tau) \left( L + \frac{\sqrt{H}}{\Lambda_\tau} \right) \right)^2 + \omega^2 L^2 \left( 1 + \varphi'(\zeta_\tau) \right)^2}$$

$$\leq D\omega L\sqrt{6t} + \sqrt{6}D \sqrt{\sum_{\tau=1}^{t} \left( \omega L\varphi'(\zeta_\tau) + \varphi'(\zeta_\tau) \right)^2} + D\omega L\sqrt{6} \sqrt{\sum_{\tau=1}^{t} \left( 1 + \varphi'(\zeta_\tau) \right)^2}$$

$$\leq D\omega L\sqrt{6t} + \sqrt{6}D \sqrt{\sum_{\tau=1}^{t} \left( \omega L\varphi'(\zeta_\tau) + \varphi'(\zeta_\tau) \right)^2} + D\omega L\sqrt{6} \sqrt{\sum_{\tau=1}^{t} \left( 1 + \varphi'(\zeta_\tau) \right)^2}$$

$$\leq D\omega L\sqrt{6t} + \sqrt{12}D \sqrt{\sum_{\tau=1}^{t} \omega^2 L^2 \varphi'(\zeta_\tau)^2 + \varphi'(\zeta_\tau)^2} + D\omega L\sqrt{12} \sqrt{\sum_{\tau=1}^{t} 1 + \varphi'(\zeta_\tau)^2}$$

$$\leq D\omega L\sqrt{6t} + D\omega L\sqrt{12} \sqrt{\sum_{\tau=1}^{t} \varphi'(\zeta_\tau)^2} + D\sqrt{12} \sqrt{\sum_{\tau=1}^{t} \varphi'(\zeta_\tau)^2} + D\omega L\sqrt{12t} + D\omega L\sqrt{12} \sqrt{\sum_{\tau=1}^{t} \varphi'(\zeta_\tau)^2}$$

On putting $\omega = \frac{1}{2LD}$ and employing the non-decreasing property of $\varphi'(\cdot)$,

$$T_1 \leq \frac{\sqrt{6t}}{2} + \frac{\sqrt{12t}}{2}\varphi'(\zeta_t) + D\sqrt{12t} \cdot \varphi'(\zeta_t) + \frac{\sqrt{12t}}{2} + \frac{\sqrt{12t}}{2}\varphi'(\zeta_t)$$

$$= \sqrt{12t} \cdot \varphi'(\zeta_t) + \frac{\sqrt{6t}}{2} + \frac{\sqrt{12t}}{2} + D\sqrt{12t} \cdot \varphi'(\zeta_t). \tag{75}$$

By the Cauchy-Schwarz inequality, $|\langle \rho_\tau - \rho^\star, \boldsymbol{b}_\tau \rangle| \leq \|\rho_\tau - \rho^\star\| \cdot \|\boldsymbol{b}_\tau\| \leq D \|\boldsymbol{b}_\tau\|$. Therefore, we have the following upper bound on $T_2$:

$$
\begin{aligned}
T_2 &= \sum_{\tau=1}^{t} \langle \rho_\tau - \rho^\star, \boldsymbol{b}_\tau \rangle \\
&\leq D \sum_{\tau=1}^{t} \|\boldsymbol{b}_\tau\| \\
&\leq D \sum_{\tau=1}^{t} \omega L + \omega \varphi'(\zeta_\tau)\left( L + \frac{\sqrt{H}}{\Lambda_\tau} \right) \\
&\leq D\omega L t + D\omega L \sum_{\tau=1}^{t} \varphi'(\zeta_\tau) + D\omega\sqrt{H} \sum_{\tau=1}^{t} \frac{\varphi'(\zeta_\tau)}{\Lambda_\tau} \\
&\leq D\omega L t + D\omega L \sum_{\tau=1}^{t} \varphi'(\zeta_\tau) + D \sum_{\tau=1}^{t} \varphi'(\zeta_\tau) \\
&\leq \frac{t}{2} + \frac{t}{2} \cdot \varphi'(\zeta_t) + Dt \cdot \varphi'(\zeta_t).
\end{aligned}
\tag{76}
$$

### A.6 Bounding the term "Error" in Eqn. 50 of Section 5.1

From Eqn. 50, we have:

$$
\text{Error} = \sum_{\tau=1}^{t} \langle \rho_\tau - \widehat{\rho}_\tau, \nabla_\tau \rangle.
$$

Since $\|\nabla_t\| \leq \omega L \big(1 + \varphi'(\zeta_t)\big)$, and by the Cauchy-Schwarz inequality,

$$
\begin{aligned}
\text{Error} &\leq \sum_{\tau=1}^{t} \|\rho_\tau - \widehat{\rho}_\tau\| \cdot \|\nabla_\tau\| \\
&\leq \omega L \sum_{\tau=1}^{t} \|\rho_\tau - \widehat{\rho}_\tau\| \cdot \big(1 + \varphi'(\zeta_\tau)\big).
\end{aligned}
$$

Since $\widehat{\rho}_\tau$ is obtained from a transition function in the confidence set $\mathcal{P}_{i_\tau}$ (where $i_\tau$ is the epoch index for episode $\tau$), Lemma 3 implies that with probability at least $1 - 6\delta$:

$$
\sum_{\tau=1}^{t} \|\rho_\tau - \widehat{\rho}_\tau\|_1 \leq HS\sqrt{At \ln\left(\frac{SAt}{\delta}\right)},
$$

where $\|\cdot\|_1$ is the $L^1$-norm. Owing to the fact that $\|\rho_\tau - \widehat{\rho}_\tau\| \leq \|\rho_\tau - \widehat{\rho}_\tau\|_1$, we get:

$$
\sum_{\tau=1}^{t} \|\rho_\tau - \widehat{\rho}_\tau\| \leq \sum_{\tau=1}^{t} \|\rho_\tau - \widehat{\rho}_\tau\|_1 \leq HS\sqrt{At \ln\left(\frac{SAt}{\delta}\right)}.
$$

Combining all the above results, we have the final bound on "Error" as:

$$
\text{Error} \leq \omega L H S \sqrt{At \ln\left(\frac{SAt}{\delta}\right)} \cdot \big(1 + \varphi'(\zeta_t)\big).
\tag{77}
$$

### A.7 Upper bound of the surrogate regret in Section 5.1

From Eqn. 50 and Eqn. 77, we see:

$$
\begin{aligned}
\text{Regret}'_t(\rho^\star) &\leq \overbrace{\sum_{\tau=1}^{t} \langle \widehat{\rho}_\tau - \rho^\star, \nabla_\tau \rangle}^{\text{Reg}} + \overbrace{\sum_{\tau=1}^{t} \langle \rho_\tau - \widehat{\rho}_\tau, \nabla_\tau \rangle}^{\text{Error}} \\
&\leq \sqrt{2}D\sqrt{\sum_{\tau=1}^{t} \|\nabla_\tau\|^2} + \omega LHS\sqrt{At\ln\left(\frac{SAt}{\delta}\right)} \cdot \left(1 + \varphi'(\zeta_t)\right) \\
&\leq \sqrt{2}D\omega L\sqrt{\sum_{\tau=1}^{t} \left(1 + \varphi'(\zeta_\tau)\right)^2} + \omega LHS\sqrt{At\ln\left(\frac{SAt}{\delta}\right)} \cdot \left(1 + \varphi'(\zeta_t)\right) \\
&= 2D\omega L\sqrt{t} + 2D\omega L\sqrt{\sum_{\tau=1}^{t} \varphi'(\zeta_\tau)^2} + \omega LHS\sqrt{At\ln\left(\frac{SAt}{\delta}\right)} \cdot \left(1 + \varphi'(\zeta_t)\right) \\
&\leq 2D\omega L\sqrt{t} + 2D\omega L\sqrt{t} \cdot \varphi'(\zeta_t) + \omega LHS\sqrt{At\ln\left(\frac{SAt}{\delta}\right)} \cdot \left(1 + \varphi'(\zeta_t)\right) \\
&= \left(1 + \varphi'(\zeta_t)\right)\left(2D\omega L\sqrt{t} + \omega LHS\sqrt{At\ln\left(\frac{SAt}{\delta}\right)}\right).
\end{aligned}
\tag{78}
$$

### A.8 Bounding the components of Eqn. 60 in Section 5.2

We set $\Lambda_t = \omega\sqrt{H}$ for all $t \in [T]$, to bound each component of Eqn. 60. First, we bound the term "Reg":

$$
\begin{aligned}
\text{Reg} = \sum_{\tau=1}^{t} \langle \widehat{\rho}_\tau - \rho^\star, \widehat{\nabla}_\tau \rangle &\leq \sqrt{2}D\sqrt{\sum_{\tau=1}^{t} \left\|\widehat{\nabla}_\tau\right\|^2} \\
&\leq \sqrt{2}D\sqrt{\sum_{\tau=1}^{t} \frac{\omega^2 H}{\Lambda_\tau^2}\left(1 + \varphi'(\zeta_\tau)\right)^2} \\
&= \sqrt{2H}D\omega\sqrt{\sum_{\tau=1}^{t} \frac{\left(1 + \varphi'(\zeta_\tau)\right)^2}{\Lambda_\tau^2}} \\
&= \frac{\sqrt{2H}D\omega}{\omega\sqrt{H}}\sqrt{\sum_{\tau=1}^{t} \left(1 + \varphi'(\zeta_\tau)\right)^2} \\
&= D\sqrt{2}\sqrt{\sum_{\tau=1}^{t} \left(1 + \varphi'(\zeta_\tau)\right)^2} \\
&\leq D\sqrt{2}\sqrt{\sum_{\tau=1}^{t} 2\left(1 + \varphi'(\zeta_\tau)^2\right)} \\
&\leq 2D\sqrt{t} + 2D\sqrt{\sum_{\tau=1}^{t} \varphi'(\zeta_\tau)^2} \\
&\leq 2D\sqrt{t} + 2D\sqrt{t} \cdot \varphi'(\zeta_t) \\
&= 2D\sqrt{t}\left(1 + \varphi'(\zeta_t)\right).
\end{aligned}
\tag{79}
$$

We have the bound on "Error" from Eqn. 77 as,

$$\text{Error} = \sum_{\tau=1}^{t} \langle \rho_\tau - \widehat{\rho}_\tau, \nabla_\tau \rangle \leq \omega LHS \sqrt{At \ln\left(\frac{SAt}{\delta}\right)} \cdot \left(1 + \varphi'(\zeta_t)\right). \tag{80}$$

From Section 5.2, we know that $\|\boldsymbol{b}_t\| = \left\|\mathbb{E}_t[\widehat{\nabla}_t] - \nabla_t\right\| \leq \omega L + \omega\varphi'(\zeta_t)\left(L + \sqrt{H}/\Lambda_t\right)$. Now, we upper bound the term "Bias1",

$$\begin{aligned}
\text{Bias1} &= \sum_{\tau=1}^{t} \langle \widehat{\rho}_\tau, \nabla_\tau - \widehat{\nabla}_\tau \rangle \\
&= \sum_{\tau=1}^{t} \langle \widehat{\rho}_\tau, \nabla_\tau - \mathbb{E}_\tau[\widehat{\nabla}_\tau] \rangle + \sum_{\tau=1}^{t} \langle \widehat{\rho}_\tau, \mathbb{E}_\tau[\widehat{\nabla}_\tau] - \widehat{\nabla}_\tau \rangle \\
&= \underbrace{\sum_{\tau=1}^{t} \langle \widehat{\rho}_\tau, \mathbb{E}_\tau[\widehat{\nabla}_\tau] - \widehat{\nabla}_\tau \rangle}_{T_1} - \underbrace{\sum_{\tau=1}^{t} \langle \widehat{\rho}_\tau, \mathbb{E}_\tau[\widehat{\nabla}_\tau] - \nabla_\tau \rangle}_{T_2}. 
\end{aligned} \tag{81}$$

It is easily seen that $T_2 = \sum_{\tau=1}^{t} \langle \widehat{\rho}_\tau, \mathbb{E}_\tau[\widehat{\nabla}_\tau] - \nabla_\tau \rangle = \sum_{\tau=1}^{t} \langle \widehat{\rho}_\tau, \boldsymbol{b}_\tau \rangle$. By the Cauchy-Schwarz inequality:

$$\begin{aligned}
T_2 &\leq \sum_{\tau=1}^{t} \|\widehat{\rho}_\tau\| \cdot \|\boldsymbol{b}_\tau\| \\
&\leq \sum_{\tau=1}^{t} \|\boldsymbol{b}_\tau\| \\
&\leq \omega L + \omega\left(L + \sqrt{H}/\Lambda_\tau\right) \sum_{\tau=1}^{t} \varphi'(\zeta_\tau) \\
&\leq \omega L + \omega t\left(L + \sqrt{H}/\Lambda_t\right)\varphi'(\zeta_t).
\end{aligned}$$

Finally, we have on putting $\Lambda_t = \omega\sqrt{H}$ for all $t \in [T]$ that:

$$T_2 \leq \omega L + \omega Lt \cdot \varphi'(\zeta_t) + t \cdot \varphi'(\zeta_t).$$

We define a random variable $X_\tau = \langle \widehat{\rho}_\tau, \mathbb{E}_\tau[\widehat{\nabla}_\tau] - \widehat{\nabla}_\tau \rangle$ for all $\tau \in [t]$. Here, $\widehat{\rho}_\tau$ is $\mathcal{F}_{\tau-1}$-measurable and $\mathbb{E}_\tau[\cdot] = \mathbb{E}[\cdot \mid \mathcal{F}_{\tau-1}]$ is the conditional expectation. By construction, $\mathbb{E}_\tau[X_\tau] = 0$, so $\{X_\tau\}_{\tau=1}^{t}$ is a martingale difference sequence adapted to the filtration $\{\mathcal{F}_\tau\}$. For each $\tau$, we have $|X_\tau| \leq n_\tau$, where $n_\tau = \frac{2\omega}{\Lambda_\tau}\left(1 + \varphi'(\zeta_\tau)\right)$. Considering $\epsilon = \sqrt{2 \ln(2/\delta) \cdot \sum_{\tau=1}^{t} n_\tau^2}$, where $\delta \in (0,1)$, and applying the Azuma-Hoeffding inequality, we get: $T_1 \leq \frac{2\omega\left(1+\varphi'(\zeta_t)\right)}{\Lambda_t}\sqrt{2t \ln(2/\delta)}$. Therefore, we have the following upper bound on "Bias1":

$$\text{Bias1} \leq \frac{2\left(1 + \varphi'(\zeta_t)\right)}{\sqrt{H}}\sqrt{2t \ln(2/\delta)} - \omega L - \omega Lt \cdot \varphi'(\zeta_t) - t \cdot \varphi'(\zeta_t). \tag{82}$$

Before proceeding to bound the term "Bias2", we state and prove the following lemma, which is a slightly different form of Lemma 1 from Neu (2015). The proof draws inspiration from the techniques given in the proof of Lemma 1 of Neu (2015).

**Lemma 4.** *For all $t \in [T]$ and for all $h \in [H]^{-1}$, let $\{\alpha_{t,h}\}$ be a sequence such that each $\alpha_{t,h} \in [0, 2\Lambda_t]^{S \times A}$ is $\mathcal{F}_t$-measurable. Then, with probability at least $1 - \mathcal{O}(\delta)$, we get:*

$$\sum_{t=1}^{T} \sum_{(s,a,h)} \alpha_{t,h}(s,a)\left(\widehat{c}_{t,h}(s,a) - \frac{\rho_t(s,a)}{u_t(s,a)}c_{t,h}(s,a)\right) \leq H \ln\frac{H}{\delta}, \text{ and}$$

$$\sum_{t=1}^{T} \sum_{(s,a,h)} \alpha_{t,h}(s,a)\left(\widehat{\ell}_{t,h}(s,a) - \frac{\rho_t(s,a)}{u_t(s,a)}\ell_{t,h}(s,a)\right) \leq H \ln\frac{H}{\delta}.$$

*Proof.* Recall $\frac{x}{1+\frac{x}{2}} \leq \ln(1+x)$ for all $x \geq 0$. For any pair $(s,a)$ and let $\Delta = 2\Lambda_t$, we get:

$$
\begin{aligned}
\widehat{c}_{t,h}(s,a) &= \frac{c_{t,h}(s,a)}{u_t(s,a) + \Lambda_t}\mathbf{1}_t(s,a) \\
&\leq \frac{c_{t,h}(s,a)}{u_t(s,a) + \Lambda_t c_{t,h}(s,a)}\mathbf{1}_t(s,a) \\
&= \frac{\mathbf{1}_t(s,a)}{\Delta} \times \frac{2\Lambda\frac{c_{t,h}(s,a)}{u_t(s,a)}}{1 + \Lambda_t \frac{c_{t,h}(s,a)}{u_t(s,a)}} \\
&\leq \frac{1}{\Delta}\ln\left(1 + \frac{\Delta c_{t,h}(s,a)\mathbf{1}_t(s,a)}{u_t(s,a)}\right).
\end{aligned}
\tag{83}
$$

For all $h \in [H]^{-1}$, let us have

$$
\widehat{J}_{t,h} = \sum_{(s,a,h)} \alpha_{t,h}(s,a)\widehat{c}_{t,h}(s,a), \text{ and}
$$

$$
J_{t,h} = \sum_{(s,a,h)} \alpha_{t,h}(s,a)\frac{\rho_t(s,a)}{u_t(s,a)}c_{t,h}(s,a).
$$

By Eqn. 83, we have:

$$
\begin{aligned}
\mathbb{E}_t\left[\exp(\widehat{J}_{t,h})\right] &\leq \mathbb{E}_t\left[\exp\left(\sum_{(s,a,h)}\frac{\alpha_{t,h}(s,a)}{\Delta}\ln\left(1 + \frac{\Delta c_{t,h}(s,a)\mathbf{1}_t(s,a)}{u_t(s,a)}\right)\right)\right] \\
&\leq \mathbb{E}_t\left[\prod_{(s,a,h)}\left(1 + \frac{\alpha_{t,h}(s,a)c_{t,h}(s,a)\mathbf{1}_t(s,a)}{u_t(s,a)}\right)\right] \\
&= \mathbb{E}_t\left[1 + \sum_{(s,a,h)}\frac{\alpha_{t,h}(s,a)c_{t,h}(s,a)\mathbf{1}_t(s,a)}{u_{t,h}(s,a)}\right] \\
&= 1 + J_{t,h} \leq \exp(J_{t,h}).
\end{aligned}
$$

The second inequality is because $a\ln(1+b) \leq \ln(1+ab)$ for all $b \geq -1$ and $a \in [0,1]$, and we apply it with $a = \frac{\alpha_{t,h}(s,a)}{\Delta}$ which is in $[0,1]$ by the condition $\alpha_{t,h}(s,a) \in [0, 2\Lambda_t]$. The first arises since $\mathbf{1}_t(s,a)\mathbf{1}_t(s',a') = 0$ for any $s \neq s'$ or $a \neq a'$. On using Markov's inequality, we get:

$$
\begin{aligned}
\mathbb{P}\left[\sum_{t=1}^T(\widehat{J}_{t,h} - J_{t,h}) > \ln\left(\frac{H}{\delta}\right)\right] &\leq \frac{\delta}{H}\cdot\mathbb{E}\left[\exp\left(\sum_{t=1}^T(\widehat{J}_{t,h} - J_{t,h})\right)\right] \\
&= \frac{\delta}{H}\cdot\mathbb{E}\left[\exp\left(\sum_{t=1}^{T-1}(\widehat{J}_{t,h} - J_{t,h})\right)\mathbb{E}_T\left[\exp\left(\widehat{J}_{T,h} - J_{T,h}\right)\right]\right] \\
&\leq \frac{\delta}{H}\cdot\mathbb{E}\left[\exp\left(\sum_{t=1}^{T-1}(\widehat{J}_{t,h} - J_{t,h})\right)\right] \\
&\leq \cdots \leq \frac{\delta}{H}.
\end{aligned}
\tag{84}
$$

On applying the union bound over all $h \in [H]^{-1}$, we have the following holds with probability at least $1 - \mathcal{O}(\delta)$,

$$
\sum_{t=1}^T\sum_{(s,a,h)}\alpha_{t,h}(s,a)\left(\widehat{c}_t(s,a) - \frac{\rho_t(s,a)}{u_t(s,a)}c_{t,h}(s,a)\right) = \sum_{h=0}^{H-1}\sum_{t=1}^T(\widehat{J}_{t,h} - J_{t,h}) \leq H\ln\frac{H}{\delta}.
$$

Similarly, we can also show that $\sum_{t=1}^T\sum_{(s,a,h)}\alpha_{t,h}(s,a)\left(\widehat{\ell}_{t,h}(s,a) - \frac{\rho_t(s,a)}{u_t(s,a)}\ell_{t,h}(s,a)\right) \leq H\ln\frac{H}{\delta}$. $\qquad\square$

Recall the definitions of $\widehat{\nabla}_t$ and $\nabla_t$ from Section 5.2 and Section 5.1,

$$\widehat{\nabla}_t = \begin{cases} \omega\widehat{\boldsymbol{\ell}}_t + \varphi'(\zeta_t)\omega\widehat{\boldsymbol{c}}_t, & \text{if } \mathcal{C}_t > 0, \\ \omega\widehat{\boldsymbol{\ell}}_t, & \text{if } \mathcal{C}_t \leq 0, \end{cases} \quad \text{and} \quad \nabla_t = \begin{cases} \omega\boldsymbol{\ell}_t + \varphi'(\zeta_t)\omega\boldsymbol{c}_t, & \text{if } \langle\widehat{\rho}_t, \omega\boldsymbol{c}_t\rangle > 0, \\ \omega\boldsymbol{\ell}_t, & \text{if } \langle\widehat{\rho}_t, \omega\boldsymbol{c}_t\rangle \leq 0. \end{cases}$$

We perform the decomposition below for "Bias2":

$$\begin{aligned} \text{Bias2} &= \sum_{\tau=1}^{t} \langle\rho^\star, \widehat{\nabla}_\tau - \nabla_\tau\rangle \\ &= \sum_{\tau=1}^{t} \langle\rho^\star, \omega\widehat{\boldsymbol{\ell}}_\tau + \varphi'(\zeta_\tau)\omega\widehat{\boldsymbol{c}}_\tau \cdot \mathbf{1}_{\{\mathcal{C}_t > 0\}} - \omega\boldsymbol{\ell}_\tau - \varphi'(\zeta_\tau)\omega\boldsymbol{c}_\tau \cdot \mathbf{1}_{\{\langle\widehat{\rho}_t, \omega\boldsymbol{c}_t\rangle > 0\}}\rangle \\ &= \underbrace{\omega\sum_{\tau=1}^{t} \langle\rho^\star, \widehat{\boldsymbol{\ell}}_\tau - \boldsymbol{\ell}_\tau\rangle}_{L_1} + \underbrace{\omega\sum_{\tau=1}^{t} \varphi'(\zeta_\tau)\langle\rho^\star, \widehat{\boldsymbol{c}}_\tau \cdot \mathbf{1}_{\{\mathcal{C}_t > 0\}} - \boldsymbol{c}_\tau \cdot \mathbf{1}_{\{\langle\widehat{\rho}_t, \omega\boldsymbol{c}_t\rangle > 0\}}\rangle}_{L_2}. \end{aligned} \tag{85}$$

Note that $\rho^\star(s,a) \in [0,1] \subseteq [0, 2\Lambda_\tau]$ for $\Lambda_\tau \geq 1/2$. Since $\frac{\rho_\tau(s,a)}{u_\tau(s,a)} \leq 1$ and $\ell_{\tau,h}(s,a) \in [0,1]$, the term $\sum_{(s,a,h)} \rho^\star(s,a)\left(\frac{\rho_\tau(s,a)}{u_\tau(s,a)} - 1\right)\ell_{\tau,h}(s,a) \leq 0$. Thus,

$$\langle\rho^\star, \widehat{\boldsymbol{\ell}}_\tau - \boldsymbol{\ell}_\tau\rangle \leq \sum_{(s,a,h)} \rho^\star(s,a)\left(\widehat{\ell}_{\tau,h}(s,a) - \frac{\rho_\tau(s,a)}{u_\tau(s,a)}\ell_{\tau,h}(s,a)\right).$$

Using Lemma 4, with $\alpha_{\tau,h}(s,a) = \rho^\star(s,a)$, we have with probability at least $1 - \mathcal{O}(\delta)$:

$$L_1 = \omega\sum_{\tau=1}^{t} \langle\rho^\star, \widehat{\boldsymbol{\ell}}_\tau - \boldsymbol{\ell}_\tau\rangle \leq \omega\sum_{\tau=1}^{t} \sum_{(s,a,h)} \rho^\star(s,a)\left(\widehat{\ell}_{\tau,h}(s,a) - \frac{\rho_\tau(s,a)}{u_\tau(s,a)}\ell_{\tau,h}(s,a)\right) \leq \omega H \ln\frac{H}{\delta}. \tag{86}$$

We split $L_2 = \omega\sum_{\tau=1}^{t} \varphi'(\zeta_\tau)\langle\rho^\star, \widehat{\boldsymbol{c}}_\tau \cdot \mathbf{1}_{\{\mathcal{C}_t > 0\}} - \boldsymbol{c}_\tau \cdot \mathbf{1}_{\{\langle\widehat{\rho}_t, \omega\boldsymbol{c}_t\rangle > 0\}}\rangle$ into two components as:

$$L_2 = \omega\left(\sum_{\tau=1}^{t} \varphi'(\zeta_\tau)\langle\rho^\star, (\widehat{\boldsymbol{c}}_\tau - \boldsymbol{c}_\tau) \cdot \mathbf{1}_{\{\mathcal{C}_t > 0\}}\rangle - \sum_{\tau=1}^{t} \varphi'(\zeta_\tau)\langle\rho^\star, \boldsymbol{c}_\tau \cdot \left(\mathbf{1}_{\{\langle\widehat{\rho}_t, \omega\boldsymbol{c}_t\rangle > 0\}} - \mathbf{1}_{\{\mathcal{C}_t > 0\}}\right)\rangle\right). \tag{87}$$

First Term of $L_2$: Again, as $\frac{\rho_\tau(s,a)}{u_\tau(s,a)} \leq 1$, so:

$$\langle\rho^\star, \widehat{\boldsymbol{c}}_\tau - \boldsymbol{c}_\tau\rangle \leq \sum_{(s,a)} \rho^\star(s,a)\left(\widehat{c}_\tau(s,a) - \frac{\rho_\tau(s,a)}{u_\tau(s,a)}c_\tau(s,a)\right).$$

With probability at least $1 - \mathcal{O}(\delta)$, on using Lemma 4, we have:

$$\sum_{\tau=1}^{t} \varphi'(\zeta_\tau)\langle\rho^\star, \widehat{\boldsymbol{c}}_\tau - \boldsymbol{c}_\tau\rangle \leq \sum_{\tau=1}^{t} \varphi'(\zeta_\tau) \sum_{(s,a,h)} \rho^\star(s,a)\left(\widehat{c}_{\tau,h}(s,a) - \frac{\rho_\tau(s,a)}{u_\tau(s,a)}c_{\tau,h}(s,a)\right) \leq \varphi'(\zeta_t) \cdot H \ln\frac{H}{\delta}. \tag{88}$$

Second Term of $L_2$: Let $F_\tau = \mathbf{1}_{\{\langle\widehat{\rho}_t, \omega\boldsymbol{c}_t\rangle > 0\}} - \mathbf{1}_{\{\mathcal{C}_t > 0\}}$. Note that $|F_\tau| \leq 1$ for all $\tau$. Additionally, since $\rho^\star$ is a probability distribution and each component of $\boldsymbol{c}_\tau$ lies in $[-1, 1]$, we have $|\langle\rho^\star, \boldsymbol{c}_\tau\rangle| \leq 1$. Therefore, $\left|\sum_{\tau=1}^{t} \varphi'(\zeta_\tau)\langle\rho^\star, \boldsymbol{c}_\tau \cdot \left(\mathbf{1}_{\{\langle\widehat{\rho}_t, \omega\boldsymbol{c}_t\rangle > 0\}} - \mathbf{1}_{\{\mathcal{C}_t > 0\}}\right)\rangle\right| \leq \sum_{\tau=1}^{t} \varphi'(\zeta_\tau) \cdot |\langle\rho^\star, \boldsymbol{c}_\tau\rangle| \cdot |F_\tau| \leq \sum_{\tau=1}^{t} \varphi'(\zeta_\tau) \leq t \cdot \varphi'(\zeta_t)$.

Hence, we have an upper bound on $L_2$ as

$$L_2 \leq \omega\varphi'(\zeta_t) \cdot H \ln\frac{H}{\delta} - \omega t \cdot \varphi'(\zeta_t). \tag{89}$$

Combining Eqn. 86 and Eqn. 89 we obtain an upper bound on "Bias2":

$$\text{Bias2} \leq \omega H \ln\frac{H}{\delta} + \omega\varphi'(\zeta_t) \cdot H \ln\frac{H}{\delta} - \omega t \cdot \varphi'(\zeta_t). \tag{90}$$

## A.9 Results of `FAG-K`

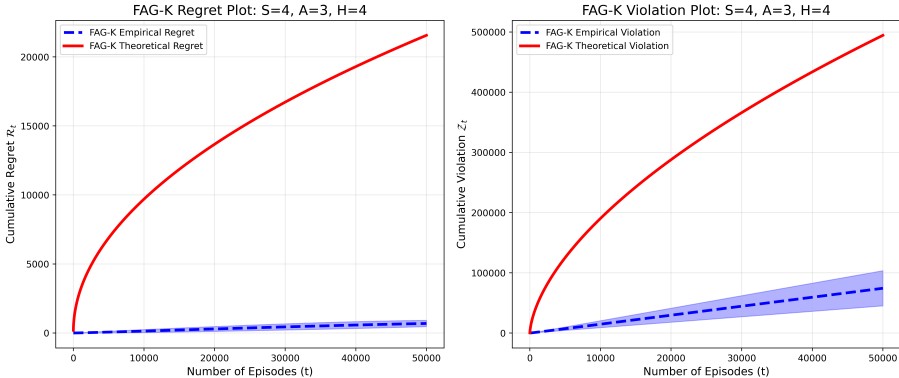

Figure 6: Theoretical regret (and violation) vs empirical regret (and violation) of `FAG-K` on a CMDP with $S = 4$, $A = 3$, $H = 4$. The empirical curves are averaged over five runs, with 95% confidence intervals.

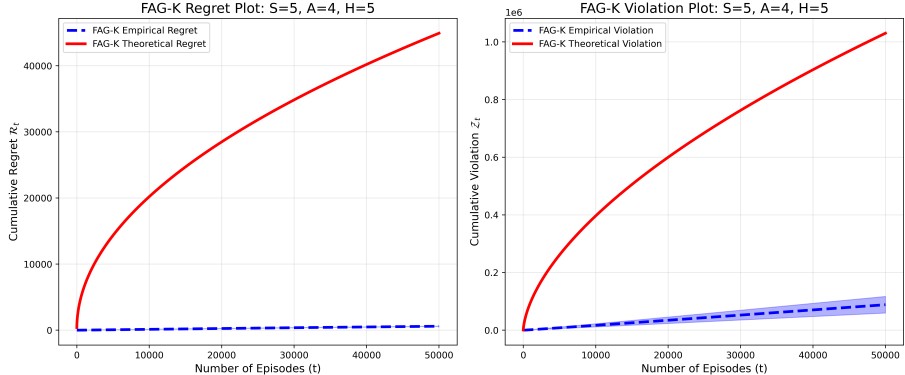

Figure 7: Theoretical regret (and violation) vs empirical regret (and violation) of `FAG-K` on a CMDP with $S = 5$, $A = 4$, $H = 5$. The empirical curves are averaged over five runs, with 95% confidence intervals.

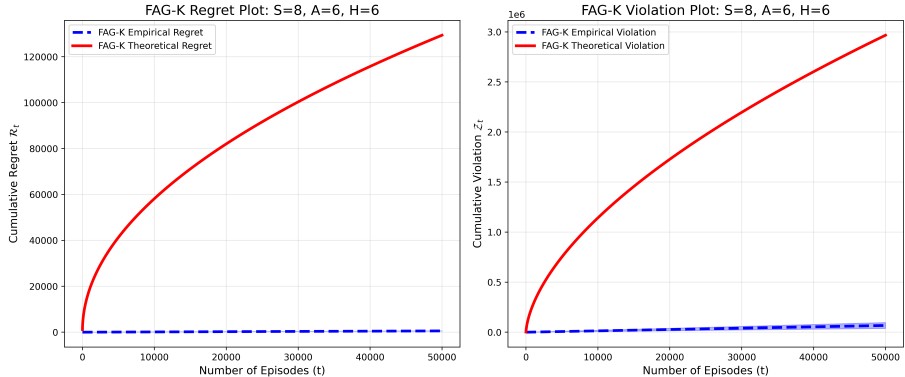

Figure 8: Theoretical regret (and violation) vs empirical regret (and violation) of `FAG-K` on a CMDP with $S = 8$, $A = 6$, $H = 6$. The empirical curves are averaged over five runs, with 95% confidence intervals.

### A.10 Results of BAG-K

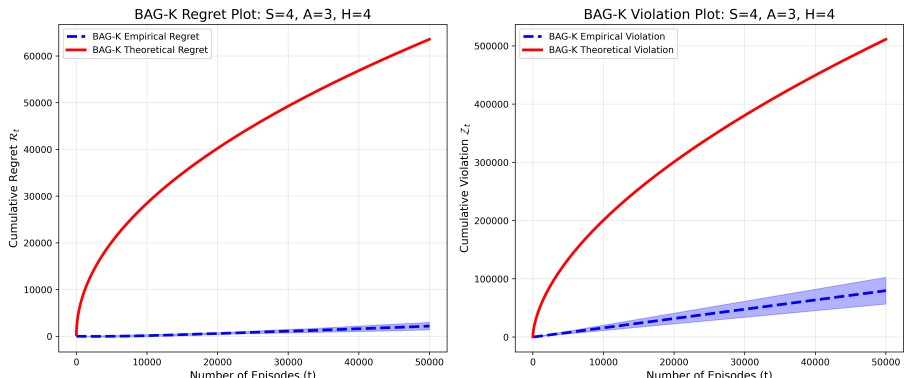

Figure 9: Theoretical regret (and violation) vs empirical regret (and violation) of BAG-K on a CMDP with $S = 4$, $A = 3$, $H = 4$. The empirical curves are averaged over five runs, with 95% confidence intervals.

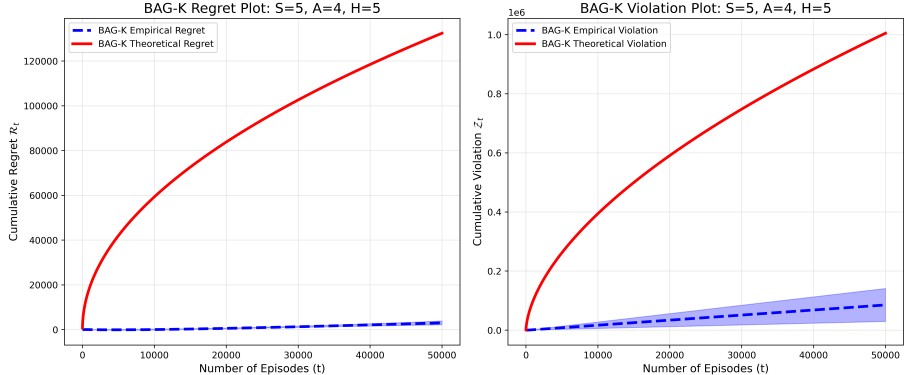

Figure 10: Theoretical regret (and violation) vs empirical regret (and violation) of BAG-K on a CMDP with $S = 5$, $A = 4$, $H = 5$. The empirical curves are averaged over five runs, with 95% confidence intervals.

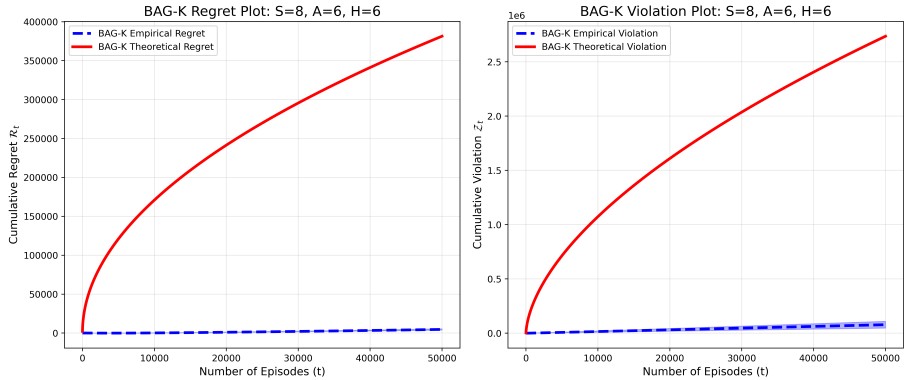

Figure 11: Theoretical regret (and violation) vs empirical regret (and violation) of BAG-K on a CMDP with $S = 8$, $A = 6$, $H = 6$. The empirical curves are averaged over five runs, with 95% confidence intervals.

### A.11  Results of `FAG-U`

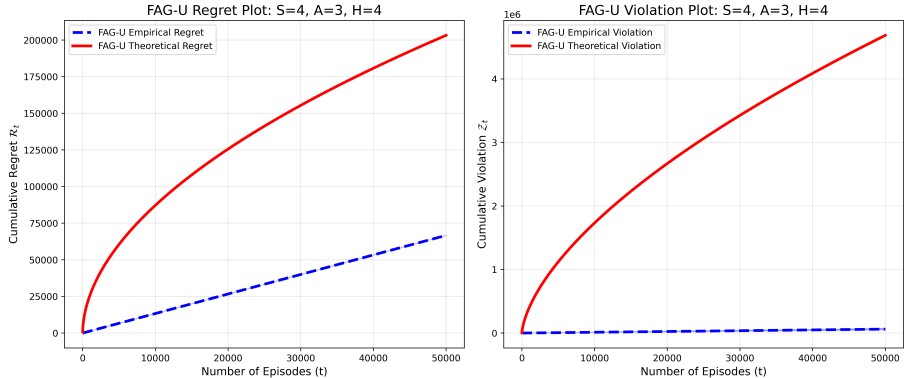

Figure 12: Theoretical regret (and violation) vs empirical regret (and violation) of `FAG-U` on a CMDP with $S = 4$, $A = 3$, $H = 4$. The empirical curves are averaged over five runs, with 95% confidence intervals.

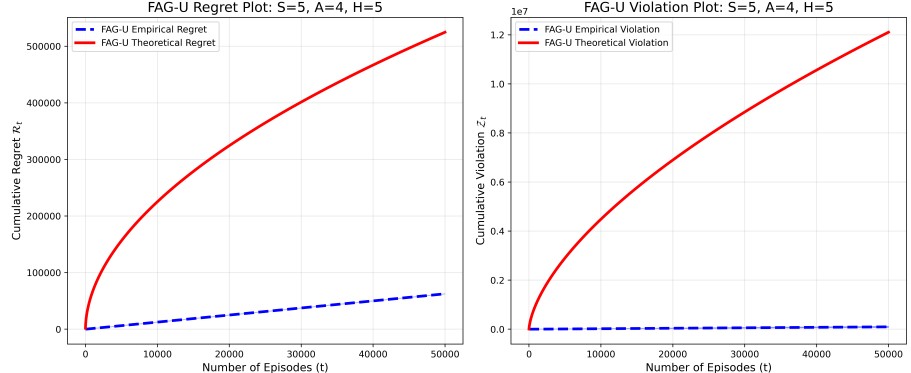

Figure 13: Theoretical regret (and violation) vs empirical regret (and violation) of `FAG-U` on a CMDP with $S = 5$, $A = 4$, $H = 5$. The empirical curves are averaged over five runs, with 95% confidence intervals.

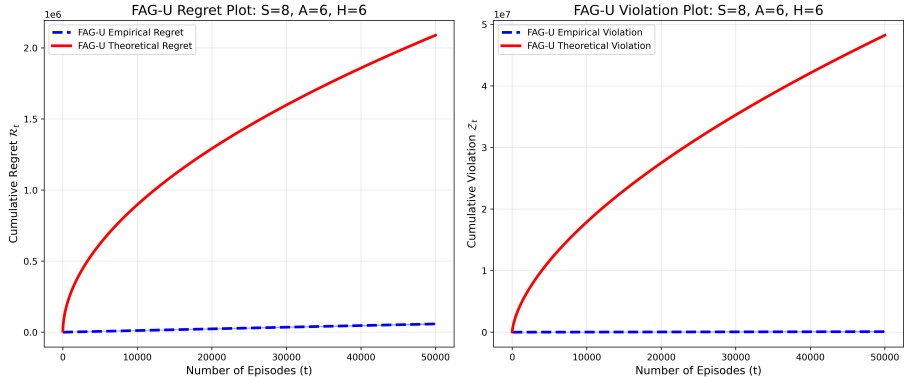

Figure 14: Theoretical regret (and violation) vs empirical regret (and violation) of `FAG-U` on a CMDP with $S = 8$, $A = 6$, $H = 6$. The empirical curves are averaged over five runs, with 95% confidence intervals.

