# OpenReview forum: "Optimal Regret and Hard Violation for Constrained Markov Decision Processes with Adversarial Losses and Constraints"
_TMLR — Accepted by TMLR_

### Review · Reviewer_GDPQ · 2025-11-26

**Summary Of Contributions:**

This paper investigates a regret bound of online learning in finite-horizon episodic constrained Markov decision process. The authors consider adversarial losses and constraints, bandit feedback, and unknown transitions.

The paper assumes realizability condition, convexity and Lipschitz assumption on objective and constraint function, which seems to be standard in the literature.

The paper considers CMDP problem as constrained online convex optimization problem.

To deal with unknown transition function , the authors consider constructing confidence set for the transition function as in Jin et al., 2020.

Finally, the authors show a $\tilde{\mathcal{O}} (\sqrt{T})$ regret bound.

**Strength**

1. The paper is clearly written and well structured, which makes it easy to follow. The comparison with related works are clearly stated with tables and figures.

2. The authors manage to derive a comparable regret bound under challenging settings by using AdaGrad Policy and the Lyapunov function motivated by stochastic drift-plus-penalty framework (Neely).

**Weakness**

1. The choice of surrogate loss function in equation (22) is not well explained. It is not clear why we need such scaling by $w$ and adoption of surrogate loss.


2. Under the adversarial and unknown transition model, the regret bound has been also analyzed in [R1, R2]


3. As mentioned by the authors, the dependency on the constants, $S,H,A$ does not seem to be optimal.

[R1] Ito, Shinji, et al. "Adapting to Stochastic and Adversarial Losses in Episodic MDPs with Aggregate Bandit Feedback." arXiv preprint arXiv:2510.17103 (2025).

[R2] Lancewicki, Tal, and Yishay Mansour. "Near-optimal regret using policy optimization in online mdps with aggregate bandit feedback." arXiv preprint arXiv:2502.04004 (2025).

**Audience:**

Yes

**Audience Explanation:**

The paper deals the regret bound of online learning and considers a challenging setting of adversarial loss and bandit feedback setting.

**Claims And Evidence:**

Yes

**Claims Explanation:**

The proof of the theorems and lemmas are provided in detail.

**Requested Changes:**

1. As mentioned in the weakness, the authors can clarify on the choice of using a surrogate loss function in equation (22) and the chocie of $w$.


2. Running experiments on simple toy examples can enhance the completeness of the paper.

---

### Review · Reviewer_3dru · 2025-12-15

**Summary Of Contributions:**

This paper presents a number of interesting theoretical results. It studies online learning in finite-horizon Constrained Markov Decision Processes (CMDPs) under a very challenging setting: adversarial losses and constraints, unknown transition dynamics, and bandit feedback. It introduces a principled reduction from CMDPs to constrained online convex optimization (COCO) using occupancy measures and Lyapunov-based surrogate losses. And apply and adapt COCO-style online algorithms, leading to simple, efficient algorithms that require only one projection per episode. Without relying on Slater’s condition or access to a strictly feasible policy, the proposed methods achieve optimal regret and hard cumulative constraint violation guarantees. The analysis systematically covers all combinations of known/unknown transitions and full/bandit feedback, closing several open gaps in the literature. Additionally, the paper develops biased subgradient estimators for the bandit setting, which may be of independent theoretical interest.

**Additional Comments:**

From the authors’ perspective, what are the most important open problems or promising future research directions that remain in this area? Since the problem setting in this paper is already the most demanding, are there theoretical aspects of CMDPs that the authors believe are especially important or challenging to address next?

**Audience:**

Yes

**Audience Explanation:**

This paper addresses a very challenging and less often studied setting in online learning and CMDPs, the theoretical results close multiple open gaps in the literature (adversarial constraints, unknown transitions, bandit feedback, no Slater condition) and are presented in a clear fashion. These results can be quite interesting to TMLR readers who follow topics such as reinforcement learning and constrained optimization.

**Claims And Evidence:**

Yes

**Claims Explanation:**

The paper provides precise problem formulations, explicit algorithms, and rigorous regret and hard constraint-violation analyses for each considered setting. Table 1 and Figure 1 are quite helpful in distinguishing the contributions of this paper from prior works and improves clarity of the paper. Key claims such as achieving optimal regret and violation bounds without relying on Slater’s condition are justified through detailed proofs and careful reductions from CMDPs to constrained online convex optimization. The authors systematically analyze all combinations of known/unknown transitions and full/bandit feedback, and the corresponding guarantees are clearly stated and matched with formal derivations. Overall, the theoretical arguments are internally consistent, grounded in established techniques, and sufficiently detailed to support the paper’s main claims.

**Requested Changes:**

I do not have major concerns, here are some potential changes that might further strengthen the paper:

- If possible, can be interesting to have a small illustrative example or schematic showing how a CMDP maps to a COCO to help the reader understand the idea faster.
- I understand this work is focused on theoretical contributions. But I wonder if the authors have insights on how people who work in fields such as deep RL can better utilize the findings of this work to design good practical algorithms. It seems to me that in practical problems, often people will simply not use the most challenging problem setup, although as the authors discussed in the introduction that a harder setting is in fact the more practical and realistic setting.
- Is there a time or space complexity analysis for the discussed algorithm? How might it work on a problem with large state/action spaces?
- Empirical results from e.g. a toy example environment can be interesting. Although the existings contributions already seem strong.

---

### Review · Reviewer_1kfR · 2025-12-27

**Summary Of Contributions:**

This paper studies online learning in finite-horizon, episodic constrained Markov decision processes (CMDPs). The paper focuses on finite state and action spaces in the loop-free setting. In each episode, an adversary chooses the loss and constraint functions. The goal is to achieve sublinear regret and also ensure sublinear hard constraint violation over the number of episodes.

The paper develops algorithms and theoretical guarantees for all four combinations of full/bandit feedback and known/unknown transitions. Its main technique is to reformulate the online CMDP as a constrained online convex optimization (COCO) problem over occupancy measures, and a key advantage is that the results do not rely on Slater’s condition or access to a strictly feasible policy.

Strengths:
- The paper identifies some gaps in the existing literature and proposes a principled approach to addressing them.
- The connection drawn between adversarial CDMPs and COCO, in particular, is clean and interesting.
- The works provides a fairly comprehensive treatment across different regimes, with guarantees for both regret and constraint violation.

Weaknesses:
- The results are limited to finite state/action spaces and the loop-free setting, and it is unclear how they extend to more general CMDPs.
- Some of the main technical ingredients are fairly standard or have been used in similar settings. This is not inherently a drawback, but it does make the degree of technical novelty somewhat limited.
- The empirical evaluation is nice to have (especially as a sanity check), but it is relatively small in scale to assess the practical scalability of the approach.

**Audience:**

Yes

**Audience Explanation:**

The paper highlights a few gaps in the literature on online learning in CMDPs. The central question, which assumes adversarial losses and constraints, unknown transitions, and bandit feedback, among others, is quite challenging, and the paper addresses it through a clean COCO reformulation. The results should be of interest to at least some of TMLR’s audience working on RL theory, constrained RL, and online learning.

**Broader Impact Concerns:**

Theory paper; broader impact sufficiently discussed.

**Claims And Evidence:**

Yes

**Claims Explanation:**

The main results are well-supported by clear, formal theorems.

**Requested Changes:**

Regarding modeling/problem formulation:
- The paper evaluates performance through both regret and cumulative hard constraint violation. How does this connect to the standard formulation where the learner minimizes loss subject to a constraint/budget? How should one interpret optimality when regret and constraint violation trade off?
- The loss values are in $[0,1]$, whereas the constraint values are in $[-1,1]$? It would be helpful to motivate this modeling choice. What does negative constraint violation mean?
- In terms of hard vs soft constraint violations, it makes sense that violations should not cancel out across episodes. That said, one may also interpret “hard constraint” as trajectory-level guarantees (e.g., with high probability the agent avoids unsafe events such as hitting a wall). Some discussion on this might be helpful.
- A brief motivating example of when adversarially chosen loss and constraint functions arise in CMDPs would help strengthen the framing.

Clarity:
- In a few places, such as the beginning of Section 4, it is suggested that when the transition function is known, there is no randomness linked with the next state. This can be misleading because the transition is still stochastic. Could you clarify?
- In the last sentence of the first paragraph of Section 3.3, it is unclear why $\nu_t(x_t) \le 0$.

Others:
- In Figures 2–4, the blue curves do not visually appear sublinear over $T$ as claimed (unless I am missing something). This may benefit from additional explanation.
- If possible, a discussion on any conjectures of lower bounds on $S$, $A$, and $H$ would be helpful.

Grammar:
- abstract: "Thus, creating ..." is not a complete sentence.
- below Eqn (40): "deffered"
- Table 2: "Algortihm"

---

> ### Author Response · Authors · 2026-01-09
> **Response to Reviewer 1kfR (Part 1 of 2)**
>
> We sincerely thank the reviewer for recognizing the contributions and the importance of our work. Specifically, the "clean and interesting" connection between adversarial CMDPs and COCO, and the comprehensive treatment of the different regimes. We also express our gratitude to the reviewer for their insightful comments and constructive suggestions.
>
> We have thoroughly and carefully incorporated the reviewer's suggested revisions to the best of our ability. All changes are in orange in the revised PDF. We provide our responses to each comment (from weaknesses to requested changes) in two parts. The first part is as follows.
>
> Abbreviations used: Wx = Weakness #x, RCx = Requested Changes #x.
>
> **W1--Results limited to finite state/action spaces:** We thank the reviewer for this comment. Our work focuses on the tabular and loop-free setting. As this is the first work to solve the open question of achieving $\mathcal{\widetilde{O}}(\sqrt{T})$ regret and hard violation for adversarial CMDPs with unknown transitions, bandit feedback, without Slater's condition, and no access to a feasible policy--we followed the standard path of establishing optimality in the tabular setting first (as in seminal works [1, 2]). Extending such guarantees to general CMDPs with function approximation requires complex mechanisms (e.g., covering numbers or Eluder dimensions) to handle estimation in large or continuous spaces. We have briefly discussed such extensions in "Future Directions" (Section 9), but we view our work as the requisite foundational step for this line of research.
>
> **W2--Standard technical ingredients used:** We thank the reviewer for this comment. While we use established tools such as confidence sets [3] and Lyapunov's [4], the primary novelty lies in the reduction framework itself. By directly mapping the CMDP problem to a COCO instance, we bypass the complex primal-dual updates and assumptions often required in prior work. This reduction allows for a much simpler algorithm (a single projection) while simultaneously closing the theoretical gap in the most demanding setting (adversarial + bandit + unknown transition). We believe the "clean and interesting" nature of this connection, as kindly noted by the reviewer, is a key conceptual contribution that would simplify future analysis in this domain.
>
> **W3--Small-scale empirical evaluation:** We thank the reviewer for this comment. As the reviewer kindly noted, the primary purpose of the experiments in this theoretical work is to serve as a sanity check. The current simulations on multiple CMDP instances confirm that our algorithms consistently perform well within the theoretical envelopes. We clarified in the earlier revision that these experiments are intended to validate the theoretical bounds rather than demonstrate scalability to high-dimensional tasks that require function approximation.
>
> **RC1--Regret, CCV, and tradeoff interpretation:** We sincerely thank the reviewer for this comment. In the standard online CMDP formulation [5], the learner aims to minimize the cumulative loss and the cumulative constraint violation jointly. Our formulation precisely aligns with this standard. Regarding the tradeoff, our algorithm manages it adaptively via the virtual queue $\zeta\_{t}$. When violations occur, the price of violation increases via $\varphi^{\prime}(\zeta\_{t})$, forcing the algorithm to prioritize the constraint. Optimality is defined as achieving sublinear bounds for both metrics simultaneously. The tradeoff arises because aggressive loss minimization might violate constraints more often; our $\mathcal{\widetilde{O}}(\sqrt{T})$ bounds show that both can be sublinear, meaning the learner approaches optimality without unbounded violations. This framework connects directly to budget-constrained settings (e.g., auction bidding): regret quantifies the lost utility, and violation tracks the budget overrun. For interpretation, sublinear regret/violation implies that the average per-episode performance converges to the optimal feasible policy.
>
> In the revised PDF, we have added the above clarifications in Sections 7 and 8 (in orange). We sincerely hope that our response and revision fully address the reviewer's comments.
>
> The second part contains detailed responses to the remaining comments from the reviewer and is provided in the box below.
>
> [1] T. Jaksch et al. "Near-optimal regret bounds for reinforcement learning (2010)."
>
> [2] C. Jin et al. "Is Q-learning provably efficient (2018)."
>
> [3] C. Jin et al. "Learning adversarial MDPs with bandit feedback and unknown transition (2020)."
>
> [4] Sinha and Vaze. "Optimal algorithms for OCO with adversarial constraints (2024)."
>
> [5] F.E. Stradi et al. "Online learning in cmdps: handling stochastic and adversarial constraints (2024)."

---

> > ### Author Response · Authors · 2026-01-09
> > **Response to Reviewer 1kfR (Part 2 of 2)**
> >
> > Below is the second part of our response to the reviewer's comments. All revisions (e.g., newly added texts) are in orange.
> >
> > **RC2--Range of loss and constraint values:** Our thanks to the reviewer for this comment. The losses are in $[0, 1]$ and model non-negative, bounded costs. It is standard practice in adversarial MDP/CMDPs [3, 5] to ensure tractable regret analysis, as it bounds the per-step regret contribution and prevents unbounded adversarial manipulation. Non-negative losses also reflect the intuition that actions incur penalties rather than arbitrary gains in the minimization objective, consistent with our regret definition. We model the constraint function such that a value $\leq 0$ implies satisfaction. The range $[-1, 1]$ allows for the concept of "slack" or "budget".
> >
> > **Meaning of Negative Violation:** A negative sum of constraint costs in an episode implies the agent operated strictly within the safety budget. While soft violation formulations allow these "savings" (i.e., negative values) to cancel out future "overspending" (i.e., positive values), our hard violation metric uses the operator $(\cdot)^{+}$, ensuring that "savings" from safe episodes cannot offset the costs of unsafe ones. The negative values in the input vector $\pmb{c}\_{t}$ are necessary to define the feasible region, but only the positive excursions count towards the violation metric $\mathcal{Z}\_{T}$.
> >
> > We have accordingly revised the first paragraph of Section 3.1 in the revised PDF. We firmly believe that our response and revision answer the reviewer's question.
> >
> > **RC3--Discussion on hard constraint violation as trajectory-level guarantees:** We thank the reviewer for mentioning this point. Our work focuses on bounding the cumulative hard-violation metric, ensuring that negative violations never offset positive violations. We acknowledge that achieving per-trajectory safety with high probability is a more stringent goal, and might require different algorithmic techniques and assumptions. While our current bounds do not provide such trajectory-wise guarantees, our framework and results represent a critical step toward understanding the most challenging adversarial setting without relying on Slater's condition or strictly feasible policies.
> >
> > We believe the reviewer's suggestion enriches the discussion of safety criteria, and we have included a clarifying statement at the end of Section 3.2 in the revised PDF. Inspired by the reviewer's comment, we have also highlighted this point as an intriguing future work in Section 9 of the revised PDF.
> >
> > **RC4--Motivating example of adversarial CMDPs:** Following the reviewer's suggestion, we have added a brief and dedicated paragraph in Section 3.1 of the revised PDF right before Eqn. (1). Two real-life motivating examples--network routing and online advertising auction--are given to describe when adversarially chosen loss and constraint functions arise in CMDPs.
> >
> > **RC5--Known transition function:** Our gratitude to the reviewer for this comment. We meant that there is no uncertainty about the transition function $\mathcal{P}$, not that the environment is deterministic. The next state $s_{h + 1}$ is indeed stochastic. In the revised PDF, we have corrected this to: "When the transition function $\mathcal{P}$ is known for the CMDP $\mathcal{M}$, there is no model uncertainty regarding $\mathcal{P}$, but there will be randomness linked with the next-state $s_{h + 1}$ in an episode $t \in [T]$." We have also added $s\_{h + 1} \sim \mathcal{P}(\cdot \mid s\_{h}, a\_{h})$ in Eqn. (15) and Eqn. (16).
> >
> > **RC6--Typo $\nu\_{t}(x\_{t}) \leq 0$:** We have corrected the typo to $\nu\_{t}(x\_{t})$.
> >
> > **RC7--On the blue curves:** In Figures 3-5, the red line represents the theoretical worst-case $\mathcal{\widetilde{O}}(\sqrt{T})$ bound, which is sublinear. The blue curves are always below the red curves. Since "empirical" $\leq$ "theoretical" = $C\sqrt{T}$, for some $C > 0$, the empirical regret is necessarily sublinear. The visual linearity often arises because $\sqrt{T}$ looks relatively linear for large $T$ without a log-log plot. To clarify this, we added an explanatory paragraph at the end of Section 6 in the revised PDF.
> >
> > **RC8--Conjectures of lower bounds on $S, A, H$:** As already noted in Section 7, lower bounds for adversarial CMDPs are not yet established. However, based on adversarial MDPs scaling as $\Omega(\sqrt{H^{2}SAT})$, and COCO scaling as $\Omega(\sqrt{T})$, we conjecture the tight lower bound for adversarial CMDPs is likely $\Omega(H\sqrt{SAT})$. We have added these lines at the end of Section 7 in the revised PDF.
> >
> > **RC9--Grammar/Typos:** In the revised PDF, we have corrected the grammar and fixed the typos. In the Abstract, we write: "..., thus creating simple and elegant algorithms...". We fixed the typo below Eqn. (40) as "deferred." We fixed the typo in Table 2 to "Algorithm."
> >
> > We firmly believe all revisions further strengthen the paper and resolve all the questions.

---

### Decision · Action_Editor_hMPz · 2026-03-16

**Recommendation:** Accept as is

**Audience:**

Yes

**Audience Explanation:**

The paper addresses several gaps in the literature on online learning in constrained Markov decision processes (CMDPs). The results may be of interest to researchers working on reinforcement learning theory, online learning, and constrained optimization.

**Claims And Evidence:**

Yes

**Claims Explanation:**

The paper presents an analysis of regret bounds for online learning in finite-horizon episodic constrained Markov decision processes, which appears to be novel in the literature. The analysis is supported by well-written proofs that are clear and readily verifiable.